EMBO
Molecular Medicine

# Lactylation-driven FTO targets CDK2 to aggravate microvascular anomalies in diabetic retinopathy

Xue Chen [1,4]✉, Ying Wang[1,4], Jia-Nan Wang[1,4], Yi-Chen Zhang[1], Ye-Ran Zhang[1], Ru-Xu Sun[1], Bing Qin [2], Yuan-Xin Dai[3], Hong-Jing Zhu[1], Jin-Xiang Zhao[2], Wei-Wei Zhang[1], Jiang-Dong Ji[1], Song-Tao Yuan[1], Qun-Dong Shen [3] & Qing-Huai Liu [1]✉

## Abstract

Diabetic retinopathy (DR) is a leading cause of irreversible vision loss in working-age populations. Fat mass and obesity-associated protein (FTO) is an $N^6$-methyladenosine ($m^6A$) demethylase that demethylates RNAs involved in energy homeostasis, though its influence on DR is not well studied. Herein, we detected elevated FTO expression in vitreous fibrovascular membranes of patients with proliferative DR. FTO promoted cell cycle progression and tip cell formation of endothelial cells (ECs) to facilitate angiogenesis in vitro, in mice, and in zebrafish. FTO also regulated EC-pericyte crosstalk to trigger diabetic microvascular leakage, and mediated EC–microglia interactions to induce retinal inflammation and neurodegeneration in vivo and in vitro. Mechanistically, FTO affected EC features via modulating CDK2 mRNA stability in an $m^6A$-YTHDF2-dependent manner. FTO up-regulation under diabetic conditions was driven by lactate-mediated histone lactylation. FB23-2, an inhibitor to FTO's $m^6A$ demethylase activity, suppressed angiogenic phenotypes in vitro. To allow for systemic administration, we developed a nanoplatform encapsulating FB23-2 and confirmed its targeting and therapeutic efficiency in mice. Collectively, our study demonstrates that FTO is important for EC function and retinal homeostasis in DR, and warrants further investigation as a therapeutic target for DR patients.

Keywords Diabetic Retinopathy (DR); FTO; $N^6$-Methyladenosine ($m^6A$); Vascular Endothelial Cells; Pericyte
Subject Categories Molecular Biology of Disease; Vascular Biology & Angiogenesis

## Introduction

Diabetic retinopathy (DR), a major microvascular complication of diabetes, is emerging as a leading threat to vision in working-age

populations. Clinically, DR is divided into the early and the advanced stages based on its disease course. Non-proliferative DR (NPDR) represents the early stage of DR, which is characterized by increased vascular permeability and capillary occlusions. Proliferative DR (PDR) is the advanced form of DR with the clinical hallmark of neovascularization. Patients may be asymptomatic in NPDR but may experience severe vision loss in PDR when vitreous hemorrhage or tractional retinal detachment happens. Retinal microvasculopathy, inflammation and neurodegeneration are major pathological features of DR (Wang et al, 2018). Microvascular endothelial cells (ECs) are major targets of hyperglycemic injuries and are mostly studied in DR (Yang et al, 2022). Loss of cell-to-cell contact in EC monolayers contributes to increased blood-retinal barrier (BRB) permeability, and its transformation into tip cells leads to sprouting angiogenesis (Ren et al, 2022). Pericyte is another major cellular constituent in neural retinal microvessels. Formation, maturation, and stabilization of the micro-vasculatures require EC-pericyte interactions, which are perturbed in DR, resulting in BRB rupture and other micro-angiopathies (Huang, 2020). Interrupted homeostasis in diabetic retina induces microglia activation and inflammatory responses, which drive sustained vascular damages, further resulting in increased vascular permeability and angiogenesis (Huang, 2020; Ren et al, 2022). Retinal neurodegeneration, especially axonal degeneration of retinal ganglion cell (RGC), is one of the earliest events in DR progression (Ren et al, 2022). Loss of RGCs, the most sensitive retinal neurons to diabetes-induced stress, leads to severe visual impairments. Current treatments of DR mainly include intra-vitreal injection of anti-vascular endothelial growth factor (VEGF) drugs, laser photocoagulation and vitrectomy. Anti-VEGF agent, currently the mainstay of therapy for both NPDR and PDR, requires persistent injections and only targets retinal neovasculatures. Many DR patients even show inadequate response to anti-VEGF medications after a long period of treatment (Gonzalez et al, 2016). Thus, fresh insights into DR pathology are needed for identification of novel therapeutic targets.

$N^6$-methyladenosine ($m^6A$) modification, mainly catalyzed by $m^6A$ methyltransferase complex (writers), removed by $m^6A$ demethylases (erasers), and recognized by $m^6A$-binding proteins

[1]Department of Ophthalmology, The First Affiliated Hospital of Nanjing Medical University, Nanjing Medical University, Nanjing, China. [2]Department of Ophthalmology, The Affiliated Suqian First People's Hospital of Nanjing Medical University, Suqian, China. [3]Department of Polymer Science and Engineering and Key Laboratory of High-Performance Polymer Materials and Technology of MOE, School of Chemistry and Chemical Engineering, Nanjing University, Nanjing, China. [4]These authors contributed equally: Xue Chen, Ying Wang, Jia-Nan Wang. ✉E-mail: drcx1990@vip.163.com; liuqh@njmu.edu.cn

(readers), is one of the most prevalent and abundant internal modifications of mRNA in eukaryotes (Di Timoteo et al, 2020). It regulates mRNA metabolism, including splicing, stability, translation and nuclear export (Huang et al, 2018; Shi et al, 2020; Wang et al, 2014). Increasing evidence has implied the pathological involvement of aberrant m6A modification levels and expression of its modulators (writers, erasers and readers) in DR (Chen et al, 2022; Suo et al, 2022), implying the crucial roles of m6A modification in DR pathogenesis. The fat mass and obesity-associated (FTO) protein, which mediates oxidative demethylation of different RNA species, acts as a regulator of fat mass, adipogenesis and energy homeostasis (Jia et al, 2011; Wang et al, 2015a; Wei et al, 2018; Zhou et al, 2015). Although the clinical association between FTO and diabetes has long been discussed, the roles and regulatory networks of FTO in DR remain unclear.

Herein, we revealed that FTO promotes endothelial cell cycle progression and tip cell formation to facilitate angiogenesis in DR. We also found that FTO triggers diabetes-induced microvascular leakage by regulating EC-pericyte crosstalk. FTO also mediated EC-microglia interactions to interrupt retinal homeostasis by inducing microglia activation and neurodegeneration. Mechanistically, FTO regulated diabetic retinal phenotypes through its demethylation activity by modulating *CDK2* mRNA stability with YTH domain-containing family protein 2 (YTHDF2) as the reader. FTO up-regulation in ECs under diabetic conditions was triggered by lactic acid via histone lactylation. FB23-2, which directly binds to FTO and selectively inhibits FTO's m6A demethylase activity, suppressed diabetes induced endothelial phenotypes. We also developed a novel macrophage membrane coated and poly (lactic-co-glycolic acid) (PLGA)-1, 1′-dioctadecyl-3, 3, 3′, 3′-tetramethylindocarbocyanine perchlorate (Dil) based nanoplatform encapsulating FB23-2 for systemic administration. Targeting and therapeutic efficiencies of this nanoplatform have been evaluated and confirmed, indicating its promising role as a treatment agent for DR.

# Results

## Identification of FTO as a potential regulator of DR

M6A modification was involved in various pathological processes, while its role in DR is not fully understood. Herein, to determine whether m6A levels were altered in DR, we exposed human umbilical vein endothelial cells (HUVECs) to high glucose (25 mM) in vitro. Cell viability of HUVECs was intact upon high glucose treatment (Appendix Fig. S1). For in vivo analyses, we used the streptozotocin (STZ)-induced diabetic mice, which develop retinal vascular leakage without neovascularization (Feit-Leichman et al, 2005). Both dot blot and m6A RNA methylation quantification assays detected reduced m6A contents in total RNAs of HUVECs treated with high glucose (Fig. 1A,B), as well as in neural retinas collected from STZ mice (Fig. 1C,D).

We next aimed to identify the specific m6A regulator responsible for the decreased m6A levels in high glucose treated HUVECs and STZ retinas. Expression of m6A writers (METTL3, METTL14 and WTAP) and erasers (ALKBH5 and FTO) was detected. Both qPCR and immunoblotting analyses demonstrated consistent up-regulation of *FTO* mRNA and protein upon high glucose treatment in vitro and in vivo (Fig. 1E,H). To further test whether FTO directly regulates m6A

modification in ECs, we overexpressed FTO in HUVECs using lentivirus containing coding sequence (CDS) of human *FTO* gene and tagged with FLAG (L-FTO). Immunoblotting detected FLAG expression and significantly increased FTO expression in HUVECs transduced with L-FTO (Appendix Fig. S2A). Immunofluorescence staining further demonstrated that the overexpressed FTO-FLAG fusion protein is mainly located in the nuclei (Appendix Fig. S2B). M6A dot blot assay identified reduced m6A level in total RNAs of HUVECs transduced with L-FTO (Fig. 1I), supporting that FTO suppresses m6A modification in HUVECs.

We next aimed to tell the association between FTO and DR using clinical samples. We compared FTO expression in vitreous fibrovascular membranes (FVMs) obtained from PDR patients and epi-retinal membranes (ERMs) isolated from age matched controls without diabetes. Immunofluorescence staining revealed stronger FTO intensity in FVMs compared to ERMs (Fig. 1J), further implying its critical role in DR course.

## FTO promotes endothelial cell cycle progression and tip cell formation to facilitate angiogenesis in vitro

Glucose-mediated microvascular damage is one of the first events in DR (Altmann et al, 2018). Increasing evidence indicates hyperglycemia stimulated retinal vascular EC dysfunction as the pathological basis of DR (Wang et al, 2021a). We therefore initially determined FTO's roles in regulating EC features in vitro. FTO was overexpressed in HUVECs using L-FTO and was knocked down by small interfering RNAs (siRNAs). Three pairs of siRNAs targeting the *FTO* gene (FTO-siRNA) were designed and synthesized, and FTO-siRNA #3 with the highest efficiency was selected for further assessments (Appendix Fig. S2C,D).

RNA transcriptome sequencing (RNA-Seq) was further applied to annotate aberrantly changed biological processes and signaling pathways induced by FTO overexpression in HUVECs. Gene Set Enrichment Analyses (GSEA) revealed that biological processes related to cell cycle progression are enriched upon FTO over-expression (Fig. 2A), implying the potential role of FTO in endothelial proliferation and angiogenesis. RNA-Seq detected up-regulation of genes regulating all cell cycle phases, including G1, S, G2 and M phases, in HUVECs transduced with L-FTO (Fig. 2B,C). Consistently, flow cytometric analyses confirmed that cell cycle, demonstrated by percentage of cells in S and G2/M phases, was accelerated in HUVECs transduced with L-FTO (Fig. 2D) and was suppressed in cells transfected with FTO-siRNA (Fig. 2E). 5-Ethynyl-2'-deoxyuridine (EdU) assay further confirmed promoted proliferation in HUVECs overexpressing FTO (Fig. 2F) and inhibited proliferation in cells with FTO knocked down (Fig. 2G). Furthermore, immunoblotting detected that expression of cleaved caspase-3 is down-regulated in HUVECs transduced with L-FTO (Fig. 2H) and is up-regulated in cells transfected with FTO-siRNA (Fig. 2I), indicating the suppressive role of FTO in HUVEC death.

Endothelial tip cells guide vascularization in neural retina (Zarkada et al, 2021). Elevated expression of tip cell markers, including *CXCR4*, *FSCN1*, *APLN*, *ESM1* and *PLAUR*, was detected in HUVECs overexpressing FTO (Fig. 2J), and was decreased in cells with FTO knocked down (Fig. 2K), suggesting that FTO promotes endothelial tip cell formation. Effects of FTO on endothelial migration and tube formation were subsequently investigated. As shown by Transwell migration assay and scratch test, cell migration was facilitated in

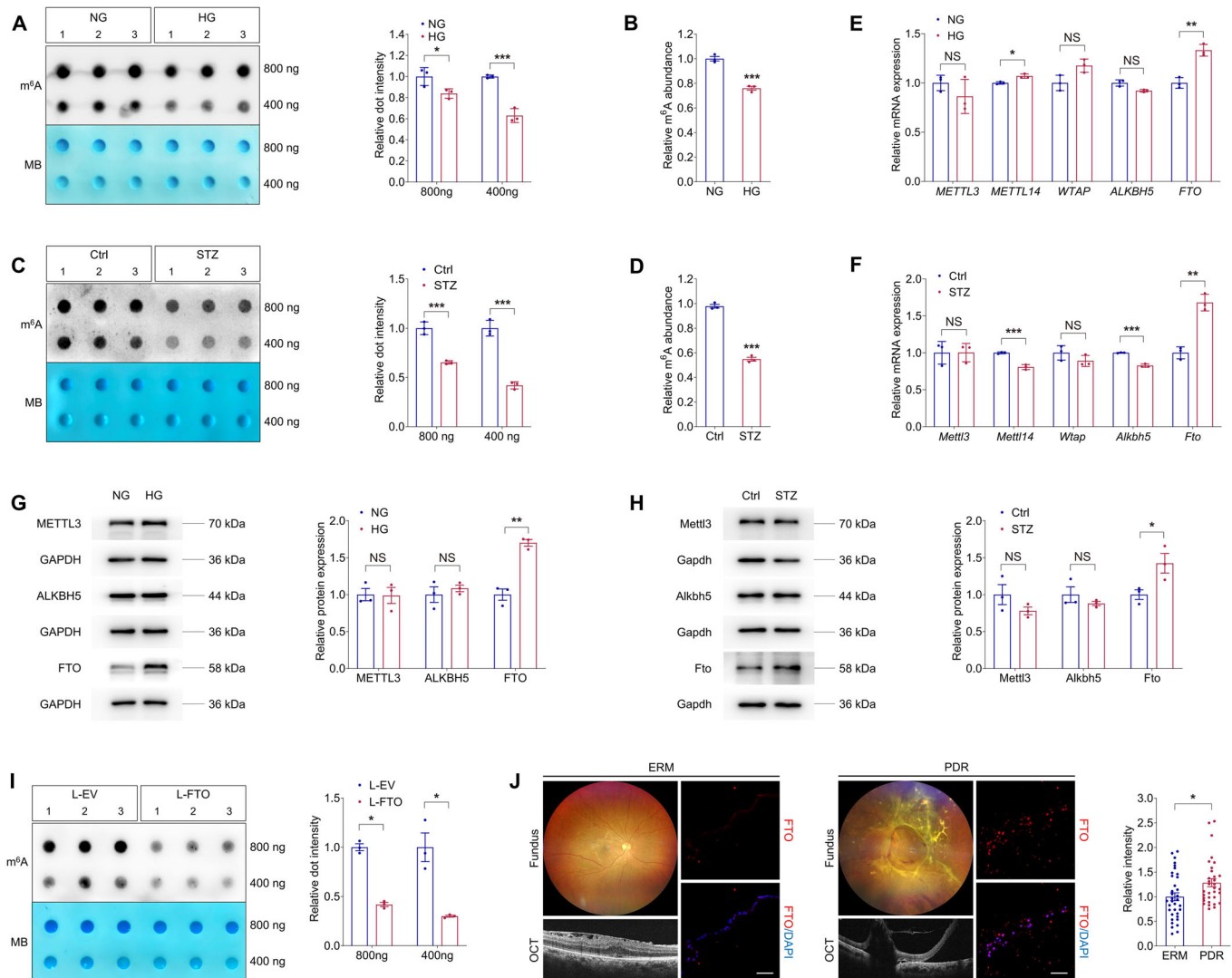

**Figure 1. Identification of FTO as a potential regulator of DR.**

(A) m⁶A dot blot assay of global m⁶A abundance in HUVECs treated with normal glucose (5 mM, NG) or high glucose (25 mM, HG) using 800 or 400 ng total RNAs. MB staining is used as a loading control. n = 3 per group. (B) m⁶A RNA methylation quantification assay of global m⁶A abundance in HUVECs treated with NG or HG. n = 3 per group. (C) m⁶A dot blot assay of global m⁶A abundance in retina originated from control or STZ mice using 800 or 400 ng total RNAs. MB staining is applied as a loading control. n = 3 per group. (D) m⁶A RNA methylation quantification assay of global m⁶A abundance in retina originated from control or STZ mice. n = 3 per group. (E) qPCR shows mRNA levels of m⁶A regulators *METTL3*, *METTL14*, *WTAP*, *ALKBH5* and *FTO* in HUVECs treated with NG or HG. n = 3 per group. (F) qPCR presents mRNA levels of m⁶A regulators *Mettl3*, *Mettl14*, *Wtap*, *Alkbh5* and *Fto* in retina originated from control or STZ mice. n = 3 per group. (G) Immunoblotting of METTL3, ALKBH5 and FTO in HUVECs treated with NG or HG. GAPDH is used as an internal control. n = 3 per group. (H) Immunoblotting of Mettl3, Alkbh5 and Fto in retina originated from control or STZ mice. GAPDH is used as an internal control. n = 3 per group. (I) m⁶A dot blot assay of global m⁶A abundance in HUVECs transduced with L-EV or L-FTO using 800 or 400 ng total RNAs. MB staining is applied as a loading control. n = 3 per group. (J) Immunofluorescence staining of FTO in FVMs obtained from PDR patients or ERMs isolated from age matched controls without diabetes. Fundus photographs and OCT images of PDR patients or age matched controls with ERM are shown. n = 35, ERM; n = 33, PDR. Scale bar: 50 μm. Data information: Data represent different numbers (n) of biological replicates. Data are shown as mean ± SEM. Two-tailed Student's t test is used. NS: not significant (p > 0.05); *p < 0.05; **p < 0.01; and ***p < 0.001. Source data are available online for this figure.

HUVECs transduced with L-FTO (Fig. 2L,N) but was restrained in cells transfected with FTO-siRNA (Fig. 2M,O). Additionally, xCELLigence real-time cell analysis (RTCA) system suggested increased migration rates in HUVECs overexpressing FTO (Fig. 2P) and decreased rates in cells with FTO knocked down (Fig. 2Q). We also revealed that tube formation, as reflected by node number and branching length, was promoted in HUVECs transduced with L-FTO (Fig. 2R) while was inhibited in cells transfected with FTO-siRNA

(Fig. 2S). Collectively, FTO promotes endothelial cell cycle progression and tip cell formation to facilitate angiogenesis in vitro.

## FTO promotes endothelial tip cell formation to facilitate neovascularization in mice and zebrafish

We next analyzed whether FTO overexpression associates with neovascularization in vivo. FTO protein sequence was highly conserved

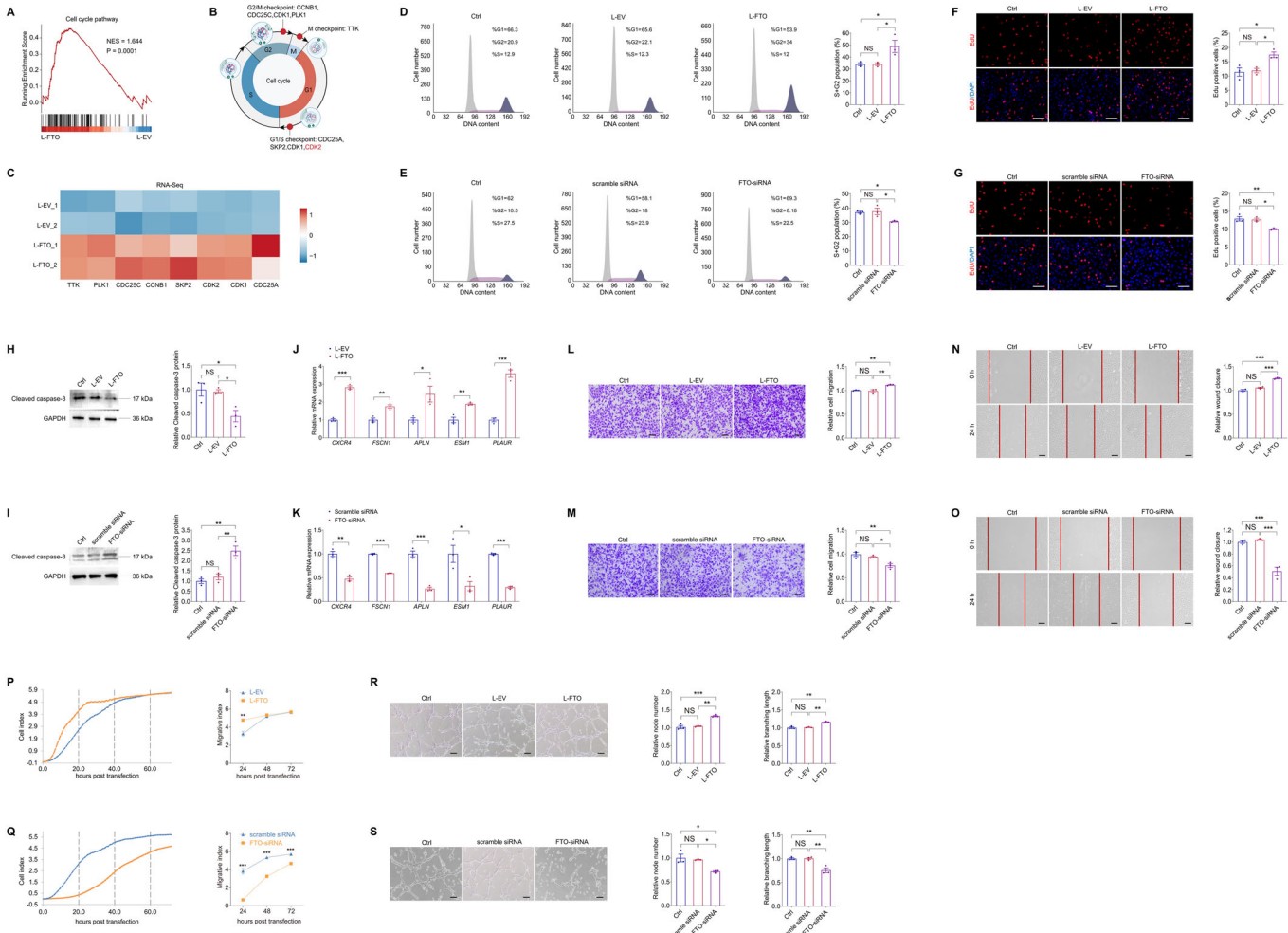

**Figure 2. FTO promotes endothelial cell cycle progression and tip cell formation to facilitate angiogenesis in vitro.**

(A) GSEA plot of pathways enriched in HUVECs transduced with L-FTO is presented. (B, C) Diagram (B) and heatmap (C) show differential expression of genes involved in cell cycle checkpoints in HUVECs transduced with L-EV or L-FTO. (D, E) Flow cytometric analyses on untreated HUVECs, HUVECs transduced with L-EV/L-FTO (D), or transfected with scramble siRNA/FTO-siRNA (E). $n = 3$ per group. (F, G) EdU assay on untreated HUVECs, HUVECs transduced with indicated lentivirus (F) or transfected with distinct siRNAs (G). Cell nuclei are counterstained with DAPI. $n = 3$ per group. Scale bar: 60 μm. (H, I) Immunoblotting of cleaved caspase-3 in untreated HUVECs, HUVECs transduced with L-EV/L-FTO (H), or scramble siRNA/FTO-siRNA (I). GAPDH is used as an internal control. $n = 3$ per group. (J, K) mRNA levels of tip cell markers (*CXCR4*, *FSCN1*, *APLN*, *ESM1* and *PLAUR*) detected by qPCR in HUVECs transduced with indicated lentivirus (J) or transfected with distinct siRNAs (K). $n = 3$ per group. (L, M) Transwell migration assay on untreated HUVECs or HUVECs transduced with indicated lentivirus (L) or transfected with distinct siRNAs (M). $n = 3$ per group. Scale bar: 50 μm. (N, O) Scratch test on untreated HUVECs or HUVECs transduced with indicated lentivirus (N) or transfected with distinct siRNAs (O). $n = 3$ per group. Scale bar: 100 μm. (P, Q) RTCA system demonstrates migration rates of HUVECs transduced with L-EV/L-FTO (P), or transfected with scramble siRNA/FTO-siRNA (Q). $n = 3$ per group. (R, S) Tube formation assay on untreated HUVECs or HUVECs transduced with L-EV/L-FTO (R), or transfected with scramble siRNA/FTO-siRNA (S). $n = 3$ per group. Scale bar: 100 μm. Data information: Data represent different numbers ($n$) of biological replicates. Data are shown as mean ± SEM. One-way ANOVA followed by Bonferroni's test is used in (D–I, L–O, R, S). Two-tailed Student's $t$ test is used in (J, K, P, Q). NS: not significant ($p > 0.05$); *$p < 0.05$; **$p < 0.01$; and ***$p < 0.001$. Source data are available online for this figure.

among human, mice and zebrafish (Appendix Fig. S3). Recombinant adeno-associated virus (AAV) containing CDS of mice *Fto* gene with Flag tag and the promoter region of mice Tie2 (AAV-Fto) was constructed and intra-vitreal injected to modulate *Fto* expression in mice vascular ECs. Immunoblotting demonstrated elevated Fto protein level in neural retinas collected from both infant (Appendix Fig. S4A,B) and adult (Appendix Fig. S4C,D) mice receiving intra-vitreal AAV-Fto injection compared to mice injected with blank AAV containing Flag tag (AAV-blank). Consistently, immunofluorescence staining showed enhanced Fto expression in the isolated retinal vasculatures from adult mice injected with AAV-Fto (Appendix Fig. S4E). Expression of Fto-

Flag fusion protein was also detected in AAV-FTO injected adult mice retina (Appendix Fig. S4F), and was localized along with the retinal vasculatures (Appendix Fig. S4G).

To annotate the role of FTO in regulating mice retinal angiogenesis, we intra-vitreal injected AAV-Fto into mice with oxygen-induced retinopathy (OIR) at P12, the beginning time point of retinal neovascularization and vascular leakage (Connor et al, 2009). Retina was collected and examined at P17 in OIR, before the regression of pathological vessels (Fig. 3A). More extensive areas of neovascular tufts (NVTs), formed in the superficial vascular plexuses, were observed in the OIR mice receiving AAV-Fto

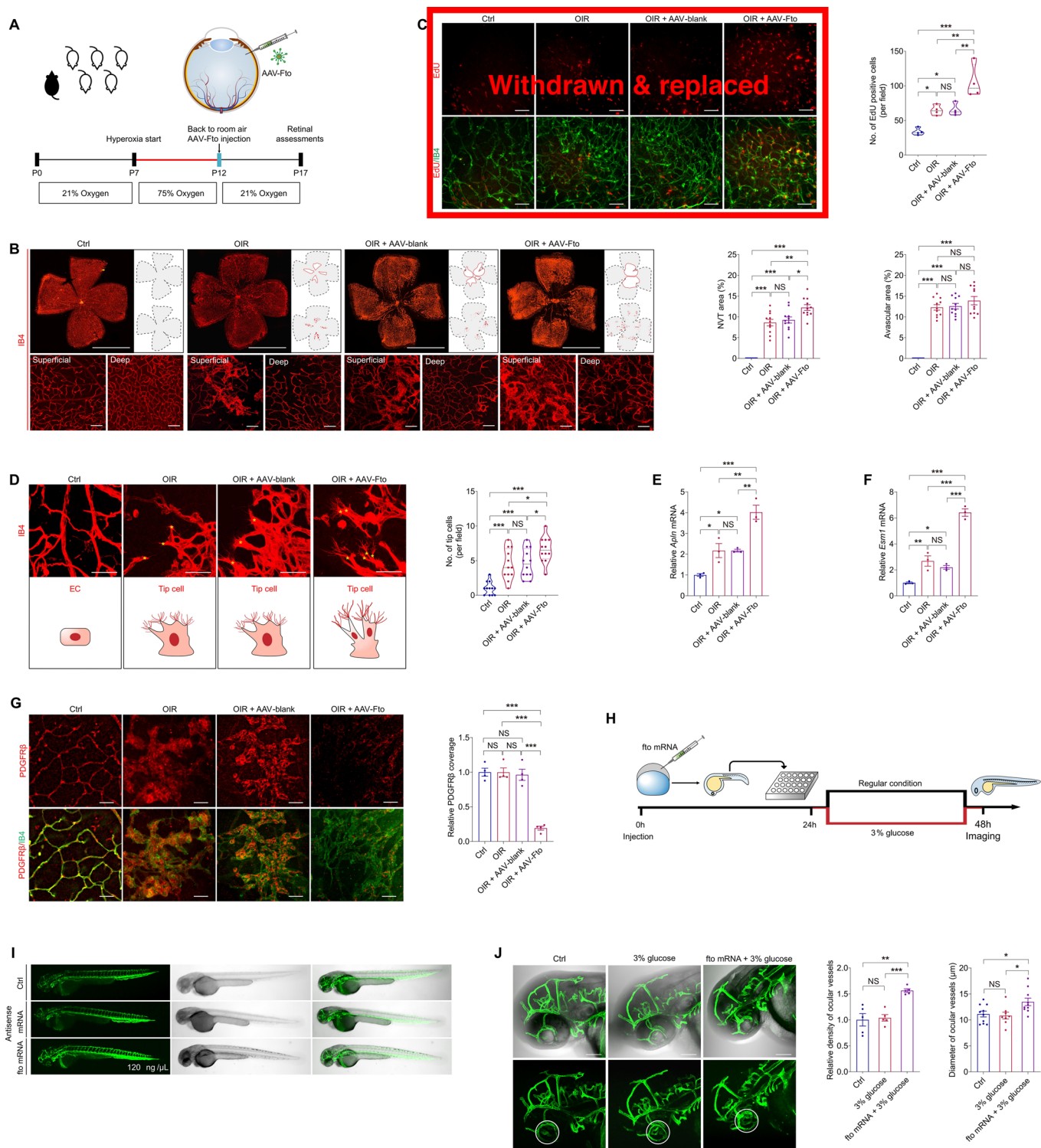

injection compared to the OIR mice without injection or the OIR mice injected with AAV-blank (Fig. 3B). In vivo EdU assay also revealed enhanced proliferation of vascular cells in neural retinas collected from the OIR mice receiving AAV-Fto injection (Fig. 3C). Moreover, we noticed that, in the angiogenic area, *Fto* over-expression leads to increased amount of endothelial tip cells (Fig. 3D). Consistently, qPCR demonstrated increased expression

of tip cell markers *Apln* and *Esm1* in neural retinas collected from the OIR mice injected with AAV-Fto (Fig. 3E,F). However, no difference in avascular area in the central retina and vessel density in the peripheral deep plexuses was detected among the OIR mice without injection, injected with AAV-Fto and injected with AAV-blank (Fig. 3B). No NVT or avascular area was observed in the control group (Fig. 3B). Pericyte loss in retinal capillaries can

**Figure 3.   FTO promotes endothelial tip cell formation to facilitate neovascularization in mice and zebrafish.**

(A) Experimental scheme for (B–G) (B) Fluorescent staining of IB4 in retinal flat mounts originated from control, OIR mice without other treatment, and OIR mice intra-vitreal injected with AAV-blank/AAV-Fto at P17. $n = 11$ per group. Magnificent images are shown below to better visualize superficial and deep vascular plexuses. Gray dotted lines indicate the edge of the retina. Red lines suggest the avascular area. NVTs are represented by red dots. Representative images along with quantification results of NVT and avascular areas are shown. Scale bar: 2000 μm (up); 50 μm (below). (C) Fluorescent staining of EdU and IB4 in retinal flat mounts of control, OIR mice without other treatment, and OIR mice intra-vitreal injected with AAV-blank/AAV-Fto at P17. $n = 4$ per group. (D) Fluorescent staining of IB4 in retinal flat mounts of control, OIR mice without other treatment, and OIR mice intra-vitreal injected with AAV-blank/AAV-Fto at P17. $n = 12$ per group. Tip cells are indicated by yellow asterisks. Representative images along with quantification results of tip cell number per field are shown. Scale bar: 50 μm. (E, F) mRNA expression of tip cell markers *Apln* (E) and *Esm1* (F) detected by qPCR in neural retinas of control, OIR mice without other treatment, and OIR mice intra-vitreal injected with AAV-blank/AAV-Fto at P17. $n = 3$ per group. (G) Immunofluorescence staining of PDGFRβ and IB4 in retinal flat mounts of control, OIR mice without other treatment, and OIR mice intra-vitreal injected with distinct virus at P17. $n = 4$ per group. (H) Experimental scheme for (I, J). (I) Truncal vasculatures demonstrated by endogenous EGFP and morphological structures of control and zebrafish injected with antisense or *fto* mRNA. $n = 3$ per group. (J) Ocular vasculatures shown by endogenous EGFP and morphological structures of control and zebrafish receiving indicated treatments. Ocular vasculatures are indicated by white circles. Representative images along with quantification results of ocular vessel density ($n = 5$ per group) and diameters ($n = 8$–10 per group) are shown. Scale bar: 100 μm. Data information: Data represent different numbers ($n$) of biological replicates. Data are shown as mean ± SEM. One-way ANOVA followed by Bonferroni's test is used. NS: not significant ($p > 0.05$); *$p < 0.05$; **$p < 0.01$; and ***$p < 0.001$. Source data are available online for this figure.

induce BRB destruction (Cai et al, 2002; Jiang et al, 2020). We next used immunofluorescence staining of platelet derived growth factor receptor β (PDGFRβ), a pericyte marker, to annotate FTO's function in regulating EC-pericyte crosstalk. Reduced pericyte coverage was detected in NVTs of OIR mice overexpressing Fto (Fig. 3G), indicating FTO's role in interrupting EC-pericyte crosstalk. Collectively, our data suggested that FTO facilitated pathological NVTs development and promoted endothelial tip cell formation in the diseased vessel remodeling period of OIR.

We further analyzed FTO's effects on mediating vascular functions in zebrafish. We overexpressed FTO in zebrafish through embryonic injection of zebrafish *fto* mRNA in the living transgenic zebrafish strain *Tg(LR57:GFP)*, which contains the enhanced green fluorescent protein (EGFP) cDNA under control of the *fli1* promoter, thus allowing us to visualize its systemic vessels, including ocular vasculatures. Zebrafish embryos injected with *fto* mRNA at the concentration of 120 ng/μL showed no remarkable systemic changes, and were collected and examined at 48 h post fertilization (Fig. 3H,I). To investigate the role of fto under diabetic condition, fish were cultured in water added with 3% glucose. Fluorescence staining detected enhanced density as well as dilation of ocular vasculatures in zebrafish injected with *fto* mRNA and maintained in 3% glucose water compared to the uninjected zebrafish in 3% glucose water and the control group (Fig. 3J). These data indicated that FTO promotes zebrafish ocular vascularization under diabetic conditions. Collectively, we found that FTO regulates endothelial tip cell formation to promote neovascularization in mice and zebrafish.

## FTO regulates EC-pericyte crosstalk and triggers diabetes-induced microvascular dysfunction in mice

To further annotate the role of FTO in regulating diabetes-induced microvascular dysfunction, we assessed retinal vasculatures in STZ mice receiving twice intra-vitreal injections of AAV-Fto (Fig. 4A). The STZ mice develop retinal vascular leakage without neovascularization (Feit-Leichman et al, 2005). Fundus photograph identified cotton wool spot-like lesions in STZ mice injected with AAV-Fto (Fig. 4B). Fluorescence fundus angiography (FFA), Texas red dextran and Evans blue assays were conducted to further explore FTO's role in retinal vascular leakage. All three experiments consistently revealed that FTO overexpression aggravated diabetes-induced retinal vascular leakage (Fig. 4C–E).

Vascular leakage in the diabetic retina is usually caused by breakdown of the BRB, a biological unit comprised of capillary ECs firmly connected by intercellular tight junctions and their surrounding cells (Nishikiori et al, 2007). Thus, we stained retinal flat mounts with vascular-endothelial-specific cadherin (VE-cadherin), which maintains the integrity of the EC barrier and attenuates VEGF signaling to suppress angiogenesis (Lampugnani et al, 2006; Sidibe et al, 2014). VE-cadherin in diabetic retinas was discontinuous compared to non-diabetic animals (Fig. 4F). Both intensity and coverage area of VE-cadherin were remarkably reduced in STZ mice injected with AAV-Fto compared to those without injection or injected with AAV-blank (Fig. 4F). Consistently, immunoblotting also revealed that expression of ZO-1, which forms a confluent tight junction at the vascular endothelial cell membrane, is reduced in neural retinas collected from STZ mice injected with AAV-Fto compared to the other groups (Fig. 4G). We further used NG2/PDGFRβ (pericyte markers) and IB4 immunofluorescence staining to detect pericyte coverage of retinal vessels. Consistently, both NG2 and PDGFRβ staining demonstrated reduced pericyte coverage of retinal vessels upon the combination of FTO overexpression and diabetes (Fig. 4H,I).

Vascular lesions, such as acellular capillaries and intraretinal microvascular abnormalities (IRMA), are typical pathological features of diabetic retinas (Jiang et al, 2020). We used trypsin digestion and periodic acid Schiff (PAS) staining to detect FTO associated structural changes in retinal vessels. Increased number of acellular capillaries was detected in STZ mice injected with AAV-Fto compared to the control group, STZ mice without injection, and STZ mice injected with AAV-blank (Fig. 4J). Additionally, fluorescence staining of IB4 in retinal flat mounts demonstrated enlarged areas of IRMA in STZ mice receiving AAV-Fto injection compared to STZ mice without injection or injected with AAV-blank (Fig. 4K). Uneven and tortuous retinal vasculatures were also observed upon the combination of FTO overexpression and diabetes in mice (Fig. 4K). Collectively, above data suggested that FTO overexpression and diabetes regulated EC-pericyte crosstalk and aggravated microvascular pathology in a synergistic manner.

## FTO triggers vascular inflammation and regulates EC-microglia crosstalk in vitro

Growing evidence revealed the critical role of retinal inflammation in impaired endothelial function, vascular leakage, pericyte loss and

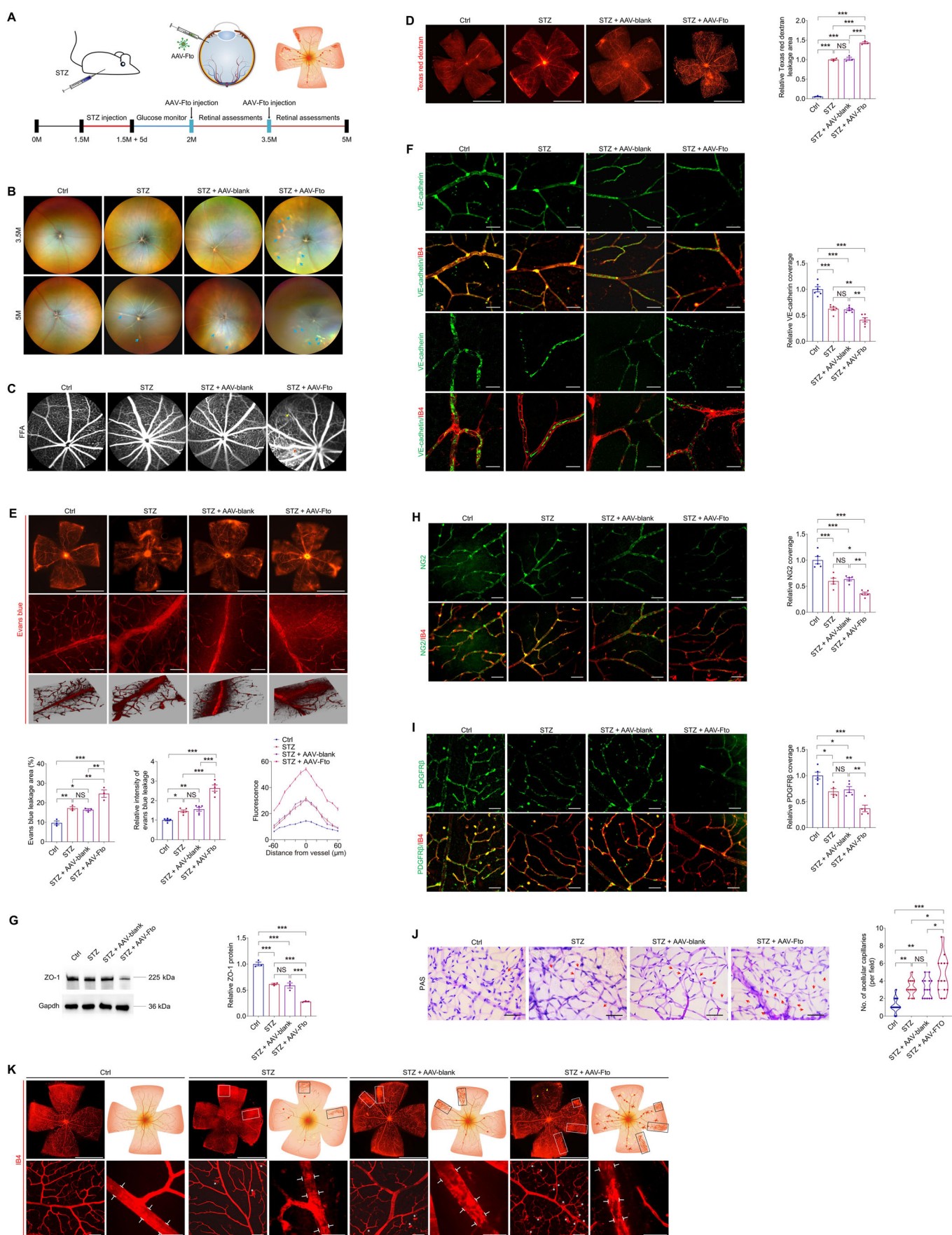

**Figure 4.  FTO regulates EC-pericyte crosstalk to trigger diabetes-induced microvascular dysfunction in mice.**

(A) Experimental scheme for (B–K). (B, C) Representative images of fundus photos (B) and FFA (C) of control, STZ mice without injection, and STZ mice intra-vitreal injected with AAV-blank/AAV-Fto at indicated time points after injection. $n = 3$ per group. Blue arrows represent cotton wool spot-like lesions. Red arrow indicates leakage spot. Yellow arrow suggests abnormal vascular perfusion. (D, E) Fluorescence of Texas red dextran (D) and Evans blue (E) are visualized in retinal flat mounts of control, STZ mice without injection, and STZ mice intra-vitreal injected with AAV-blank/AAV-Fto. $n = 3$ per group. Representative images along with quantification results of leakage area and leaky flourescence intensity are shown. Scale bar: 2000 μm ((D) and up in (E)); 50 μm (below in (E)). (F) Immunofluorescence staining of VE-cadherin and IB4 in retinal flat mounts of control, STZ mice without injection, and STZ mice intra-vitreal injected with AAV-blank/AAV-Fto. $n = 6$ per group. Scale bar: 50 μm (upper); 20 μm (below). (G) Immunoblotting of ZO-1 in neural retinas collected from control, STZ mice without injection, and STZ mice intra-vitreal injected with AAV-blank/AAV-Fto. Gapdh is used as internal control. $n = 3$ per group. (H) Immunofluorescence staining of NG2 and IB4 in retinal flat mounts of control, STZ mice without injection, and STZ mice intra-vitreal injected with AAV-blank/AAV-Fto. $n = 5$ per group. Yellow arrowheads indicate pericytes. Scale bar: 50 μm. (I) Immunofluorescence staining of PDGFRβ and IB4 in retinal flat mounts of control, STZ mice without injection, and STZ mice intra-vitreal injected with AAV-blank/AAV-Fto. $n = 5$ per group. Scale bar: 50 μm. (J) PAS staining of trypsin digested retinal vessels isolated from control, STZ mice without injection, and STZ mice intra-vitreal injected with AAV-blank/AAV-Fto. $n = 14$ per group. Scale bar: 50 μm. (K) Fluorescence staining of IB4 in retinal flat mounts of control, STZ mice without injection, and STZ mice intra-vitreal injected with AAV-blank/AAV-Fto. $n = 3$ per group. IRMAs are represented by white and black rectangles. Capillary dropout regions are suggested by yellow arrowheads. White brackets indicate structures of main vessels, and white asterisks represent activated microglia cells wrapping around retinal vessels. Scale bar: 2000 μm (up); 50 μm (below). Data information: Data represent different numbers ($n$) of biological replicates. Data are shown as mean ± SEM. One-way ANOVA followed by Bonferroni's test is used. NS: not significant ($p > 0.05$); *$p < 0.05$; **$p < 0.01$; and ***$p < 0.001$. Source data are available online for this figure.

retinal neovascularization (Semeraro et al, 2019), we therefore tested whether FTO associates with vascular inflammation. VEGF-A is a proangiogenic and proinflammatory mediator in DR (Uemura et al, 2021). Inhibition of FTO suppressed VEGF-A release in macrophages and retinal pigment epithelial (RPE) cells, which restrained angiogenesis and macrophage infiltration in choroidal neovascularization (Wang et al, 2023a). Consistent with previous findings, enzyme-linked immunosorbent assay (ELISA) identified promoted VEGF-A secretion in HUVECs overexpressing FTO, implying the potential role of FTO in facilitating angiogenesis and inflammation (Fig. 5A). Additionally, both RNA-Seq and tandem mass tag (TMT)-based quantitative proteomic analyses detected up-regulation of pro-inflammatory genes/proteins and down-regulation of anti-inflammatory genes/proteins in HUVECs overexpressing FTO (Fig. 5B,C). Liquid protein chip (LiquiChip), also known as flexible multi-analyte profiling technology, further identified increased levels of pro-inflammatory chemokines (CSF, IL-18 and RANTES) and decreased level of anti-inflammatory cytokines (LIF, IL-4, IL-3 and IL-10) in the culture medium of HUVECs overexpressing FTO (Fig. 5D). Above data suggested that FTO overexpression in EC triggers vascular inflammation.

Vascular inflammation and subsequent microglia activation are typical features of DR and are critical in DR progression (Tang et al, 2023; Zeng et al, 2008). To further annotate the role of FTO in regulating EC-microglia crosstalk, we co-cultivated the human microglial clone 3 (HMC3) cells with HUVECs transduced with L-FTO (Fig. 5E). Immunofluorescence staining of ionized calcium binding adapter molecule 1 (Iba-1), a calcium-binding protein that participates in membrane ruffling and phagocytosis of activated microglia, revealed that the percentage of activated microglia, represented by an ameboid morphology, was increased upon co-cultivation with HUVECs overexpressing FTO (Fig. 5F). Immuno-fluorescence staining also identified decreased intensity of transmembrane protein 119 (TMEM119), which specifically labels resident and resting microglia (Bennett et al, 2016; Liu et al, 2022), after co-culture with HUVECs transduced with L-FTO (Fig. 5G).

Activation of microglia cells additionally enhanced their migration and proliferation (Altmann and Schmidt, 2018; Zeng et al, 2008). RNA expression of *ITGB1*, which is responsible for the recruitment and migration of microglia (Kim et al, 2014), was increased in HMC3 cells co-cultured with HUVECs overexpressing FTO as revealed by qPCR (Fig. 5H). Transwell migration assay also identified facilitated migration of HMC3 cells sharing medium with

HUVECs overexpressing FTO (Fig. 5I). Proliferation of HMC3 cells co-cultured with HUVECs transduced with L-FTO was also accelerated as indicated by EdU assay (Fig. 5J). Collectively, our data suggested that FTO overexpression in EC triggers vascular inflammation and regulates EC-microglia crosstalk to promote microglia activation, migration and proliferation.

## FTO triggers microglia activation and retinal neurodegeneration in mice

Consistent with the above in vitro findings, fluorescence staining of IB4 in retinal flat mounts demonstrated accumulation of microglia surrounding the retinal vasculature in STZ mice overexpression FTO (Fig. 4I). We further asked whether FTO associates with retinal inflammatory response in vivo. As revealed by qPCR assay, Fto overexpression aggravated retinal inflammation by elevating expression of pro-inflammatory genes *Il1b* (Rangaraju et al, 2018) and *Ccl2* (Scholz et al, 2015) (Fig. 6A). We further annotated the role of FTO in regulating microglia features in mice. As revealed by immunofluorescence staining, accumulation of Iba-1 positive microglia was noticed in STZ mice intra-vitreal injected with AAV-Fto (Fig. 6B,C). Further morphological assessments identified that endothelial FTO overexpression and diabetes promote microglia activation, represented by an ameboid morphology, in a synergistic manner (Fig. 6B,D). Endothelial FTO overexpression and diabetes also synergistically increased the soma area as well as decreased the territory projection area and number of interactions of microglia (Fig. 6B,E–H). Consistently, decreased mRNA expression of *Tmem119* (resting microglia) and increased mRNA levels of *Trem2*, *Lgals3* and *Cd11c* (activated microglia) were detected in neural retinas of STZ mice intra-vitreal injected with AAV-Fto, indicating that endothelial FTO overexpression induces microglia activation (Fig. 6I).

Activation of microglia severely affected retinal neurons, leading to neurodegeneration and vision loss (Lampugnani et al, 2006). We thus tested whether FTO overexpression was accompanied with RGC loss using the RGC specific marker tubulin beta-III (TUBB3) (Tonari et al, 2012). We found that diabetes and FTO over-expression decreased mRNA level of *Tubb3* in mice neural retina in a synergistic manner (Fig. 6J). Axonal degeneration and reduced number of Tubb3 positive RGCs were also noticed in neural retinas of STZ mice injected with AAV-Fto (Fig. 6K). Thus, our data

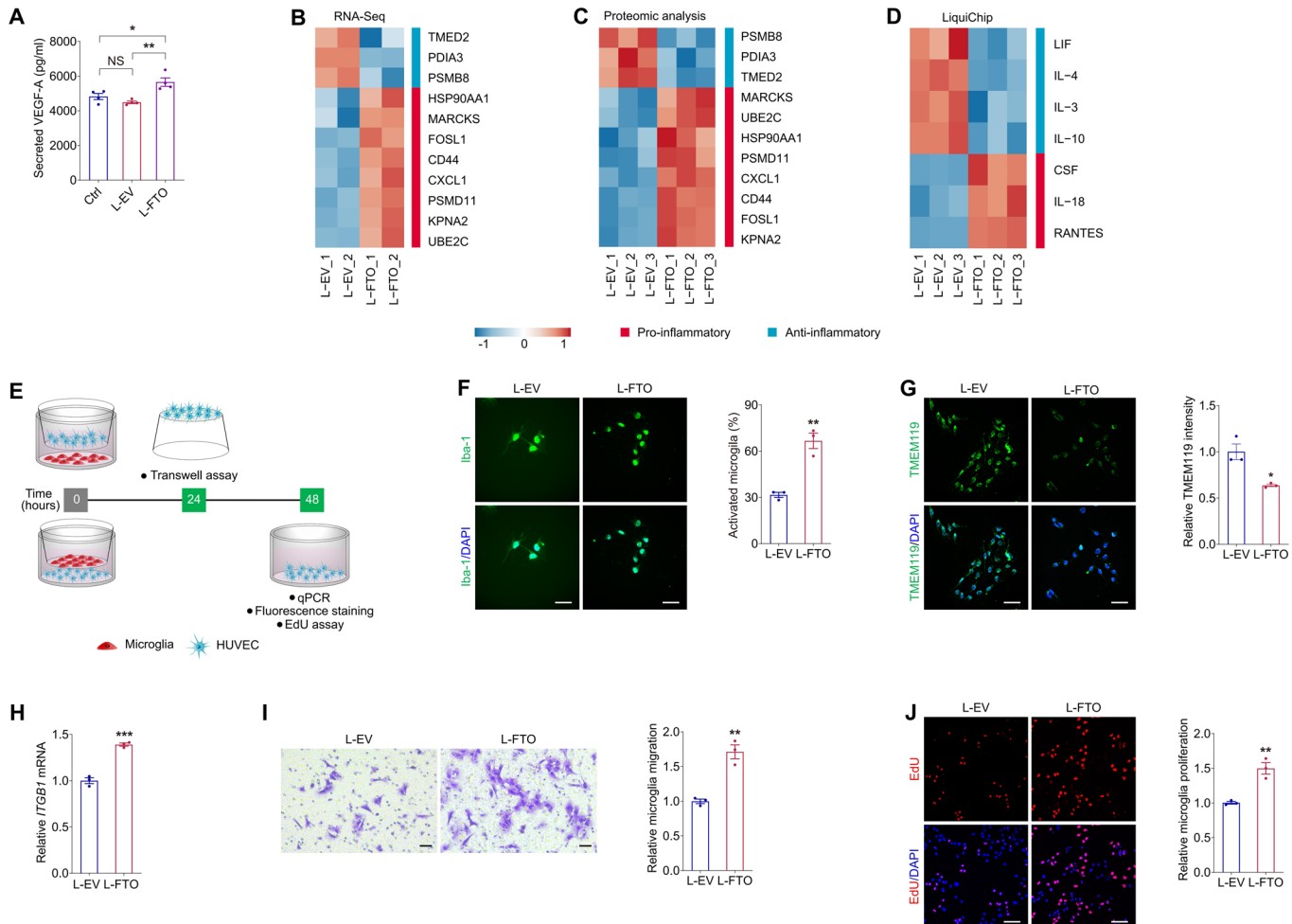

**Figure 5. FTO triggers vascular inflammation and regulates EC–microglia crosstalk in vitro.**

(A) VEGF-A secretion in HUVECs detected by ELISA in the control group and HUVECs transduced with L-EV or L-FTO. $n = 4$ per group. (B, C) Hierarchical clustering of pro- and anti-inflammatory genes detected by RNA-Seq (B) and proteins revealed by TMT-based quantitative proteomic analyses (C) in HUVECs transduced with L-EV or L-FTO. (D) Heatmap of pro- and anti-inflammatory cytokines in the culture medium of HUVECs transduced with L-EV or L-FTO as detected by LiquiChip. (E) Experimental scheme for (F–J). (F, G) Immunofluorescence staining of Iba-1 (F) and TMEM119 (G) in microglia co-cultured with HUVECs transduced with L-EV or L-FTO. Cell nuclei are counterstained with DAPI. $n = 3$ per group. Scale bar: 65 μm. (H). mRNA level of *ITGB1* detected by qPCR in microglia co-cultured with HUVECs transduced with L-EV or L-FTO. $n = 3$ per group. (I). Transwell migration assay on microglia co-cultured with HUVECs transduced with L-EV or L-FTO. $n = 3$ per group. Scale bar: 50 μm. (J). EdU assay on microglia co-cultured with HUVECs transduced with L-EV or L-FTO. $n = 3$ per group. Scale bar: 60 μm. Data information: Data represent different numbers ($n$) of biological replicates. Data are shown as mean ± SEM. One-way ANOVA followed by Bonferroni's test is used in (A). Two-tailed Student's $t$ test is used in (F–J). NS: not significant ($p > 0.05$); **$p < 0.01$; and ***$p < 0.001$. Source data are available online for this figure.

indicated that FTO associates with retinal inflammation, microglia activation and neurodegeneration in mice.

## Demethylation activity is required for FTO to regulate EC features

To annotate whether effects of FTO on EC function depend on its demethylation activity, we introduced two catalytically inactive mutations, H231A and D233A (Huang et al, 2020; Jia et al, 2011), into L-FTO to generate L-FTO^MU. Dot blot assay indicated that the two mutations remarkably inhibited demethylation activity of FTO (Fig. 7A), but immunoblotting suggested that FTO protein expression was not affected (Fig. 7B). Flow cytometric analyses demonstrated accelerated cell cycle process in HUVECs

overexpressing wild type FTO protein, but was not affected in cells overexpressing mutant FTO compared to the control group (Fig. 7C), supporting that the mutations reverse the promotive role of FTO on endothelial cell cycle progression. Facilitated proliferation, indicated by EdU positive cells, was identified in HUVECs transduced with L-FTO^WT, but was not found in cells infected with L-FTO^MU (Fig. 7D). Moreover, cell migration, demonstrated by scratch test and Transwell migration assay, was facilitated in HUVECs transduced with L-FTO^WT but not the mutant form (Fig. 7E,F), suggesting that mutant FTO protein does not associate with EC migration. Tube formation assay also revealed increased node number and branching length in HUVECs transduced with L-FTO^WT but not L-FTO^MU (Fig. 7G). Collectively, the effects of FTO on EC will be abolished by

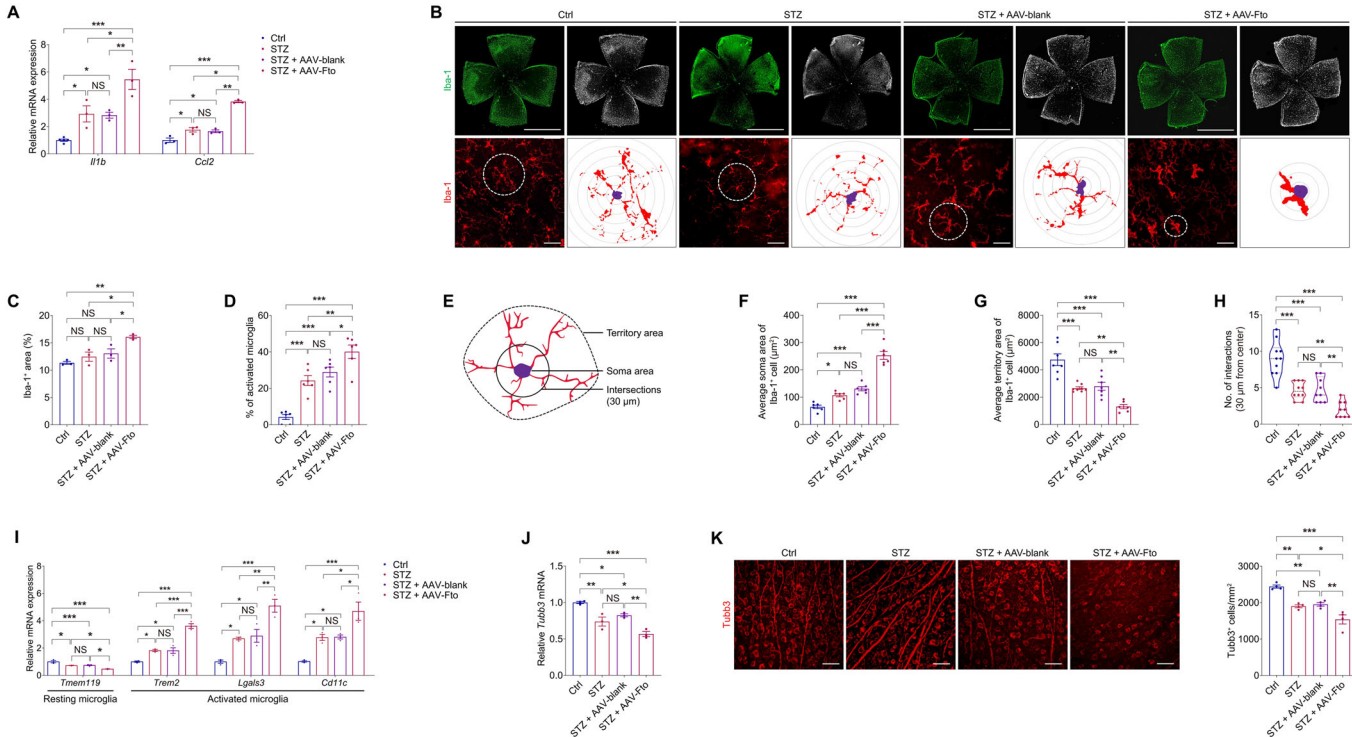

**Figure 6. FTO triggers microglia activation and neurodegeneration in mice.**

(A) qPCR shows mRNA levels of inflammatory factors *Il1b* and *Ccl2* in neural retinas of control, STZ mice without injection, and STZ mice intra-vitreal injected with AAV-blank/AAV-Fto. $n = 3$-5 per group. (B–D) Fluorescence staining of Iba-1 in retinal flat mounts of mice receiving indicated treatments is demonstrated. $n = 3$-6 per group. Representative images (B) along with quantification of Iba-1 positive microglia area (C) and activated microglia (D) are shown. Scale bar: 2000 μm (up); 50 μm (below). (E) The inset depicts the parameters regarding morphology of microglia. (F–H) Quantification results of soma area (F), territory area (G) and intersection numbers at 30 μm from nuclei (H) in retinal microglia from mice receiving indicated treatments. $n = 6$-10 per group. (I, J) qPCR demonstrates mRNA levels of *Tmem119, Trem2, Lgals3, Cd11c* (I) and *Tubb3* (J) in neural retinas of control, STZ mice without injection, and STZ mice intra-vitreal injected with AAV-blank/AAV-Fto. $n = 3$ per group. (K) Fluorescence staining of Tubb3 in retinal flat mounts of mice receiving indicated treatments is demonstrated. Tubb3[+] cells, recognized by the round RGC bodies, were counted in three random areas within the radius of 0.5 to 1.5 mm from the optic disc, and averaged to estimate the RGC per mm$^2$ in four retinal flat mounts per group. $n = 4$ per group. Scale bar: 50 μm. Data information: Data represent different numbers ($n$) of biological replicates. Data are shown as mean ± SEM. One-way ANOVA followed by Bonferroni's test is used. NS: not significant ($p > 0.05$); *$p < 0.05$; **$p < 0.01$; and ***$p < 0.001$. Source data are available online for this figure.

introduction of the H231A and D233A mutations. Intact demethylation activity of FTO was required for its regulations on EC features and its pathogenic roles in microvascular dysfunctions.

## FTO regulates CDK2 mRNA stability with YTHDF2 as the reader in an m⁶A-dependent manner

Since FTO mediates EC features through its demethylation activity, we then applied methylated RNA immunoprecipitation-sequencing (MeRIP-Seq) to identify downstream targets of FTO in HUVECs. A total of 465 m⁶A-hypo peaks and 586 m⁶A-hyper peaks (Log₂ FC > 0.5 or <−0.5; $p < 0.05$) were initially identified in HUVECs overexpressing FTO compared to cells transduced with L-EV. MeRIP-Seq revealed a dominant distribution of m⁶A peaks in mRNAs (Fig. 8A), especially in CDS and 3'-untranslated region (3'-UTR) of RNA transcripts (Fig. 8B). Consistent with previous reports (Dominissini et al, 2012; Meyer et al, 2012), the m⁶A sites displayed the RRACH motif (Fig. 8C). Given the demethylation activity of FTO, we focused on genes with m⁶A-hypo peaks in HUVECs upon FTO overexpression. We analyzed whether protein

expression of these genes was altered using the proteomic data. A total of 9 genes containing m⁶A-hypo peaks with altered protein expression were sorted out (Fig. 8D). Among all, the *CDK2* gene, encoding a serine/threonine protein kinase that participates in cell cycle regulation, was found involved in cell proliferation and DR (Jiang et al, 2020; Lolli et al, 2005). However, the pathological involvement of all the other 8 genes in DR or retinal neovascularization has never been revealed. We thus select CDK2 as a potential downstream target of FTO in HUVECs for further investigations. MeRIP-Seq data showed that FTO overexpression suppresses the m⁶A level of a peak containing an m⁶A site (Chr1: 56365520-56365521) in the *CDK2* transcript in HUVECs (Fig. 8E). We also detected elevated *CDK2* mRNA and protein expression in HUVECs exposed to high glucose (Fig. 8F,G) and neural retinas of STZ mice (Fig. 8H,I). Thus, we emphasized on CDK2 as a potential key regulator of FTO in subsequent studies.

We next investigated whether CDK2 was a direct target of FTO in HUVECs. MeRIP-qPCR confirmed the direct binding between m⁶A antibody and the m⁶A site within the *CDK2* transcript in HUVECs (Fig. 8J). RNA immunoprecipitation (RIP) assay further validated the binding between the FTO protein and the *CDK2*

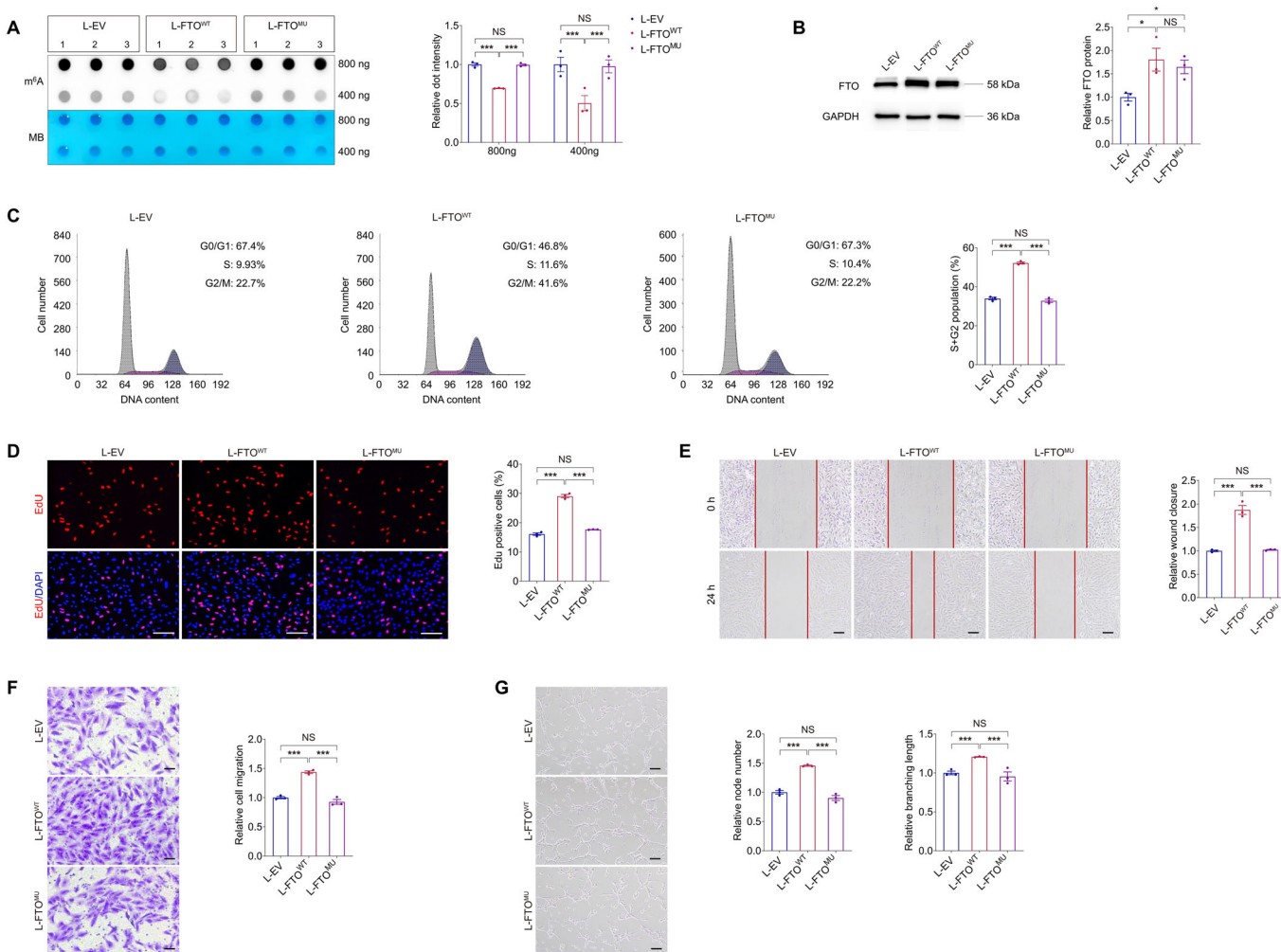

**Figure 7. Demethylation activity is required for FTO to regulate EC features.**

(A) m6A dot blot assay of global m6A abundance in HUVECs transduced with L-EV, L-FTO^WT or L-FTO^MU using 800 or 400 ng total RNAs. MB staining is used as a loading control. $n = 3$ per group. (B) Immunoblotting of FTO in HUVECs transduced with L-EV, L-FTO^WT or L-FTO^MU. GAPDH is used as an internal control. $n = 3$ per group. (C) Flow cytometric analyses of cell cycle process in HUVECs transduced with L-EV, L-FTO^WT or L-FTO^MU. $n = 3$ per group. (D) EdU assay on HUVECs transduced with indicated lentivirus. $n = 3$ per group. Scale bar: 60 μm. (E) Scratch test on HUVECs transduced with L-EV, L-FTO^WT or L-FTO^MU. $n = 3$ per group. Scale bar: 100 μm. (F) Transwell migration assay on HUVECs transduced with indicated lentivirus. $n = 3$ per group. Scale bar: 50 μm. (G) Tube formation analyses on HUVECs transduced with L-EV, L-FTO^WT or L-FTO^MU. $n = 3$ per group. Scale bar: 100 μm. Data information: Data represent different numbers ($n$) of biological replicates. Data are shown as mean ± SEM. One-way ANOVA followed by Bonferroni's test is used. NS: not significant ($p > 0.05$); *$p < 0.05$; and ***$p < 0.001$. Source data are available online for this figure.

mRNA, while the binding was partly abandoned upon over-expression of the mutant protein (Fig. 8K). In line with the MeRIP-Seq data (Fig. 8D,E), MeRIP-qPCR identified suppressed m6A level in the *CDK2* transcript in HUVECs overexpressing wild type FTO, but not its mutant form (Fig. 8L), suggesting the inhibitory role of FTO on m6A contents in CDK2. As revealed by qPCR and immunoblotting, both mRNA and protein expression of CDK2 were increased in HUVECs upon transduction of L-FTO^WT, while such change was abolished in cells transduced with L-FTO^MU (Fig. 8M,N). Consistently, CDK2 mRNA and protein levels were decreased in HUVECs transfected with FTO-siRNA (Fig. 8O–P). Collectively, our data implied that FTO regulates CDK2 expression through its demethylation on the *CDK2* transcript.

We further annotated the specific m6A reader that binds to CDK2 in EC. Above data suggested that FTO decreased m6A level

but increased mRNA expression of CDK2, therefore YTHDF2, a well-recognized m6A reader that promotes targeted mRNA decay (Wang et al, 2014; Wang et al, 2015b), is identified as a potential binding protein to CDK2 in EC. We next investigated whether FTO promotes CDK2 expression by enhancing its mRNA stability with YTHDF2 as a reader. Actinomycin D was used to inhibit the process of RNA transcription in HUVECs. Prolonged half-life of the *CDK2* transcript was detected in cells transduced with L-FTO^WT, but not in cells overexpressing the mutant FTO protein (Fig. 8Q), supporting that FTO intensifies *CDK2* mRNA stability via its demethylation activity. RIP assay further confirmed the direct binding between the YTHDF2 protein and the *CDK2* transcript (Fig. 8R). We then analyzed whether YTHDF2 knocking down alleviates its mediated decay of the *CDK2* mRNA. Among all three pairs of siRNAs targeting the *YTHDF2* gene, YTHDF2-siRNA

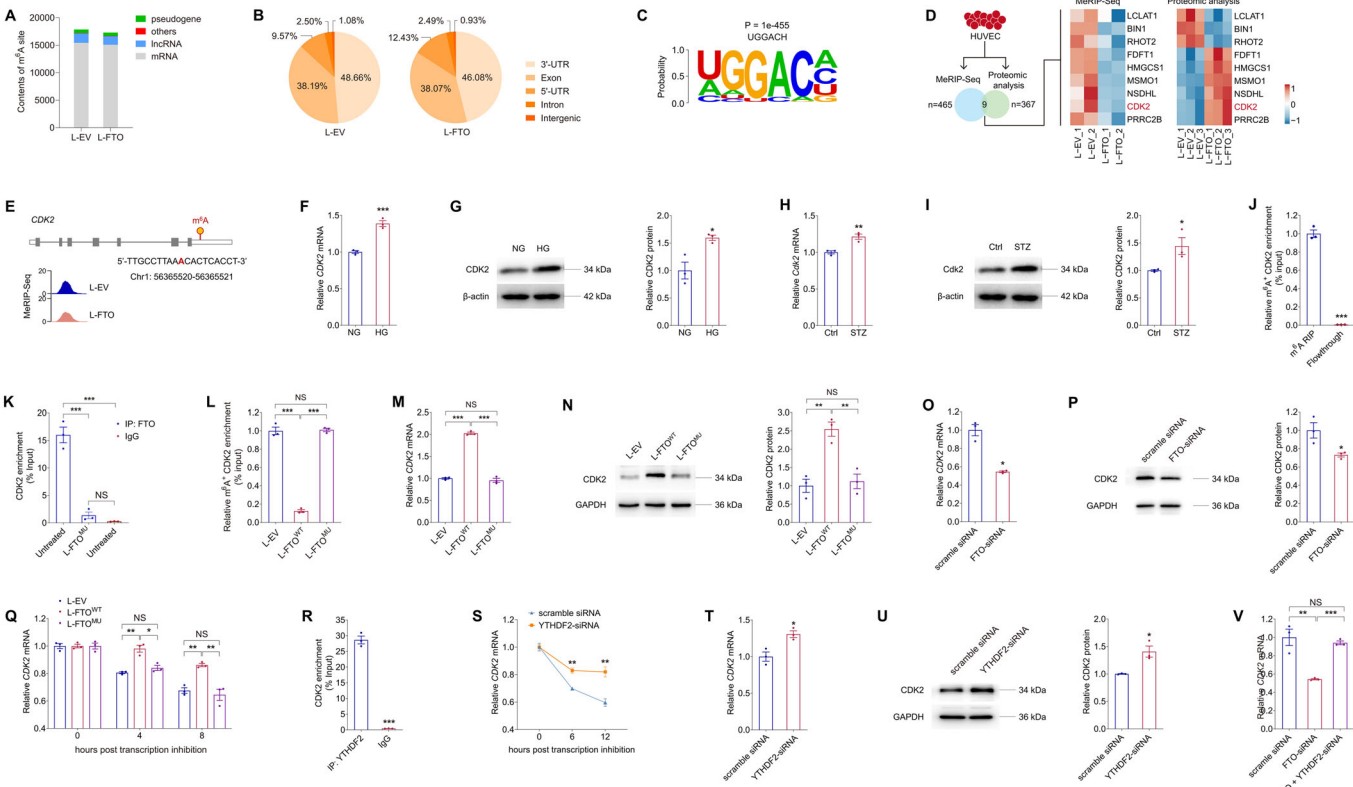

**Figure 8. FTO regulates CDK2 mRNA stability with YTHDF2 as the reader in an m⁶A-dependent manner.**

(A, B) MeRIP-Seq data of HUVECs transduced with L-EV or L-FTO. Contents of m⁶A site in all types of RNAs (A) and distribution of total m⁶A peaks in distinct regions of mRNA transcripts (B) are presented. (C). Sequence of enriched motif displayed by HOMER. (D) Genes containing m⁶A-hypo peaks with altered protein expression are sorted out. (E) Abundance of m⁶A peak in the *CDK2* transcript in HUVECs transduced with L-EV or L-FTO detected by MeRIP-Seq. Genomic location of its containing m⁶A site is annotated. (F, G) mRNA (F) and protein (G) expression of CDK2 in HUVECs exposed to NG or HG. β-actin is used as an internal control. *n* = 3 per group. (H, I) Cdk2 mRNA and protein levels detected in neural retinas of control and STZ mice. β-actin is used as an internal control. *n* = 3 per group. (J) MeRIP-qPCR analysis of m⁶A enrichment on *CDK2* mRNA in HUVECs. *n* = 3 per group. (K) FTO-RIP-qPCR validates the binding between FTO protein and *CDK2* mRNA in untreated HUVECs but not in HUVECs transduced with L-FTO^MU. *n* = 3 per group. (L). MeRIP-qPCR analysis of m⁶A enrichment on *CDK2* mRNA in HUVECs transduced with L-EV, L-FTO^WT or L-FTO^MU. (M, N) mRNA (M) and protein (N) expression of CDK2 in HUVECs transduced with indicated lentivirus. GAPDH is used as an internal control. *n* = 3 per group. (O, P) CDK2 mRNA (O) and protein (P) levels in HUVECs transfected with scramble siRNA or FTO-siRNA. GAPDH was used as an internal control. *n* = 3 per group. (Q) *CDK2* mRNA levels detected by qPCR in HUVECs transduced with indicated lentivirus at 0, 4 and 8 h post actinomycin D treatment. *n* = 3 per group. (R) YTHDF2-RIP-qPCR validation of YTHDF binding to *CDK2* mRNA in HUVECs. *n* = 3 per group. (S) *CDK2* mRNA levels detected by qPCR in HUVECs transfected with scramble siRNA or YTHDF2-siRNA at 0, 6 and 12 h post actinomycin D treatment. *n* = 3 per group. (T, U) CDK2 mRNA (T) and protein (U) levels in HUVECs transfected with scramble siRNA or YTHDF2-siRNA. GAPDH is used as an internal control. *n* = 3 per group. (V) *CDK2* mRNA levels detected by qPCR in HUVECs transfected with scramble siRNA, FTO-siRNA, or FTO- and YTHDF2-siRNA. *n* = 3 per group. Data information: Data represent different numbers (*n*) of biological replicates. Data are shown as mean ± SEM. Two-tailed Student's *t* test is used in (F–J, O–P, R–U). One-way ANOVA followed by Bonferroni's test is used in (K–N, Q, V). NS: not significant (*p* > 0.05); *p < 0.05; **p < 0.01; and ***p < 0.001. Source data are available online for this figure.

#2 presented highest efficiency and was chosen for the following studies (Appendix Fig. 2E,F). Stability of the *CDK2* mRNA was enhanced upon YTHDF2 knocking down (Fig. 8S), followed by increased expression of the *CDK2* mRNA and protein (Fig. 8T,U). We also found that YTHDF2 knocking down could restore the reduced *CDK2* mRNA level induced by FTO-siRNA in HUVECs (Fig. 8V). Collectively, these findings indicated that FTO regulates *CDK2* mRNA stability in an m⁶A-YTHDF2-dependent manner.

## Lactic acid regulates FTO expression via histone lactylation

We continued to analyze the potential upstream mechanisms responsible for the enhanced FTO expression in DR. Disturbed

lactate homeostasis in retina is a common feature of DR (Kolko et al, 2016), we thus aimed to ascertain whether lactate is responsible for the diabetes-driven FTO up-regulation. Increased lactate concentration was detected in HUVECs exposed to high glucose (Fig. 9A) and in STZ retinas (Fig. 9B). Addition of lactate into the culture medium of HUVECs up-regulated both mRNA and protein expression of FTO and CDK2 (Fig. 9C,D), implying that lactate is potentially responsible for FTO up-regulation in HUVEC.

We further explored the modification pattern of lactate in HUVECs. We asked whether lactate mediates FTO expression through direct lactylation of FTO or histone lactylation. We used molecular docking in the molecular operating environment (MOE) to predict the binding affinity between L-lactate molecules and the FTO protein. However, no lysine site in the FTO protein was predicted to bind with

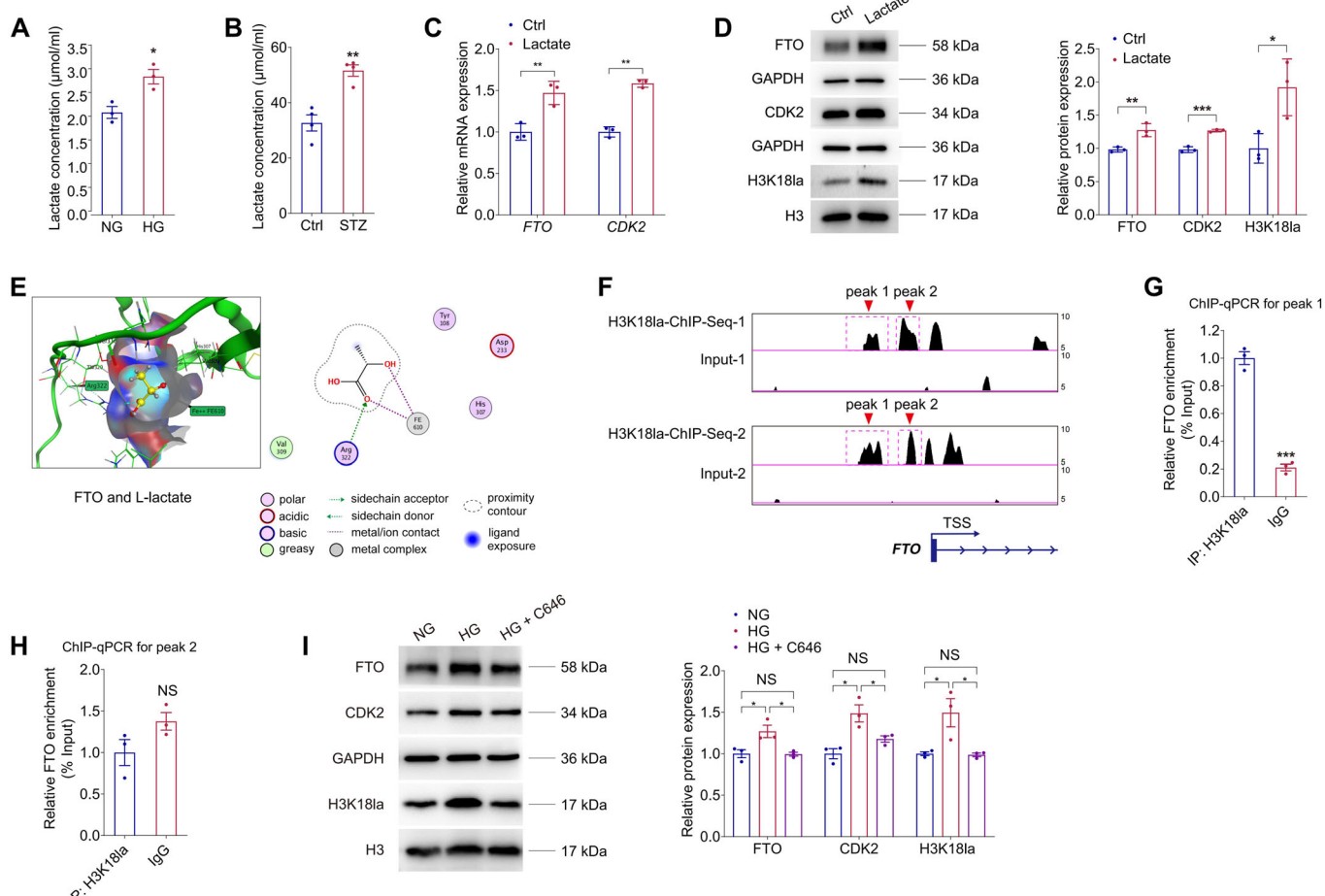

**Figure 9. Lactic acid regulates FTO expression via histone lactylation.**

(A) Lactate concentration in HUVECs treated with NG or HG. $n = 3$ per group. (B) Lactate concentration in neural retinas originated from control and STZ mice. $n = 4$ per group. (C) qPCR presents mRNA levels of *FTO* and *CDK2* in HUVECs treated with or without lactate (10 mM). $n = 3$ per group. (D) Immunoblotting of FTO, CDK2 and H3K18la in HUVECs treated with or without lactate. GAPDH and H3 are used as internal controls. $n = 3$ per group. (E) MOE is used to predict the binding affinity between the FTO protein and L-lactate. (F) Enrichment of the H3K18la signal (two peaks) in the promoter region of the *FTO* gene is demonstrated by ChIP-Seq using anti-H3K18la antibodies (GEO accession number: GSE156675). (G, H) ChIP-qPCR using anti-H3K18la antibodies validates H3K18la enrichment in peak 1 but not peak 2 within the promoter region of FTO in HUVECs. $n = 3$ per group. (I) Immunoblotting of FTO, CDK2 and H3k18la in HUVECs with different treatments. GAPDH and H3 are used as internal controls. $n = 3$ per group. Data information: Data represent different numbers ($n$) of biological replicates. Data are shown as mean ± SEM. Two-tailed Student's *t* test is used in (A–D, G–H). One-way ANOVA followed by Bonferroni's test is used in (I). NS: not significant ($p > 0.05$); *$p < 0.05$; **$p < 0.01$; and ***$p < 0.001$. Source data are available online for this figure.

lactate (Fig. 9E). Next, we tried to verify whether lactate-H3K18la pathway facilitates FTO expression. Elevated H3K18la level was detected in HUVECs added with lactate (Fig. 9D). Chromatin immunoprecipitation followed by sequencing (ChIP-Seq) using anti-H3K18la antibodies (GEO accession number: GSE156675) demonstrated a significant enrichment of the H3K18la signal (two peaks) in the promoter region of the *FTO* gene (Fig. 9F). ChIP-qPCR assay further validated that H3K18la was enriched in peak 1 (Fig. 9G,H). Acetyltransferase p300 is a histone Kla "writer" enzyme (Zhang et al, 2019). Notably, we found the elevated FTO, CDK2 and H3K18la levels in HUVECs treated with high glucose were reduced after addition of the p300 inhibitor C646 (Fig. 9I). C646 at the concentration of 100 μM was selected, which sufficiently suppressed *FTO* and *CDK2* expression in HUVECs (Appendix Fig. S5A) without affecting cell viability (Appendix Fig. S5B–D). Collectively, the above data implied that lactic acid regulates FTO expression via H3K18la.

## FB23-2 suppresses demethylation activity of FTO to inhibit diabetes induced endothelial phenotypes in vitro

FB23-2, which directly binds to FTO and selectively inhibits FTO's m⁶A demethylase activity, is reported to exhibit FTO-dependent anti-proliferation activity via up-regulating global m⁶A levels (Huang et al, 2019). We thus tested the effect of FB23-2 on FTO inhibition in HUVECs using dot blot assay. We found that FB23-2 suppresses the demethylation activity of FTO in HUVECs without affecting its protein expression (Fig. 10A,B). We next detected whether FB23-2 could rescue the high glucose induced pathogenic endothelial phenotypes. EdU assay implied that the high glucose associated proliferation of HUVECs was partly restrained by FB23-2 treatment (Fig. 10C). FB23-2 also alleviated the accelerated HUVEC migration caused by high glucose (Fig. 10D). Moreover, high glucose induced tube formation of HUVECs, as revealed by

increased node number and branching length, was restored by the addition of FB23-2 (Fig. 10E). Collectively, our data suggested that the FTO inhibitor, FB23-2, is a potential agent for diabetes correlated pathogenic endothelial phenotypes.

## Characteristics of NP-FB23-2 and evaluation of its therapeutic efficacy on retinal neovascularization in mice

To further explore the therapeutic potential of FB23-2 in DR, we developed a novel macrophage membrane coated and PLGA-DiI based nanoplatform encapsulating FB23-2 for systemic administration (Fig. 11A). Considering the poor solubility of FB23-2 in water, the polymer cores were prepared using PLGA-DiI as a hydrophobic and fluorescent drug carrier to encapsulate FB23-2. Macrophage membrane coated nanoparticles exhibited significantly enhanced accumulation in retinal neovascular lesions (Xia et al, 2022), we thereby wrapped the PLGA-DiI-FB23-2 particle in macrophage membrane-derived vesicles (M-vesicles) to get the M-FB23-2 (Fig. 11A). In addition, the RGD peptide, a vasculature-targeting tri-peptide motif containing arginine, glycine and aspartic acid, was further linked to M-FB23-2 using 1,2-Distearoyl-sn-

glycero-3-phosphoethanolamine (DSPE) and polyethylene glycol (PEG) to obtain NP-FB23-2 (Fig. 11A). Nanoparticles without FB23-2 but with all the rest components were named as unloaded-NP. Transmission electron microscopy (TEM) demonstrated that NP-FB23-2 was spherical in shape with a core-shell structure, indicating the successful cloaking of PLGA-DiI-FB23-2 with M-vesicles (Fig. 11B). Dynamic light scattering (DLS) measurements demonstrated that NP-FB23-2s show narrow size distributions with the average diameter of $201.71 \pm 22.34$ nm (Fig. 11C). The surface Zeta potential of NP-FB23-2s was $-26.31 \pm 3.09$ mV (Fig. 11D). The negative surface charge will make the nanoparticles penetrate through the retina more easily (Lyu et al, 2021). We uncovered that ~50% of encapsulated FB23-2 is released from NP-FB23-2 after being incubated in phosphate buffer saline (PBS; pH 7.4) at 37 °C for 24 h, while only ~6% of FB23-2 is released within the next 24 h, providing a basis for daily injection (Fig. 11E).

The cellular uptake of NP-FB23-2s into HUVECs was then examined. As revealed by fluorescence microscopy, the DiI labeled NP-FB23-2s and unloaded-NPs presented red fluorescence (Fig. 11F). Fluorescence microscopy demonstrated that NP-FB23-2s exhibited strong fluorescent signal inside the HUVECs

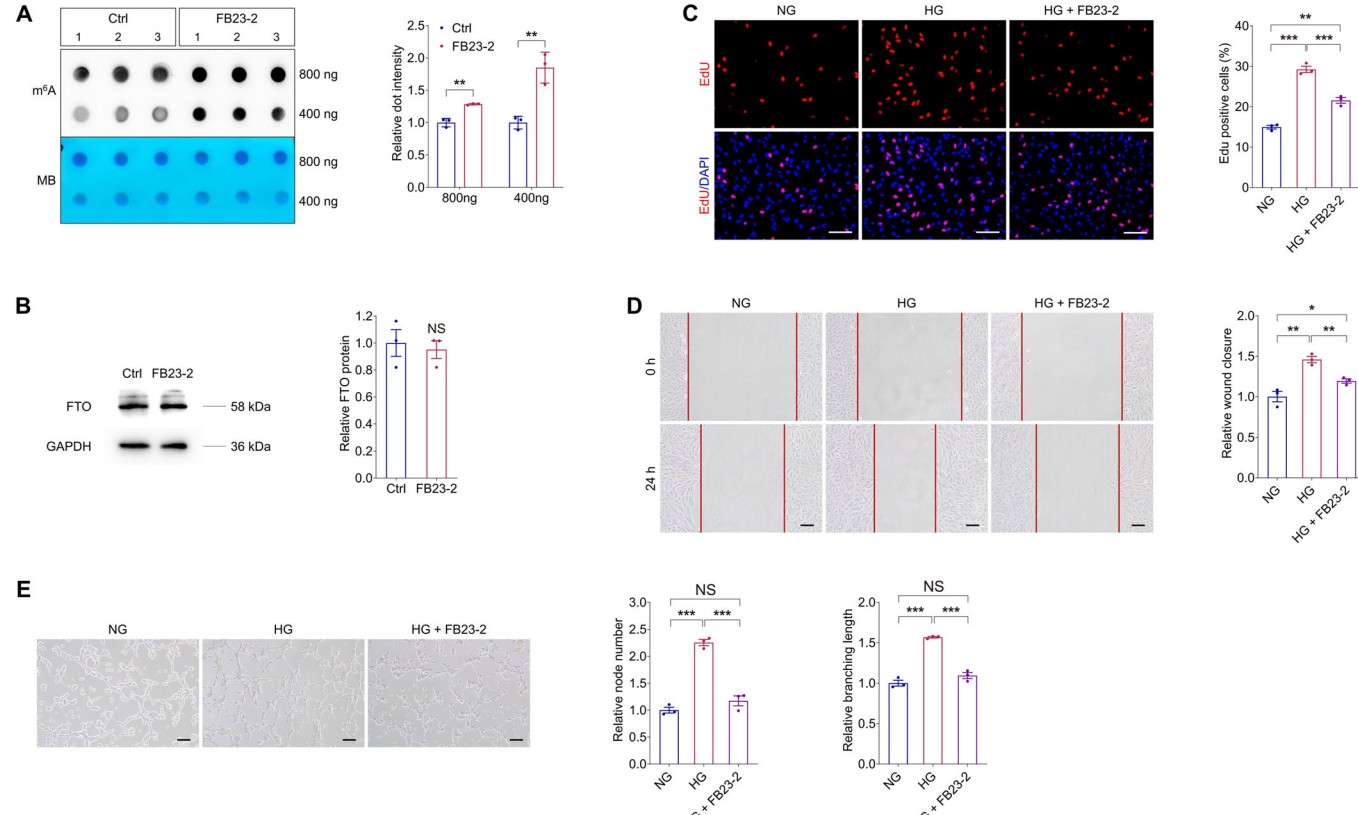

**Figure 10. FB23-2 suppresses demethylation activity of FTO to inhibit diabetes induced endothelial phenotypes in vitro.**

(A) m⁶A dot blot assay of global m⁶A abundance in HUVECs treated with or without FB23-2 (2 µM) using 800 or 400 ng total RNAs. MB staining is used as a loading control. $n = 3$ per group. (B) Immunoblotting of FTO in HUVECs treated with or without FB23-2. GAPDH is used as an internal control. $n = 3$ per group. (C) EdU assay on HUVECs treated with NG, HG, as well as HG and FB23-2. $n = 3$ per group. Scale bar: 60 µm. (D) Scratch test on HUVECs receiving indicated treatments. $n = 3$ per group. Scale bar: 100 µm. (E) Tube formation analyses on HUVECs treated with NG, HG, as well as HG and FB23-2. $n = 3$ per group. Scale bar: 100 µm. Data information: Data represent different numbers ($n$) of biological replicates. Data are shown as mean ± SEM. Two-tailed Student's $t$ test is used in (A, B). One-way ANOVA followed by Bonferroni's test is used in (C–E). NS: not significant ($p > 0.05$); *$p < 0.05$; **$p < 0.01$; and ***$p < 0.001$. Source data are available online for this figure.

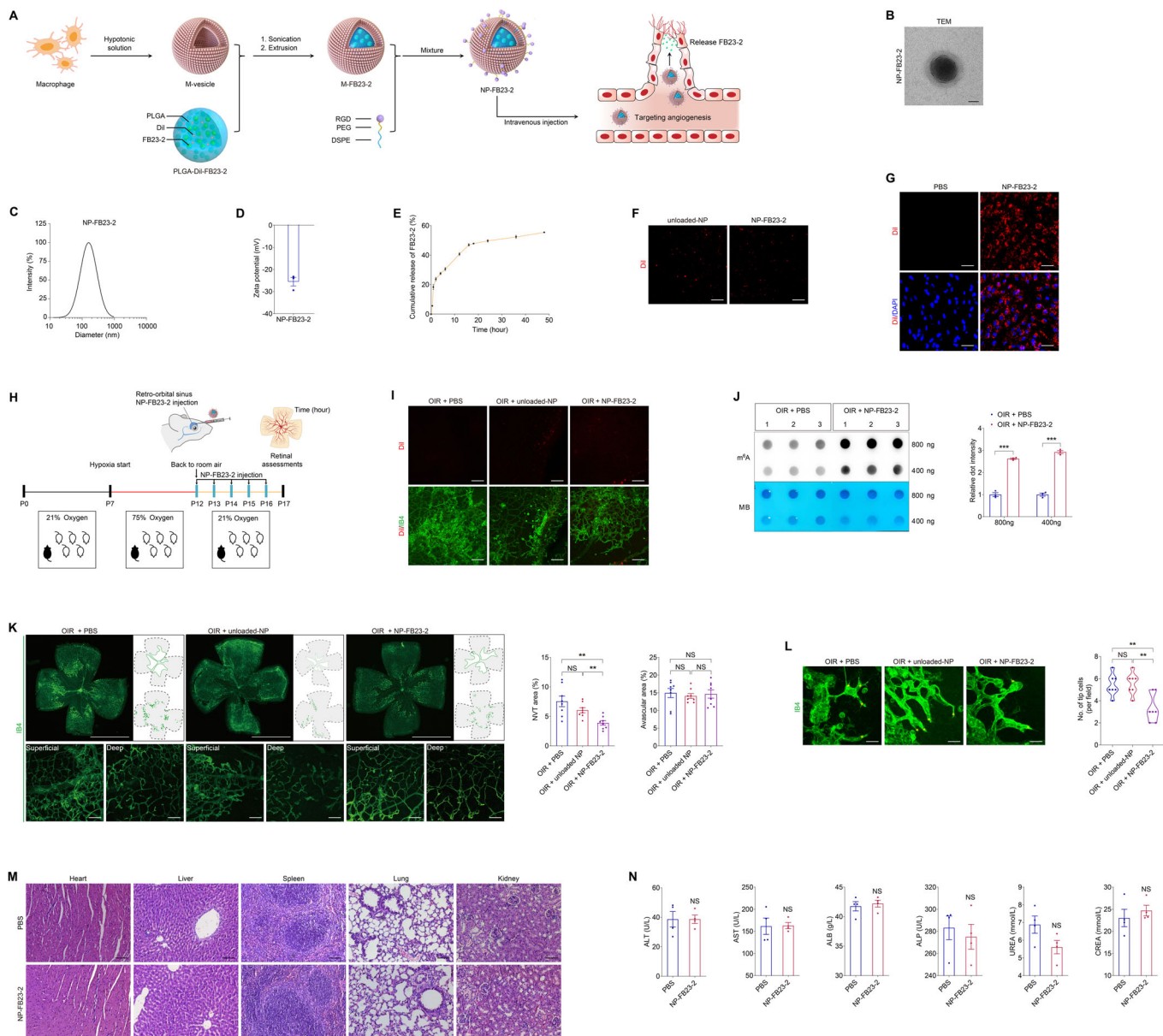

**Figure 11. Characteristics of NP-FB23-2 and evaluation of its therapeutic efficacy on retinal neovascularization in mice.**

(A) Experimental scheme illustrating the synthetic procedures of NP-FB23-2. (B) TEM of DiI labeled NP-FB23-2. *n* = 3. Scale bar: 50 nm. (C) The size distribution profile of NP-FB23-2s. (D) The surface zeta potential of NP-FB23-2s. *n* = 3. (E) Release of FB23-2 from NP-FB23-2s in PBS at 37 °C over the course of 48 h. (F) Fluorescence demonstrated by DiI labeled unloaded-NP and NP-FB23-2s. Scale bar: 20 μm. (G) Fluorescent images of HUVECs treated with PBS or NP-FB23-2s. Cell nuclei are counterstained with DAPI. *n* = 3 per group. Scale bar: 65 μm. (H) Experimental scheme for (I–L). (I) Fluorescent images of neural retinas collected from OIR mice intravenously injected with PBS, unloaded-NP or NP-FB23-2s. Retinal vasculatures are stained with IB4. *n* = 3 per group. Scale bar: 50 μm. (J) m6A dot blot assay of global m6A abundance in retinas of OIR mice intravenously injected with PBS or NP-FB23-2s at P17 using 800 or 400 ng total RNAs. MB staining is applied as a loading control. *n* = 3 per group. (K) Fluorescence staining of IB4 in retinal flat mounts originated from OIR mice intravenously injected with PBS, unloaded-NP or NP-FB23-2s at P17. *n* = 8 per group. Superficial and deep vascular plexuses are shown by magnificent images. Gray dotted lines indicate the edge of the retina. Red lines suggest the avascular area. NVTs are represented by red dots. Representative images along with quantification results of NVT and avascular areas are shown. Scale bar: 2000 μm (up); 50 μm (below). (L) Fluorescence staining of IB4 in retinal flat mounts originated from OIR mice with indicated treatments at P17. *n* = 7 per group. Tip cells are represented by yellow asterisks. Scale bar: 30 μm. (M) Representative H&E staining images of major organs including the heart, liver, spleen, lung and kidney from mice intravenously injected with PBS or NP-FB23-2s. *n* = 3 per group. Scale bar: 75 μm. (N) Blood ALT, AST, CREA, UREA, ALB and ALP levels in mice intravenously injected with PBS or NP-FB23-2s. *n* = 4 per group. Data information: Data represent different numbers (*n*) of biological replicates. Data are shown as mean ± SEM. Two-tailed Student's *t* test is used in (J, N). One-way ANOVA followed by Bonferroni's test is used in (K, L). NS: not significant (*p* > 0.05); **p* < 0.05; ***p* < 0.01; and ****p* < 0.001. Source data are available online for this figure.

(Fig. 11G). Targeting properties of NP-FB23-2s were further evaluated in OIR mice intra-retro-orbital sinus injected with NP-FB23-2s (Fig. 11H). Fluorescent signal of NP-FB23-2s and unloaded-NPs was enriched in retinal vessels of OIR mice (Fig. 11I). No fluorescent signal was detected in OIR mice intravenously injected with PBS (Fig. 11I). Our data demonstrated the in vivo retinal neovasculature-targeting capacity of NP-FB23-2.

We next verified the therapeutic efficacy of NP-FB23-2s in OIR mice intravenously injected with NP-FB23-2s. M⁶A dot blot assay identified increased m⁶A level in total RNAs of OIR neural retinas from mice injected with NP-FB23-2s compared to PBS (Fig. 11J), implying that NP-FB23-2 alleviates the m⁶A modification reduction in OIR retinas. We also noticed that OIR induced NVTs, formed in the superficial vascular plexuses, are suppressed upon NP-FB23-2 injection (Fig. 11K). OIR associated increased amount of endothelial tip cells in the angiogenic area was reduced by intravenous NP-FB23-2 injection (Fig. 11L). However, no difference in avascular area in the central retina and vessel density in the peripheral deep plexuses was detected among OIR mice injected with PBS, unloaded-NP or NP-FB23-2 (Fig. 11K).

Systemic toxicity of intravenous NP-FB23-2 administration was further evaluated using Hematoxylin and Eosin (H&E) staining. No significant toxicity to major organs including heart, liver, spleen, lung and kidney was exhibited in mice intravenously injected with NP-FB23-2 (Fig. 11M). In addition, we conducted blood index analyses of liver function biomarkers [alanine transaminase (ALT) and aspartate transaminase (AST)], kidney function biomarkers [creatinine (CREA) and urea nitrogen (UREA)] and nutritional markers [albumin (ALB) and alkaline phosphatase (ALP)]. We confirmed the absence of significant hepatotoxicity and renotoxicity in mice receiving intravenous NP-FB23-2 injection (Fig. 11N). Collectively, the above data confirmed the therapeutic efficacy of intravenous NP-FB23-2 injection on retinal neovascularization in mice.

# Discussion

Accumulating evidence suggests that dysregulation of m⁶A modulators participates in the occurrence and progression of DR (Ma et al, 2021; Niu et al, 2022; Suo et al, 2022; Zha et al, 2020), while the roles and regulatory network of FTO in DR have never been elucidated. Herein, we detected elevated expression of RNA demethylase FTO in DR. Both in vivo and in vitro studies revealed that FTO overexpression in vascular ECs contributes to DR phenotypes, including angiogenesis, vascular leakage, inflammation and neurodegeneration (Fig. 12). Further assessments validated *CDK2* as a contributor to the FTO induced retinal phenotypes, and lactate-H3K18la pathway as the upstream regulator of FTO under diabetic conditions. We also developed a novel macrophage membrane coated and PLGA-Dil based nanoplatform encapsulating FB23-2, an FTO inhibitor that suppresses diabetes associated endothelial phenotypes, thus providing a promising nanotherapeutic approach for DR.

Recent studies identified that dysregulated m⁶A modification mediated by aberrantly expressed m⁶A writers, erasers and readers contributes to DR progression, showing involvements in various pathological processes of DR. In ECs, Cao et al reported the DR promoting effects of METTL3 by regulating endothelial–mesenchymal transition via the SNHG7/KHSRP/MKL1 axis (Cao et al, 2022), while Zhao et al revealed its DR inhibitory role by mediating endothelial tight and adherens junctions in an ANXA1-dependent manner (Zhao et al, 2022). YTHDF2 inhibits EC proliferation and tube formation to alleviate DR progression by mediating *ITGB1* mRNA instability (Qi et al, 2021). In pericytes, METTL3 governs pericyte dysfunction in DR by repressing PKC-η, FAT4 and PDGFRA expression through YTHDF2-dependent mRNA decay (Suo et al, 2022). In RPE cells, both METTL3 and YTHDF2 show alleviatory roles in high glucose induced RPE pyroptosis (Huang et al, 2022; Zha et al, 2020). Additionally, dysregulated m⁶A modification also contributes to diabetes correlated retinal inflammation. In DR, YTHDF2 reduces the activity and inflammatory responses in retinal Müller cells (Qi et al, 2021), and the m⁶A eraser ALKBH5 mediates m⁶A modification of A20 to enhance M1 polarization of retinal microglia (Chen et al, 2022). These findings imply the critical and extensive involvements of m⁶A regulators in all pathological processes of DR. Nevertheless, the role of m⁶A eraser FTO in DR has been rarely discussed.

FTO dysregulation contributes to various oculopathies and participates in multiple pathogenesis. Reportedly, FTO promotes corneal neovascularization by regulating EC functions in an m⁶A-YTHDF2-dependent manner (Shan et al, 2020). FTO also alleviates AMD by preventing Aβ1-40 induced RPE degeneration via the PKA/CREB signaling pathway (Hu et al, 2023). Moreover, FTO shows uveitis curing effects by modulating m⁶A level of ATF4 to suppress inflammatory cytokine secretion and maintain tight junctions in RPE cells (Tang et al, 2022). Although FTO's role in DR has not been annotated before, its involvement in diabetes and glucose metabolism has been widely discussed. Association between *FTO* gene polymorphisms and type 2 diabetes has been verified in many races (Frayling et al, 2007; Legry et al, 2009), supporting the clinical correlation between FTO and diabetes. Elevated *FTO* mRNA expression is detected in peripheral blood samples from type 2 diabetic patients, resulting in decreased m⁶A content and disturbed glucose metabolism (Shen et al, 2015; Yang et al, 2019). Noteworthy, accumulating evidence suggests the direct regulation of FTO on glucose metabolism (Bravard et al, 2014; Guo et al, 2015). These findings provide fresh insights into the regulatory roles of FTO in oculopathies and diabetes. Herein, our study demonstrated the pathological involvement of FTO in DR. We revealed that FTO up-regulation in EC contributes to DR features by mediating *CDK2* expression in an m⁶A-YTHDF2-dependent manner. However, due to its ubiquitous expression, FTO dysregulation in other retinal cells, including glia cells, pericytes and neural cells, could also contribute to DR. Other potential downstream targets and pathways of FTO in ECs may also exist. The pathogenic role and regulating network of FTO in DR need to be explored systematically in further studies.

Investigating the upstream transcriptional regulation of FTO in EC is important for better understanding of the molecular mechanism of DR. Guo et al elucidated that the transcription factor Foxa2 negatively regulates the basal transcription and expression of FTO in HEK293 cells (Guo et al, 2012). In mouse TM3 cells, aromatic hydrocarbon receptor affects FTO expression through transcriptional regulation (Zhou et al, 2023). STAT3 also binds with FTO promoter to activate its transcription in breast cancer cells (Wang et al, 2021b). Herein, we noticed that FTO up-regulation in EC under diabetic conditions is driven by lactate mediated histone lactylation H3K18la. Lactylation, which links

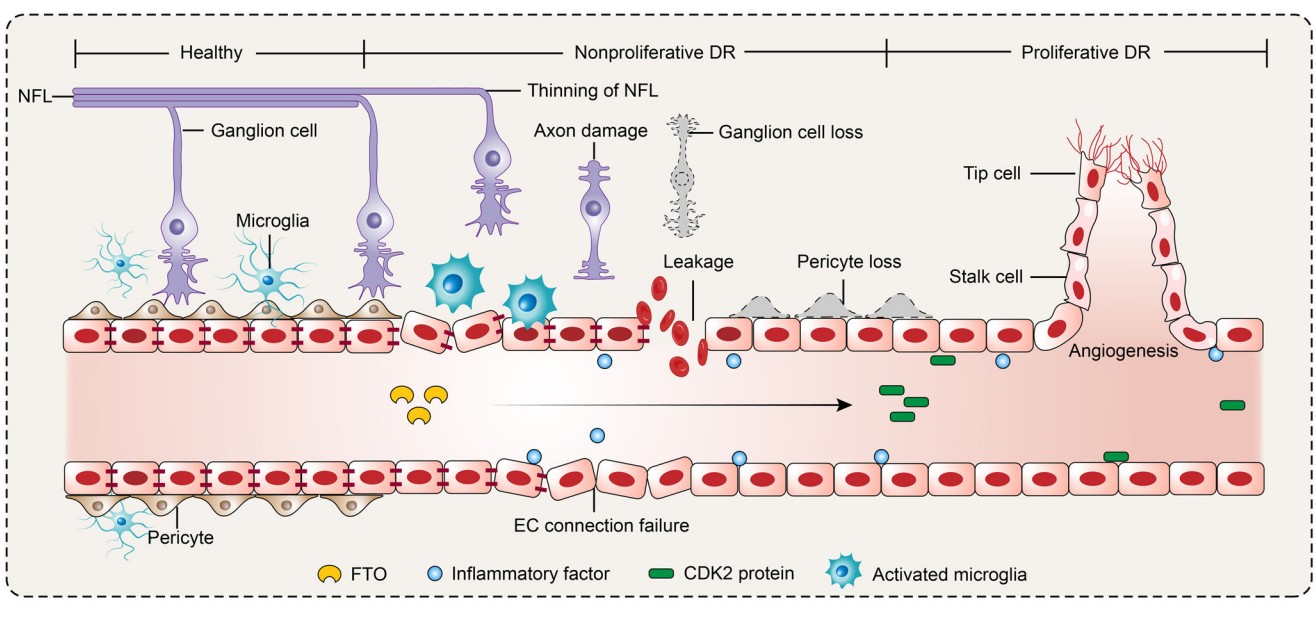

◀ **Figure 12.  Schematic diagram of FTO-mediated effects on DR.**

Driven by histone lactylation H3K18la, FTO up-regulation in ECs triggers vascular endothelial tip cell formation and its crosstalk with pericyte and microglia to aggravate diabetes-induced microvascular dysfunction. FTO mediates DR phenotypes via regulating *CDK2* mRNA stability with YTHDF2 as the reader in an m⁶A-dependent manner.

metabolism and gene regulation, shows deep and extensive involvements in biological and pathological processes (Li et al, 2022; Yu et al, 2021; Zhang et al, 2019). Recent studies identify the regulation of lactylation on oculopathies, while its role in DR has not been elucidated. Lactylation of YY1 in microglia promotes retinal angiogenesis through transcription activation-mediated up-regulation of FGF2 (Wang et al, 2023b). Histone lactylation was found to drive oncogenesis by facilitating YTHDF2 expression in ocular melanoma (Yu et al, 2021). In addition, lactylation also mediates METTL3 expression to promote immunosuppression of tumor-infiltrating myeloid cells (Wei et al, 2018), further emphasizing the interaction between lactylation and m⁶A modification. However, we cannot completely exclude that other factors may be responsible for FTO up-regulation under diabetic conditions due to the complicated etiology and tangled process of DR. Thus, more comprehensive and in-depth investigations on initiating molecular events in DR that trigger FTO dysregulation are warranted.

Herein, we also developed and characterized a macrophage membrane coated and PLGA-Dil based nanoplatform loaded with the FTO inhibitor FB23-2 to treat DR. The macrophage membrane enhances the accumulation of nanoparticles in retinal angiogenic lesions, and the PLGA-based nanosystem ensures the controlled release of FB23-2 (Xia et al, 2022). Targeting efficacy of NP-FB23-2 was validated both in vivo and in vitro. NP-FB23-2s exhibited enhanced cellular uptake into ECs and achieved high concentration in retinal neovasculatures in mice receiving intravenous administration. Moreover, therapeutic efficiency of NP-FB23-2s in promoting m⁶A content and suppressing retinal neovascularization was also validated in mice, providing novel insights and options for therapeutic strategies of DR.

In conclusion, we demonstrated the pathological involvement of m⁶A demethylase FTO in DR. FTO overexpression triggered a series of diabetic retinal phenotypes, including angiogenesis, vascular leakage, inflammation and neurodegeneration, through directly affecting EC features and mediating EC-pericyte/microglia interactions. We also annotated the up- and down-stream regulatory network of FTO in ECs, and developed a novel macrophage membrane coated and PLGA-Dil based nanoplatform encapsulating the FTO inhibitor FB23-2 for systemic administration. This work describes an FTO-mediated regulatory network that coordinates EC biology and retinal homeostasis in DR, provides novel insights into DR pathogenesis, and indicates a promising nanotherapeutic approach for DR.

# Methods

## Ethical approval

FVMs from PDR patients, ERMs from age-matched controls without diabetes and their fundus presentations (including fundus photograph and OCT images) were obtained from Department of Ophthalmology

in The First Affiliated Hospital of Nanjing Medical University. All procedures followed the Association for Research in Vision and Ophthalmology (ARVO) statement on human subjects and the Declaration of Helsinki with written informed consents signed by all individuals before donation. This study was approved and reviewed by the ethical committees from The First Affiliated Hospital of Nanjing Medical University (Approval No.: 2020-SR-545). Animal experiments, conformed to the guidelines of the Care and Use of Laboratory Animals (published by the NIH publication No. 86-23, revised 1996), were approved and consistently reviewed by the ethical review board of Nanjing Medical University (Approval No.: IACUC-2203035 for mice; IACUC-2204056 for zebrafish).

## Cell culture and treatment

HUVECs (Accession number: PCS-100-013; ATCC) were cultured in DMEM/Low Glucose medium or DMEM/High Glucose medium supplemented with 10% fetal bovine serum (FBS; Invitrogen, Carlsbad, CA, USA), penicillin (100 U/mL; Invitrogen) and streptomycin (100 U/mL; Invitrogen). HMC3 cells (Accession number: CRL3304; ATCC) were maintained in DMEM/F12 added with 10% FBS, penicillin (100 U/mL) and streptomycin (100 U/mL). RAW 264.7 cells (Accession number: TIB-71; ATCC) were cultured in DMEM/High Glucose medium supplemented with 10% FBS, penicillin (100 U/mL) and streptomycin (100 U/mL). Complete medium was used short for supplemented culture medium in the following text. Cells were maintained at 37 °C with 21% O₂ and 5% CO₂. Co-cultivation of HUVECs and HMC3 cells was achieved using 0.4-μm-pore size Transwell chambers as indicated in Fig. 5E. For actinomycin D assay, HUVECs were cultured in complete medium added with actinomycin D (1 μg/mL), and were harvested at, 4, 6, 8 and 12 h post treatment, respectively. For lactate and C646 treatment, HUVECs were maintained in complete medium supplemented with L-lactate (10 mM; Sigma-Aldrich, St. Louis, MO, USA) or C646 (MCE, Monmouth Junction, NJ, USA) for 48 h before collection. For FB23-2 and NP-FB23-2 treatment, HUVECs were cultured in complete medium added with FB23-2 (2 μM; MedChemExpress, Princeton, NJ, USA) or NP-FB23-2 (dilution: 1:100) and harvested at 48 h post treatment.

## CCK-8 assay

HUVECs were seeded into 96-well plates to determine the cell viability using the cell counting kit-8 (CCK-8; Beyotime) according to the manufacturer's protocol. Briefly, HUVECs were incubated with the CCK-8 solution at 37 °C for 2 h. The absorbance at 450 nm was determined using a Multiskan FC spectrophotometric plate reader (Thermo Fisher Scientific, Waltham, MA, USA).

## RNA isolation and qPCR

Total RNA was obtained from cell lysates and mice retina using TRIzol reagent (Invitrogen). Nano-Drop ND-1000 spectrophotometer

(Nano-Drop Technologies, Wilmington, DE, USA) was applied to detect the concentration and purity of RNA. cDNA was generated using a PrimeScript RT Kit (Takara, Otsu, Shiga, Japan). RNA levels were determined by qPCR with FastStart Universal SYBR Green Master (ROX; Roche, Basel, Switzerland) using StepOne Plus Real-Time PCR System (Applied Biosystems, Darmstadt, Germany). To normalize mRNA expression, GAPDH expression was assessed in parallel. Detailed primer information is listed in Appendix Table S1.

## M⁶A dot blot assay

Extracted RNA was diluted, denatured, and loaded onto the Amersham Hybond-N+ membrane (Millipore, Boston, MA, USA). The membranes were then subjected to UV cross-linking (10 min), methylene blue (MB) staining (5 min) and 5% BSA blocking (1 h). After that, the membranes were treated with m⁶A antibody at 4 °C overnight, and incubated with horseradish peroxidase (HRP)-conjugated secondary antibody (dilution: 1:5000; ICL, Newberg, OR, USA) at room temperature for an hour. The Tanon-5200Multi Chemiluminescent Imaging System (Tanon Science & Technology, Shanghai, China) was used to visualize the blots. Image J software (http://rsb.info.nih.gov/ij/index.html) was applied to measure dot intensity.

## M⁶A RNA methylation quantification assay

The m⁶A level of total RNA was detected using the EpiQuik m⁶A RNA Methylation Quantification Kit (Epigentek Group, Farmingdale, NY, USA) according to the manufacturer's protocol. Briefly, 200 ng of extracted RNA, negative and positive controls were coated on assay wells, incubated with capture antibody solution and treated with detection antibody solution. The m⁶A level was then determined using a Synergy 4 automatic microplate reader (Agilent BioTek; Winooski, VT, USA) at 450 nm.

## Mouse breeding and manipulations

C57BL/6J mice (strain ID: 219; sex: male), purchased from Charles River Laboratories (Wilmington, MA, USA), were raised in a specific pathogen-free facility at Nanjing Medical University with a 12 h light/dark cycle at 28.5 °C. Embryos were produced through natural mating. No randomization was applied. Before all invasive operations and examinations, mice were fully anesthetized through intra-peritoneal injection of ketamine (80 mg/kg) and xylazine (4 mg/kg) with pupils dilated using 1% cyclopentolate-HCL and 2.5% phenylephrine.

STZ mice were generated by daily intra-peritoneal injection of STZ (50 mg/kg; Sigma-Aldrich) for 5 days. Before injection, 6-week-old male mice were in abrosia for 8 h. Water was supplied during fasting. Blood glucose was continuously measured till 7 days after injection using a Contour TS blood glucose monitor glucometer (Bayer, Leverkusen, Germany). Mice with blood glucose of over 15 mmol/L were considered as diabetic mice and were selected for further investigations. To generate OIR mice, P7 neonatal mice together with their nursing mother were housed in a closed chamber with 75% O₂ for 5 days. Food and water were supplied as normal.

For intra-vitreal injection, 1 µL solution containing AAV-blank/AAV-Fto supernatant ($10^{12}$–$10^{13}$ genome copies/mL; AAV Serotype 2) was delivered into the vitreous chamber of mice through an incision into the sclera (1 mm posterior of the superior limbus) using a syringe with a 33-gauge needle (Hamilton, Bonaduz, Switzerland). For intra-cardiac injection of Texas red dextran, mouse thoracic cavity was carefully opened to expose the heart. One mL Texas red dextran solution (70,000 MW; 2 mg/mL; Invitrogen) was injected into the left ventricle at an even speed for 1 min. Mouse was sacrificed immediately after perfusion with eyes enucleated to produce retinal flat mounts. For intravenous injection, Evans blue solution (4 mL/kg; Sigma-Aldrich) was administered through the tail vein. Mice were sacrificed with eyes enucleated at 2 h post injection to make retinal flat mounts. NP-FB23-2 (3 mg/kg) was delivered through the retro-orbital sinus into neonatal mice with a 31-gauge insulin syringe (BD Biosciences, San Jose, CA, USA).

## Mounting of mice retina

For tissue preparation, mice were anesthetized and sacrificed after the ophthalmic examinations with eyes enucleated and connective tissues trimmed. After careful removal of the anterior segments and vitreous, the remaining posterior eyecups were used for further tests. For dot blot, qPCR and immunoblotting assays, neural retinas were isolated and sent for RNA and protein extractions. For retinal flat mounting staining, eyecups were fixed in 4% paraformaldehyde (PFA) at 4 °C for at least 2 h. Neural retinas were then carefully separated from the posterior eyecup and trimmed into a four-leaf clover-shaped retinal flat mount.

## Immunoblotting

For protein isolation, cells and mice tissue were collected and fragmented in lysis buffer (Beyotime, Shanghai, China) supplemented with protease inhibitors cocktail (Roche). Isolated proteins were then segregated by gel electrophoresis and transferred to polyvinylidene fluoride membranes (Millipore). Membranes were subsequently incubated in primary antibodies (Appendix Table S2) at 4 °C overnight and probed by HRP-conjugated secondary antibodies (dilution: 1:10,000; ICL) for an hour at room temperature. Blots were developed using the Tanon-5200Multi Chemiluminescent Imaging System. Proteins were quantified using Image J software.

## Cell transduction and transfection

Empty lentiviral plasmid with FLAG tag (Ubi-MCS-3FLAG-SV40-puromycin; L-EV), lentiviral plasmid containing the CDS of wild-type human *FTO* gene (Ubi-MCS-FTO-3FLAG-SV40-puromycin; L-FTO) and with the two catalytically inactive mutations H231A and D233A (Ubi-MCS-FTO^MU-3FLAG-SV40-puromycin; L-FTO^MU) were constructed by GeneChem (Shanghai, China). Lentiviral plasmid L-EV, L-FTO, or L-FTO^MU was co-transfected with pCMV-Gag/Pol and pCMV-VSVG into HEK293T cells using Lipofectamine 3000 transfection reagent (Invitrogen) according to the manufacturer's instructions. Packaged lentiviral particles in the supernatant were collected at 48 h post-transfection, mixed with polybrene (2 µg/mL; Sigma-Aldrich), and added into HUVECs at a multiplicity of infection (MOI) of 10. Transduced HUVECs were sorted out using puromycin (5 µg/mL; Sigma-Aldrich) and sent for further analyses. For transfection assay, HUVECs were transfected

with indicated siRNAs using Lipofectamine 3000. Scramble siRNA and siRNAs targeting *FTO* or *YTHDF2* were purchased from RiboBio (Guangzhou, China) with sequences detailed in Appendix Table S3.

## Immunofluorescence staining

FVMs, ERMs, cells, and posterior eyecups of mice were fixed in 4% PFA. Neural retinas were separated from the posterior eyecups to obtain neural retinal flat mounts. After being permeabilized with 0.5% Triton X-100 (Sigma-Aldrich) and blocked in 1% BSA, FVMs, ERMs, cells and neural retinal flat mounts were sequentially incubated with primary antibodies (Appendix Table S2) at 4 °C overnight, and corresponding fluorescence-conjugated secondary antibodies (dilution: 1:100 for tissue staining; 1:1000 for cell staining; Invitrogen) at room temperature for 2 h. Cell nuclei were counterstained by DAPI (Sigma-Aldrich). Fluorescence was observed with an upright microscope (DM4000 B, Leica), a cell imaging multimode reader (BioTek Cytation 1, Agilent BioTek) and an inverted microscope (DMi8, Leica) equipped with THUNDER imaging system (Leica). Fluorescence intensity was quantified with Image J software.

## RNA-Seq, MeRIP-Seq, and MeRIP-qPCR

Extracted total RNAs with high quality were sent for rRNA depletion using a human TransNGS rRNA Depletion Kit (Trans-Gen Biotech, Beijing, China) and RNA purification with MagicPure RNA beads (TransGen Biotech). RNAs were then chemically segmented into 60–200 nt fragmentations with 10% of total RNA separated as input control (RNA-Seq). The rest RNAs were incubated with Dynabeads Protein A (Thermo Fisher Scientific) and m⁶A antibody at 4 °C for 2 h. M⁶A RNA enrichment was then detected via high-throughput sequencing (MeRIP-Seq) or qPCR (MeRIP-qPCR). For RNA-Seq and MeRIP-Seq, purified input RNAs and immunoprecipitated RNA fragments were sent for library construction with the VAHTS Total RNA-Seq (H/M/R) Library Prep Kit for Illumina (Vazyme, Nanjing, China) and sequencing using the Illumina HiSeq 2500 (Illumina, San Diego, CA, USA). Quality control and raw sequencing data filtration were achieved with the FastQC and fastp software. Reads were then aligned to the human genome GRCh38/hg38 using STAR software. Reads per kilobase per million mapped reads (RPKM) values were measured in 5′-UTR, CDS, and 3′-UTR of all genes using deepTools software. For RNA-Seq, expression of mRNAs in input RNAs was analyzed using Stringtie algorithm with differential expression annotated by Deseq2 software. For MeRIP-Seq, Methylated MeRIP peaks were called using exomePeak2 software, annotated with Annovar software, and matched to motifs using Homer software. For MeRIP-qPCR, immunoprecipitated RNA fragments as well as input RNAs were reversely transcribed into cDNAs. Enrichment of m⁶A-immunopurified *FTO* mRNA was measured with qPCR and normalized to the input.

## Cell cycle analyses

Cell cycle was analyzed using the cell cycle staining kit (Multisciences Biotech, Hangzhou, China) according to the manufacturer's instructions. Collected cells were suspended in DNA staining solution containing permeabilization solution for 30 min. Cell cycle was detected by flow cytometry (Beckman Coulter, Brea, CA, USA) with data analyzed using the win cycle software. A total of 10,000 cells from each sample were measured to modulate the cell cycle.

## EdU assay

Cell proliferation was determined using the EdU Apollo567 Kit (RiboBio) per the manufacturer's protocols. For in vitro analyses, cells were incubated in EdU solution for 2 h, fixed with 4% PFA, permeabilized using 0.5% Triton X-100, and fluorescently labeled in Apollo solution. For in vivo assessments, mice were intraperitoneally injected with EdU solution (5 mg/kg) once a day from P13 to P17. Mice were sacrificed with eyes enucleated at 5 h post last injection to make neural retinal flat mounts. Neural retinas were then fixed with 4% PFA, permeabilized using 0.5% Triton X-100, and fluorescently labeled in Apollo solution. Proliferating cells showing red signals were visualized using a DM4000 B upright microscope or an inverted microscope (DMi8) equipped with THUNDER imaging system.

## Transwell migration assay

Cell migration was examined using an 8-μm-pore size Transwell migration chamber. Cells were seeded into the upper chamber with its bottom immersed in culture medium. Cell migration was allowed to proceed for 24 h at 37 °C with 21% $O_2$ and 5% $CO_2$ before collection. Cells that migrated to the below surface of the upper chamber were then stained with crystal violet (Beyotime) for 15 min. Different views were randomly chosen and recorded with an ECLIPSE Ts2 inverted microscope (Nikon, Tokyo, Japan) with average counting taken.

## Scratch test

Cell migration was also determined using scratch test. HUVEC monolayers were scraped in straight lines with pipet tips to create scratches, and gently washed to remove cell debris and smooth scratch edges. Images of the same scratch site were recorded right after the scratch and 24 h after the scratch using an ECLIPSE Ts2 inverted microscope, and were analyzed with the Image J software.

## Migration assay with the RTCA system

Real-time rates of HUVECs migration were detected using the RTCA system (Roche) according to the manufacturer's protocol. Briefly, cells were seeded into the E-Plate and maintained in complete medium. Impedance value for each well was automatically recorded by the RTCA system as a CI value. Migration rates were calculated based on the slope of the line between two given time points.

## Tube formation assay

HUVECs were seeded onto growth factor-reduced matrigel (BD Biosciences) in 24-well plates to observe formation of capillary-like structures. Images were taken at 5 h post plantation using an

ECLIPSE Ts2 inverted microscope. Image J software was applied to quantify node number and branching length of HUVECs.

## mRNA synthesis

CDS of zebrafish *fto* mRNA was synthesized and inserted into the pCS2+ plasmid (Addgene, Watertown, MA, USA) to get the linearized pCS2 + -FTO plasmid for in vivo transcription. The inserted sequence of the recombinant plasmid was validated using Sanger sequencing in both directions. Capped and tailed zebrafish *fto* mRNA was generated with a mMESSAGE mMACHINE T7 Ultra Kit (Ambion, Austin, TX, USA) and purified using an RNeasy Mini Kit (Qiagen, Hilden, Germany) according to the manufacturer's protocol.

## Zebrafish manipulations

The transgenic zebrafish strain *Tg(LR57:GFP)* was a kind gift from the Zebrafish Center of Nantong University. One- to two-cell-stage [(0 h post fertilization (hpf)] zebrafish embryos were collected and randomly microinjected with one nano-liter (nL) solution containing different dosages of purified *fto* mRNA or antisense mRNA. At 24 hpf, embryos with systemic deformities were discarded. The rest embryos were further randomly divided into two groups and maintained in fish-raising water with or without 3% glucose. At 48 hpf, embryos receiving distinct treatments and showing normal systemic appearance were selected for vascular imaging. Truncal and ocular vasculatures of living zebrafish were visualized using a Nikon A1 HD25 confocal microscope system (Nikon).

## Ophthalmic examinations

For fundus examinations, mice were anesthetized with pupils dilated. Fundus photo was taken using a ZEISS CLARUS 500 fundus camera (Carl Zeiss, Jena, Germany). FFA was conducted after intra-peritoneal injection of fluorescein sodium (International Medication Systems, South El Monte, CA, USA) at 2 μL/g body weight. Fluorescent fundus images were acquired through Heidelberg Retina Angiograph 2 (Heidelberg Engineering, Heidelberg, Germany).

## Retinal trypsin digestion and PAS staining

Fixed and isolated neural retinas were washed in distilled water overnight and digested in 3% trypsin (BioFroxx GmbH, Einhausen, Germany) at 37 °C for an hour. Digested retinas were carefully washed and gently shaken with a pipette under the microscope to separate the retinal vasculatures. For PAS staining, the isolated retinal vessels were sequentially stained with periodic acid solution and Schiff's reagent. Images were taken with an upright microscope (DM4000 B).

## ELISA

Culture medium of HUVECs was collected to determine VEGF-A secretion using a commercial human VEGF-A ELISA kit (Beijing 4 A Biotech Co., Ltd, Beijing, China) per the manufacturer's protocol. The absorbance at 450 nm was determined using a Multiskan FC spectrophotometric plate reader (Thermo Fisher Scientific).

## TMT-based quantitative proteomic analysis

Cells were harvested with intra-cellular proteins extracted and digested with trypsin. Peptides were then labeled with TMT using a TMT10plex mass tag labeling kit (Thermo Fisher Scientific) per the manufacturer's instructions. TMT-labeled peptides were then separated by high pH reverse-phase high performance liquid chromatography (HPLC) with C18 columns (Agilent BioTek) and dried in a vacuum centrifuge. Liquid chromatography-tandem mass spectrometry (LC-MS/MS) was subsequently performed. All fragments were sequentially dissolved in aqueous solution (0.1% formic acid and 2% acetonitrile), loaded onto a home-made reverse-phase analytical column, and subjected to an EASY-nLC™ 1000 ultraperformance liquid chromatography (UPLC) system (Thermo Fisher Scientific) at a constant flow rate of 400 nL/min. For MS settings, the applied electrospray voltage was 2.0 kV and the *m/z* range was 350 to 1800 for the complete scan. Peptide fragments were quantified with Parallel Reaction Monitoring (PRM). Proteins were identified by Swissprot database and quantified using Proteome Discoverer 2.0.

## LiquiChip

Culture medium of HUVECs was collected and analyzed with the Bio-Plex Pro™ Human Cytokine 27-plex Panel (Bio-Rad Laboratories, Hercules, CA, USA) in accordance with the manufacturer's protocols. Briefly, culture medium was sequentially mixed with fluorescence-labeled capture beads and highly specific monoclonal antibodies. The mixture was then resuspended and irradiated. Concentrations of 27 cytokines and chemokines in the mixture were detected using the Luminex™ 200™ instrument system (Luminex, Austin, TX, USA). Data were analyzed with the xPONENT® software.

## RIP-qPCR

RIP experiment was performed using the Magna RIP™ Kit (Millipore) per the manufacturer's instructions. Protein A/G magnetic beads were incubated with antibodies against IgG, FTO and YTHDF2 respectively. qPCR was used to measure the enrichment of protein-bounded RNA. Results were normalized to the input.

## Lactate content analysis

Lactate concentration was determined using a Lactic Acid Content Assay Kit (Solarbio, Beijing, China) according to the manufacturer's protocol. HUVECs were harvested, sonicated and incubated with indicated reagents. Cell deposits were extracted and dissolved in absolute alcohol. Lactate content was monitored using a Synergy 4 automatic microplate reader (Agilent BioTek) at 570 nm.

## ChIP-qPCR

ChIP assay was conducted using the SimpleChIP® Enzymatic Chromatin IP Kit (Cell Signaling Technology, Danvers, Massachusetts, USA) per the manufacturer's instructions. Cells were crosslinked with 1% formaldehyde, sonicated and digested to acquire chromatin fragments. DNA fragments were collected and incubated with H3K18la antibody at 4 °C overnight. Enrichment of

## The paper explained

### Problem

Diabetic retinopathy (DR), a major microvascular complication of diabetes, is a leading cause of vision loss in the working-age population. Therapeutic approaches to modulate DR are poorly developed due to the limited drug targets identified. Fresh insights into DR pathology are needed to reveal novel therapeutic targets.

### Results

We detected increased FTO, an $N^6$-methyladenosine ($m^6A$) demethylase, under diabetic conditions. We further identified that FTO facilitates angiogenesis, triggers diabetic microvascular leakage, and induces retinal inflammation and neurodegeneration in DR. Mechanistically, FTO modulated CDK2 mRNA stability in an $m^6A$-YTHDF2-dependent manner. FTO up-regulation was driven by lactate-mediated histone lactylation.

### Impact

Our demonstration that systemic administration of FB23-2, an inhibitor to FTO's $m^6A$ demethylase activity, is therapeutic for DR mice, suggests it is worth further investigation for potential future treatment of DR patients.

H3K18la-bounded DNA was measured using qPCR and normalized to the input.

## Preparation of NP-FB23-2

We used nanoprecipitation method to prepare PLGA nanoparticles encapsulating FB23-2. Briefly, PLGA (5 mg, MW = 100,000; Daigang Biology, Jinan, China), distearoyl phosphoethanolamine-poly (DSPE-PEG-COOH; 2 mg; Qiyue Biology, Xi'an, China), 1, 2-dioleoyl-sn-glycero-3-phosphocholine (DOPC; 1 mg; Qiyue Biology) and FB23-2 (2 mg) were dissolved in 1 mL organic phase. The mixture was added dropwise into 4 mL PBS buffer with stirring. The suspension was then sonicated at 100 W for 15 min and dialyzed in a 3500DA dialysis bag for 24 h. Next, 0.1 wt % DiI (Beyotime) was loaded into the collected PLGA-FB23-2 solution to obtain fluorescently labeled PLGA-DiI-FB23-2 nanoparticles.

To obtain purified macrophage membranes, RAW 264.7 cells were collected and treated according to a previously described protocol (Xia et al, 2022). Cells were resuspended in ice-cold $0.1 \times$ TM buffer (Beyotime) containing protease and phosphatase inhibitor cocktail (Thermo Fisher Scientific), and disrupted by freezing, thawing and a homogenizer (IKA, Staufen, Germany). The suspension was then sent for three rounds of centrifugations (3200 g for 5 min; 20,000 g for 25 min; 100,000 g for 60 min) to get M-vesicles. The PLGA-DiI-FB23-2 nanoparticles and the M-vesicles were sonicated separately for 5 min and subsequently mixed at a 1:1 (w/w) ratio of membrane protein to polymer. The mixture was sequentially extruded through 400 nm and 200 nm polycarbonate membranes using a micro-extruder (Avestin, Ottawa, Canada) for 10 rounds to obtain macrophage membrane coated PLGA-DiI-FB23-2s nanoparticles.

Finally, DSPE-PEG-RGD (1 mg; Qiyue Biology) dissolved in 0.5 mL of N, N-dimethylformamide (DMF; Sigma-Aldrich) was added to the macrophage membrane coated PLGA-DiI-FB23-2s nanoparticles under stirring. The nanoparticle solution was then dialyzed in a

10,000DA dialysis bag for 24 h, and concentrated using a 10,000DA MWCO ultra centrifugal filter tube (Millipore) to obtain NP-FB23-2.

## Characterization of NP-FB23-2

To visualize NP-FB23-2, solution containing NP-FB23-2 was dropped onto a copper and observed using an FEI Tecnai G2 Spirit BioTWIN electron microscope (FEI, Hillsboro, OR, USA). Size and surface zeta potential of NP-FB23-2 were measured using the NanoBrook 90Plus particle size analyzer (Brookhaven Instruments, New York, NY, USA) based on the principles of DLS. The release profile of FB23-2 from NP-FB23-2 was plotted for 48 h. Briefly, 5 mL solution containing NP-FB23-2 was added into dialysis bags (MWCO: 8–14 kDa; Solarbio) and immersed into 50 mL PBS (pH 7.4). At determined time intervals, 1 mL of the solution was extracted to measure the concentration of released FB23-2 using HPLC, and an equivalent amount of fresh PBS solution was added into the remaining sample.

## H&E staining

To obtain paraffin sections of mice heart, liver, spleen, lung and kidney, tissues were isolated and fixed in 4% PFA, dehydrated and transparentized with conventional grade of alcohol and xylo, embedded in paraffin, and sectioned at 3–5 μm thickness. Paraffin sections of mice tissues were dewaxed using xylo and conventional grade of alcohol, and subsequently stained with hematoxylin and eosin to visualize histological structures under a DM4000 B upright microscope.

## Blood biochemistry

Blood ALT, AST, CREA, UREA, ALB, and ALP levels in mice receiving distinct treatments were determined using an automatic biochemical analyzer (Chemray 800; Rayto, Shen Zhen, China).

## Quantification and statistical analysis

GraphPad Prism (v 4.0; GraphPad Software, San Diego, CA, USA) was used for statistical analyses. Two-tailed Student's $t$ test was utilized for comparison between the two groups. One-way analysis of variance (ANOVA) coupled with the Bonferroni's post hoc test was used for comparisons among three groups. Data were presented as mean ± standard error of the mean (SEM). $p < 0.05$ was considered of statistical significance. Detailed replicate information for each experiment was written in the figure legends. Quantifications were performed blinded by the investigators.

## Data availability

The RNA-Seq, MeRIP-Seq, and LiquiChip data have been deposited to the GEO dataset with the accession numbers GSE232135 (for RNA-Seq and MeRIP-Seq data) and GSE246852 (for LiquiChip data). The mass spectrometry proteomics data have been deposited to the ProteomeXchange Consortium via the iProX partner repository with the dataset identifier PXD046601.

## Peer review information

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

## Acknowledgements

We thank all donors for their donations. We are grateful to Professor Dong Li and Dr Xin-Yu Wang (Institute of Biophysics, Chinese Academy of Sciences) for technical supports on MeRIP-Seq, and Professor Dong Liu (Nantong University) for supports on zebrafish experiments. This study was supported by National Natural Science Foundation of China (82070974 to XC, 82271100 to Q-HL); Natural Science Foundation of Jiangsu Province (BK20231371 to XC); and Jiangsu "333" Advanced Talent-training Project to XC. The funders had no role in study design, data collection and analysis, decision to publish, or preparation of the manuscript.

## Author contributions

**Xue Chen**: Conceptualization; Supervision; Funding acquisition; Validation; Writing—original draft; Project administration. **Ying Wang**: Data curation; Formal analysis; Investigation; Methodology; Writing—review and editing. **Jia-Nan Wang**: Data curation; Formal analysis; Investigation; Methodology; Writing—review and editing. **Yi-Chen Zhang**: Data curation; Formal analysis; Investigation; Writing—review and editing. **Ye-Ran Zhang**: Formal analysis; Writing—review and editing. **Ru-Xu Sun**: Data curation; Formal analysis; Investigation; Visualization. **Bing Qin**: Resources; Methodology. **Yuan-Xin Dai**: Resources; Data curation; Methodology. **Hong-Jing Zhu**: Investigation; Visualization. **Jin-Xiang Zhao**: Data curation; Investigation; Methodology. **Wei-Wei Zhang**: Resources; Visualization. **Jiang-Dong Ji**: Resources; Visualization. **Song-Tao Yuan**: Resources; Validation. **Qun-Dong Shen**: Resources; Methodology. **Qing-Huai Liu**: Supervision; Funding acquisition; Project administration; Writing—review and editing.

## Disclosure and competing interests statement

The authors declare no competing interests.

