## [Peer Review File · EMBO Molecular Medicine]

Lactylation-driven FTO targets CDK2 to aggravate microvascular anomalies in diabetic retinopathy

Xue Chen, Ying Wang, Jia-Nan Wang, Yi-Chen Zhang, Ye-Ran Zhang, Ru-Xu Sun, Bing Qin, Yuan-Xin Dai, Hong-Jing Zhu, Jin-Xiang Zhao, Wei-Wei Zhang, Jiangdong Ji, Song-Tao Yuan, Qun-Dong Shen, and Qing-Huai Liu

DOI: [10.15252/emmm.202318033](https://doi.org/10.15252/emmm.202318033)

Corresponding author(s): Qing-Huai Liu (liuqh@njmu.edu.cn), Xue Chen (chenxue@jsph.org.cn)

Review Timeline:

Submission Date:	1st Jun 23
Editorial Decision:	9th Jul 23
Revision Received:	2nd Nov 23
Editorial Decision:	20th Nov 23
Revision Received:	10th Dec 23
Accepted:	2nd Jan 24

Editor: Kelly Anderson

Transaction Report:

9th Jul 2023

Dear Dr. Liu,

Thank you for the submission of your manuscript to EMBO Molecular Medicine. We have now received feedback from the reviewers who agreed to evaluate your manuscript. As you will see from the reports below, the referees acknowledge the interest of the study.

Addressing the reviewers' concerns in full will be necessary for further considering the manuscript in our journal, and acceptance of the manuscript will entail a second round of review. EMBO Molecular Medicine encourages a single round of revision only and therefore, acceptance or rejection of the manuscript will depend on the completeness of your responses included in the next, final version of the manuscript. For this reason, and to save you from any frustrations in the end, I would strongly advise against returning an incomplete revision. It is often good to discuss your plan to address referee concerns and I am available to do so via email or zoom in the coming weeks.

Revised manuscripts should be submitted within three months of a request for revision; they will otherwise be treated as new submissions, except under exceptional circumstances.

I look forward to seeing a revised form of your manuscript as soon as possible.

Yours sincerely,

Kelly

Kelly M Anderson, PhD
Scientific Editor
EMBO Molecular Medicine

We require:

- 1) A .docx formatted version of the manuscript text (including legends for main figures, EV figures and tables). Please make sure that the changes are highlighted to be clearly visible.
- 2) Individual production quality figure files as .eps, .tif, .jpg (one file per figure). For guidance, download the 'Figure Guide PDF': (<https://www.embopress.org/page/journal/17574684/authorguide#figureformat>).
- 3) A .docx formatted letter INCLUDING the reviewers' reports and your detailed point-by-point responses to their comments. As part of the EMBO Press transparent editorial process, the point-by-point response is part of the Review Process File (RPF), which will be published alongside your paper.
- 4) A complete author checklist, which you can download from our author guidelines (<https://www.embopress.org/page/journal/17574684/authorguide#submissionofrevisions>). Please insert information in the checklist that is also reflected in the manuscript. The completed author checklist will also be part of the RPF.
- 5) Please note that all corresponding authors are required to supply an ORCID ID for their name upon submission of a revised manuscript.
- 6) It is mandatory to include a 'Data Availability' section after the Materials and Methods. Before submitting your revision, primary datasets produced in this study need to be deposited in an appropriate public database, and the accession numbers and database listed under 'Data Availability'. Please remember to provide a reviewer password if the datasets are not yet public (see <https://www.embopress.org/page/journal/17574684/authorguide#dataavailability>).

13) Author contributions: You will be asked to provide CRediT (Contributor Role Taxonomy) terms in the submission system. These replace a narrative author contribution section in the manuscript.

14) A Conflict of Interest statement should be provided in the main text.

Please note: When submitting your revision you will be prompted to enter your funding and payment information. This will allow Wiley to send you a quote for the article processing charge (APC) in case of acceptance. This quote takes into account any reduction or fee waivers that you may be eligible for. Authors do not need to pay any fees before their manuscript is accepted and transferred to the publisher.

EMBO Press participates in many Publish and Read agreements that allow authors to publish Open Access with reduced/no publication charges. Check your eligibility: <https://authorservices.wiley.com/author-resources/Journal-Authors/open-access/affiliation-policies-payments/index.html>

**** Reviewer's comments ****

Referee #1 (Comments on Novelty/Model System for Author):

Lacking important controls especially undermines the in vivo experiments. The zebrafish experiment is appropriate, however is not well executed and may bring up more questions than answers. The OIR experiment and the STZ experiment lacks the naive group (OIR only, or STZ only).

Referee #1 (Remarks for Author):

Chen et al. presented a well-crafted, and data-enriched manuscript to investigate if and how FTO, an eraser gene in m6A modification, is involved in vascular changes associated with experimental diabetic retinopathy. To this end, they integrated multiple lines of in vitro and in vivo experiments to back up their claims. Major achievements include the identification of upstream regulators and downstream effectors of FTO, whose expression is inversely correlated with diabetic vascular conditions. In addition, the authors engineered an enabling platform to deliver an FTO inhibitor systemically to treat diabetic vascular complications. Overall, this paper could bring additional insights of FTO to vascular biology, and highlight its druggability in managing diabetic retinopathy. However, several key issues need to be addressed before it can be recommended for publications.

The authors utilized HUVEC extensively and extended the in vitro findings to in vivo experiments, which were less compelling and would require additional important controls. As a minor comment, I advise the authors to organize strong in vivo data first and use in vitro experiments secondarily for MOA investigation and phenotype confirmation. Below are my specific questions and comments.

For all the bar graphs, I suggest the authors plot individual data to allow clear visualization of data distribution and variability.

Lack of controls:

For in vitro experiments, it is very important to include a naive group, meaning no lentivirus, OR no siRNA treatment.

For in vivo experiments, it is very important to include OIR without AAV-Fto or AAV-blank, as well as STZ without other treatments.

Fig1J: It is quite convincing that FTO increases in PDR condition. However, the composition of ERM or proliferative membranes is rather mixed. Endothelial cells account for a very small fraction, if any. Recent scRNA-seq (GSE199317) shows FTO is expressed in a number of cell types in the retina and highly in glia populations. While the authors argue endothelial FTO plays a significant role, Muller glia and astrocyte derived FTO may be equally important and should be discussed. This would argue against the main conclusion from the manuscript, which places that endothelial FTO at the epicenter for the pathologies. Have the authors co-stain FTO with other cell markers for ERM tissues and determine which cell type expresses most FTO in human conditions.

Fig2F, H vs. G, I. Can the authors comment on the huge discrepancy of cell cycle analysis between L-EV treated, vs. Scramble siRNA groups? Both should be the control groups. This would imply the cells in use could have significant batch effects or procedure related effects. It is also very important to include a naive group meaning no lentivirus, no siRNA treatment at all and analyze the cell cycle, migration and proliferation. Alternatively, the authors could consider synchronizing the cells and then determining whether FTO overexpression or knockdown affects the cell cycle entry.

Have the authors examined cell death upon lentivirus or siRNA treatment? Cleaved caspase 3 staining for example?

Fig2J, tip cell markers. COL4A1 and VEGFA are less well-acknowledged tip cell markers. Cxcr4, Fscn1, Apln, Esm1, Angpt2 and Plaur are more appropriate and should be assessed here and in the following experiments.

Fig3 OIR experiment.

As mentioned above, the authors lacked a proper control, which is just OIR without AAV injection. The control panel shown herein is control mice that did not undergo OIR procedure. Therefore it is not justified to compare NVT area, avascular area to samples that were not treated the same and did not have the pathology. As such, Fig3D left panel showing control EC without tip cell, for example, is not appropriate.

Can the authors comment on how FTO plays a role in physiological angiogenesis? The retina is a great model for studying developmental angiogenesis. One possible experiment is to examine developmental angiogenesis in early postnatal retinas by FTO staining. This would inform if they are preferentially expressed in tip cells or stalk cells or in other cell types. And then by a single dose of injection of AAV-Fto vs. AAV-blank from P1, the authors could determine how Fto overexpression affects developmental retinal angiogenesis.

Similar to what the authors have shown in vitro, the retinal flatmount samples could be utilized for assessing proliferation (EdU+ or Ki67+ cells), tip cell markers (Cxcr4, Apln, ESM1, Angpt2, Plaur), and mural cell markers (NG2, PDGFRb) to answer whether EC-pericyte crosstalk is affected by AAV-FTO.

Related Fig S3B. Please comment on the lack of Flag band in AAV-Blank, which clearly contains Flag tag based on the design. Related Fig S3C. Can the authors stain FTO instead of Flag?

Fig3 zebrafish experiment: The vascular development in zebrafish is stereotyped and has been well characterized, this is a good anchoring point to assess if and how FTO impacts developmental angiogenesis.

For ocular vasculature in zebrafish, please refer to Kaufman R et al., 2016, BMC Dev Biol. In this manuscript, the authors solely used relative density as a measure. However, from the images presented herein, the vessels throughout the body, not limited to the eye, seem to be affected in the Fto overexpression group. The trunk and head vessels, intersomitic vessels, all seem affected, and appear dilated. So is overexpression of Fto detrimental in physiological angiogenesis? The eye structure in the Fto mRNA + 3% glucose group seems to be severely affected.

I think this experiment brings up more questions than the answers the authors hoped for.

Fig4. AAV experiment in STZ mice. Please provide a reason or supporting data for why two doses of intravitreal AAV-Fto are needed. As gene therapy, one treatment should provide long lasting expression. Repeated intravitreal injection in mice is doable but may increase the possibility of infection and subclinical microglia activation which may cloud conclusions.

It was unclear that the cotton-wool findings are associated with injection per se or treatments.

Similar to the OIR experiment, a very important STZ alone (no AAV-blank or AAV-Fto) control is missing. Without that control, it is difficult to assess overexpression over procedural-related effects.

What is the molecular weight for Texas Red Dextran used in this study?

Judging vascular permeability on these low power images is very qualitative. A quantitative assay to determine permeability is needed. General steps include injecting the dye, perfusing to remove extra dye from circulation, then processing the retinas either to quantitatively measure the fluorescence, or image to locate areas where the dye has leaked out to stain underlying neural tissues.

VE-Cadherin is a marker for adherens junction. At this resolution, it is impossible to tell if there is any break or de facto discontinuity. The reduction of VE-cadherin intensity could be due to injection procedures or sampling bias. The IB4 staining is of low quality. I suggest replacing IBA1 with PECAM1 for co-staining with VE-Cadherin and image at much higher resolution to look at the junctions. Ideally, one ought to do HRP perfusion and transmission electron microscopy to look at the integrity of junctions. Furthermore, in the retinas and brains, tight junctions are more relevant for barrier function. The authors should carefully examine tight junction proteins such as claudin 5, ZO-1 and occludin, which are known to be affected by diabetes. Please refer to work from Antonetti lab.

Counting the number of pericytes based on NG2 staining is not recommended. Pericytes in the central nervous system cover extensive length on endothelial cells. The overall intensity appears to be similar, which would imply the pericyte coverage is largely fine. No singular pericyte marker exists. NG2 expression could be downregulated when pericyte numbers are normal. Therefore, it is important to include other pericyte markers such as PDGFRb to determine whether there is an effect on pericyte survival after Fto overexpression.

Trypsin digestion of retinas followed by PAS staining to visualize acellular capillaries could be further corroborated or replaced by immunostaining of Collagen IV, which also visualizes acellular capillaries and avoids the harsh treatment of trypsin prep. IBA1 indeed stains macrophages or microglia in the retina. Again, without STZ alone, it remains unclear whether the macrophages or microglia increase is caused by the injection procedures or actual treatments.

Fig5E. Can the authors elaborate on how they achieve quantifications in Fig5E? The representative images seem to have sampling biases. IBA1 is more of a pan marker for microglia than an activated microglia marker.

How is VEGFA affected by Fto overexpression? As shown recently by Ambati lab (Wang et al., 2023, STTT), FTO induces VEGFA expression that attracts macrophages to the lesions. This very relevant reference is not cited in this manuscript. Nor VEGFA is carefully examined. If FTO overexpression induces VEGFA expression, that may explain the increased permeability,

microglia changes, etc.

Fig6. FTO expression and neuroinflammation

Again there lacks a very important control group which is STZ by itself.

Tmem119 is not nuclear staining. Please confirm the antibody. If no reliable antibody exists, an alternative approach would be to examine the mRNA expression level of Tmem119 and other activated microglia genes including Gpnmb, Lgals3, Trem2, Cst7, CD68, CD11c, etc by qPCR.

Please provide a detailed description of how RGC count based on Tubb3/Tuj1 staining is achieved.

Please comment on the discrepancies between Fig7C and Fig2F.

In Fig8H and 8J, the presented blots showed lower Gapdh and increased CDK2. This would skew the normalization. As Gapdh is often affected by cell metabolism, can the authors use a different housekeeping protein for normalization?

In Fig1E, the RNA-seq also finds an increase in CDK1 in L-FTO groups. Aleem et al., (2005) Nat Cell Biol originally showed that CDK1 compensates for CDK2 loss to regulate cell cycle entry. Have the authors examined CDK1 at protein level in their systems?

Fig9. Lactate regulates FTO in vitro. This set of experiments is solely based on the HUVEC cells. Is it also translatable to in vivo? Have the authors examined the STZ retinas for lactate level? In the retina, Müller glia produces large amounts of lactate which is then metabolized by photoreceptors. It may be relevant to determine whether in STZ retinas there is an increase of lactate.

Fig9A. What is the actual concentration of lactate?

C646 is used at 100uM. This concentration is incredibly high for a small molecule. Have the authors varied concentrations? Or does that pose cellular toxicity and affect cell survival?

Fig11. Therapeutic possibility of FB23-2.

Instead of PBS, a more proper control would be a nanoparticle with M-vesicle, Dil and RGD peptide.

Have the authors characterized the PK profile of NP-FB23-2? Is there a basis for daily injection?

What is the molecular weight of NP-FB23-2? Can it cross the blood-retinal barrier or the blood-brain barrier?

Shown in Fig11, OIR+NP-FB23-2 moderately but significantly reduced the NVT area. The error bar is outstanding. Please provide individual values to show data distribution. Did it also rescue the permeability, pericyte dropout and neuroinflammation?

Other minor comments:

Line 477. TEM is transmission electron microscopy, not transition.

Line 693. Continuous i.p. Injection? Do the authors mean daily?

Referee #2 (Comments on Novelty/Model System for Author):

This is an excellent paper. However, FTO therapy to modify mRNA methylation may have a broad impact on mRNA metabolism and protein translation. Therefore, safety is a potential concern and needs to be thoroughly investigated.

Referee #2 (Remarks for Author):

This study by Chen et al. sheds new insight into FTO's role in DR via demethylation of m6A that can affect ECs, EC-pericytes, and EC-microglia interactions. Also, a selective inhibitor of FTO, FB23-2 has been studied as a potential therapeutic agent against DR. Overall, the manuscript has a good flow with ample mechanistic data, and each conclusion was drawn after extensive validations to provide novel insights into the molecular mechanism of FTO involvement in PDR.

Here are some comments:

1. The authors showed that high glycemic conditions lead to reduced m6A levels and increased FTO levels. And overexpression of FTO leads to reduced m6A levels. For this experiment, HUVECs were grown under high glucose (25 mM) in vitro. A previous

study has shown that high glucose can inhibit HUVEC growth (PMID: 8932990). So, it would be helpful to compare the HUVEC viability in NG vs HG conditions and normalize the m6A methylation and other enzyme expressions accordingly.

2. The relative mRNA expression of METTL3 is increased in HG compared to NG but lowered in STZ compared to ctrl. But the METTL3 protein levels are unchanged in all conditions. It would be helpful to explain this observation as well.

3. It showed that FTO promotes endothelial cell cycle progression using independent techniques, including RNA-seq. It is unclear from the main text or from the methods section which RNA inputs (FTO overexpressed vs FTO KD or FTO overexpressed vs. ctrl) were used for RNA-seq analysis.

4. Tip cell markers are upregulated in L-FTO-treated HUVECs. In addition, it would be helpful to see tip cell markers' expression in FTO KD.

5. In the OIR experiment, AAV-FTO was injected at P12 and retinas were examined at P17. Five days might be too soon to assess the AAV-FTO expression in vivo. AAV in vivo takes at least 1-2 weeks for transgene to be expressed and reach to peak expression in ~4 weeks. Also in Fig S3, the timeline of examining AAV-Fto expression in mouse retinal vascular EC was not mentioned. Were neonatal or adult mice used in Fig. S3?

6. Line 208 "Collectively, our data suggested that FTO simultaneously suppressed healthy vascular network formation into the ischemic retina..". It is unclear how this conclusion was drawn.

7. Authors used extensive validation to show that FTO regulates EC-pericyte, vascular inflammation, EC-microglia crosstalk, and neurodegeneration in vitro and in vivo. FFA, Texas red dextran and Evans blue assay showed increased leakage in STZ + AAV-FTO. Moreover, VE-cadherin and NG2 staining show increased discontinuity and reduced pericyte coverage of STZ + AAV-FTO retina, respectively. An additional comparison of ctrl vs STZ w/o AAV-blank or AAV-FTO for retinal leakage and pericyte coverage may help rule out any AAV-related effects.

8. Line 288 - "revealed that percentage of activated microglia, represented by reduced number of interactions of microglia". It is unclear how an increase in activated microglia correlates to reduced interactions.

9. The incidence of 586 m6A-hyper peaks upon FTO overexpression should be explained. It seems overall m6A is increased globally (465 hypo vs 586 hyper m6A) in MeRIP-Seq, but the dot blot assay of global m6A abundance shows a decrease upon FTO overexpression. This contradiction should be discussed.

10. Other identified proteins including FDFT1 also involved in the cell cycle. So, it would be helpful to provide more insights into the rationale for selecting CDK2 for further studies.

11. In Figure 8L, mutant FTO can be used as a control in RIP-qPCR assay.

12. Line 408 "Actinomycin D was used to inhibit CDK2 transcription in HUVECs". It is unclear how Actinomycin D specifically inhibits CDK2 transcription.

13. In a previous study (PMID: 28400392), ECs isolated from fibrovascular membranes of PDR patients and healthy retinas were profiled by RNA-seq. Not sure whether FTO was upregulated in this study. The RNA-seq data available in NCBI should be mined and discussed.

Minor points

14. Minor details: In Fig S1B, No GFP signal. Only DAPI. Should GFP be changed to DAPI?

15. In Fig. S3 legend title, "mice retinal vascular" should be changed to "mouse".

16. "Proliferative membranes obtained from DR patients". If patients had PDR, "PDR patients" should be used for clarity.

17. FTO should be defined.

18. In Fig 6A, the reason for measuring Il1b and Ccl2 expression levels is not clearly stated.

Referee #1 (Comments on Novelty/Model System for Author):

Lacking important controls especially undermines the in vivo experiments. The zebrafish experiment is appropriate, however is not well executed and may bring up more questions than answers. The OIR experiment and the STZ experiment lacks the naive group (OIR only, or STZ only).

Referee #1 (Remarks for Author):

Chen et al. presented a well-crafted, and data-enriched manuscript to investigate if and how FTO, an eraser gene in m6A modification, is involved in vascular changes associated with experimental diabetic retinopathy. To this end, they integrated multiple lines of in vitro and in vivo experiments to back up their claims. Major achievements include the identification of upstream regulators and downstream effectors of FTO, whose expression is inversely correlated with diabetic vascular conditions. In addition, the authors engineered an enabling platform to deliver an FTO inhibitor systemically to treat diabetic vascular complications. Overall, this paper could bring additional insights of FTO to vascular biology, and highlight its druggability in managing diabetic retinopathy. However, several key issues need to be addressed before it can be recommended for publications.

The authors utilized HUVEC extensively and extended the in vitro findings to in vivo experiments, which were less compelling and would require additional important controls. As a minor comment, I advise the authors to organize strong in vivo data first and use in vitro experiments secondarily for MOA investigation and phenotype confirmation. Below are my specific questions and comments.

For all the bar graphs, I suggest the authors plot individual data to allow

clear visualization of data distribution and variability.

We think the reviewer has raised a very reasonable concern. As suggested by the reviewer, we have modified all bar graphs in all Figures and Supplementary materials. We hope you will find it appropriate now.

Lack of controls:

For in vitro experiments, it is very important to include a naive group, meaning no lentivirus, OR no siRNA treatment.

For in vivo experiments, it is very important to include OIR without AAV-Fto or AAV-blank, as well as STZ without other treatments.

Thank you for your comment. The second reviewer also raised a similar concern. As suggested by both reviewers, we have included control groups in both *in vitro* and *in vivo* experiments. We hope you will find it appropriate now.

Fig1J: It is quite convincing that FTO increases in PDR condition. However, the composition of ERM or proliferative membranes is rather mixed. Endothelial cells account for a very small fraction, if any. Recent scRNA-seq (GSE199317) shows FTO is expressed in a number of cell types in the retina and highly in glia populations. While the authors argue endothelial FTO plays a significant role, Muller glia and astrocyte derived FTO may be equally important and should be discussed. This would argue against the main conclusion from the manuscript, which places that endothelial FTO at the epicenter for the pathologies. Have the authors co-stain FTO with other cell markers for ERM tissues and determine which cell type expresses most FTO in human conditions.

We think the reviewer has raised a very important concern. We agree with the reviewer that FTO dysregulation in other retinal cells, including glia cells, pericytes and neural cells, could also contribute to diabetic retinopathy. We therefore have included the following statement in the third paragraph of the **Discussion** section:

“However, due to its ubiquitous expression, FTO dysregulation in other retinal cells, including glia cells, pericytes and neural cells, could also contribute to DR. Other potential downstream targets and pathways of FTO in ECs may also exist. The pathogenic role and regulating network of FTO in DR need to be explored systematically in further studies.”

We also agree with the reviewer that the composition of ERMs or FVMs is rather mixed and endothelial cells account for a very small fraction. We have co-stained FTO with cell markers for vascular endothelial cells and microglia in both ERM and PDR tissues. However, the staining results were quite confusing due to the mixed structure, based on which we were not able to tell which cell type expresses most FTO. Therefore, to better reveal the role of FTO in diabetic retinopathy, we analyzed and compared FTO’s expression in different types of retinal cells using single cell RNA-Seq data obtained from STZ rats retina (GEO datasets: GSE209872). Single cell RNA-Seq data revealed that FTO was ubiquitously expressed in all types of retinal cells (**Figure X1A**). Further assessments revealed that differential expression of FTO between STZ and control rats is solely detected in retinal vascular endothelial cells (**Figure X1B**).

Figure X1

Figure X1. (A) Feature plot showing FTO expression in different types of retinal cells (GEO accession number: GSE209872). **(B)** Violinplots comparing FTO expression in different types of retinal cells between the STZ and control mice (GEO accession number: GSE209872).

Additionally, vascular endothelial cell is the mostly studied type of retinal cells in diabetic retinopathy and retinal neovascularization. Endothelial FTO has previously been identified as a regulator of obesity-induced metabolic and

vascular changes. Loss of endothelial FTO antagonizes obesity-induced metabolic and vascular dysfunction (Krüger et al, *Circ Res*, PMID: 31801409), indicating the crucial role of FTO in vascular endothelial cells. Collectively, in the present study, we studied the role of endothelial FTO dysregulation in DR. We hope the reviewer will find it acceptable.

Fig2F, H vs. G, I. Can the authors comment on the huge discrepancy of cell cycle analysis between L-EV treated, vs. Scramble siRNA groups? Both should be the control groups. This would imply the cells in use could have significant batch effects or procedure related effects. It is also very important to include a naive group meaning no lentivirus, no siRNA treatment at all and analyze the cell cycle, migration and proliferation. Alternatively, the authors could consider synchronizing the cells and then determining whether FTO overexpression or knockdown affects the cell cycle entry.

Have the authors examined cell death upon lentivirus or siRNA treatment? Cleaved caspase 3 staining for example?

Thanks for your comment. As suggested by the reviewer, we have included the no lentivirus and no siRNA treatment group in **Figures 2F, G, H, I, J, K, N, O, P, Q, R, S, T and U**. We also synchronized the cells and re-conducted the cell cycle analyses. The cell cycle results in the current **Figures 2F and 2G** were in consistent with each other.

Additionally, we used immunoblotting of cleaved caspase-3 to test whether FTO affects cell death of HUVECs. Our data revealed the suppressive role of FTO in HUVEC death (**Figures 2J-2K**). We have included the following statement in the **Results** section under the subtitle of “***FTO promotes endothelial cell cycle progression and tip cell formation to facilitate angiogenesis in vitro***”:

“Furthermore, immunoblotting detected that expression of cleaved caspase-3 is down-regulated in HUVECs transduced with L-FTO (**Figure 2J**)

and is up-regulated in cells transfected with FTO-siRNA (**Figure 2K**), indicating the suppressive role of FTO in HUVEC death.”

We hope the reviewer will find it appropriate now.

Fig2J, tip cell markers. COL4A1 and VEGFA are less well-acknowledged tip cell markers. Cxcr4, Fscn1, Apln, Esm1, Angpt2 and Plaur are more appropriate and should be assessed here and in the following experiments.

Thank you for your comment. We have included the qPCR results of 5 tip cell markers suggested by the reviewer, including *CXCR4*, *FSCN1*, *APLN*, *ESM1* and *PLAUR* in **Figures 2L-2M**. The expression of *ANGPT2* was not detectable in HUVECs with its CT value over 35. Consistently, qPCR detected increased expression of all 5 genes in HUVECs transduced with L-FTO (**Figure 2L**) and decreased expression in cells transfected with FTO-siRNA (**Figure 2M**). We have included the following statement in the **Results** section under the subtitle of “***FTO promotes endothelial cell cycle progression and tip cell formation to facilitate angiogenesis in vitro***”:

“Consistently, expression of tip cell markers, including *CXCR4*, *FSCN1*, *APLN*, *ESM1* and *PLAUR*, was elevated in HUVECs overexpressing FTO (**Figure 2L**), and was decreased in cells with FTO knocked down (**Figure 2M**), suggesting that FTO promotes endothelial tip cell formation.”

We hope the reviewer will find it appropriate now.

Fig3 OIR experiment.

As mentioned above, the authors lacked a proper control, which is just OIR without AAV injection. The control panel shown herein is control mice that did not undergo OIR procedure. Therefore it is not justified to compare NVT area, avascular area to samples that were not treated the same and did not have the pathology. As such, Fig3D left panel showing control EC without tip cell, for example, is not appropriate.

Thank you for your comment. We think the reviewer has raised a very important concern. As suggested by the reviewer, we have included the “OIR without AAV injection” group in **Figures 3B-3G**. We have modified the relevant **Results** section as follows:

“More extensive areas of neovascular tufts (NVTs), formed in the superficial vascular plexuses, were observed in the OIR mice receiving AAV-Fto injection compared to the OIR mice without injection or the OIR mice injected with AAV-Fto (**Figure 3B**). *In vivo* EdU assay also revealed enhanced proliferation of vascular cells in neural retinas collected from the OIR mice receiving AAV-Fto injection (**Figure 3C**). Moreover, we noticed that, in the angiogenic area, *Fto* overexpression leads to increased amount of endothelial tip cells (**Figure 3D**). Consistently, qPCR demonstrated increased expression of tip cell markers *Apln* and *Esm1* in neural retinas collected from the OIR mice injected with AAV-Fto (**Figures 3E-3F**). However, no difference in avascular area in the central retina and vessel density in the peripheral deep plexuses was detected among the OIR mice without injection, injected with AAV-Fto and injected with AAV-blank (**Figure 3B**). No NVT or avascular area was observed in the control group (**Figure 3B**). Pericyte loss in retinal capillaries can induce BRB destruction (22, 23). We next used immunofluorescence staining of platelet derived growth factor receptor β (PDGFR β), a pericyte marker, to annotate FTO’s function in regulating EC-pericyte crosstalk. Reduced pericyte coverage was detected in NVTs of OIR mice overexpressing Fto (**Figure 3G**), indicating FTO’s role in interrupting EC-pericyte crosstalk.”

We hope the reviewer will find it appropriate now.

Can the authors comment on how FTO plays a role in physiological angiogenesis? The retina is a great model for studying developmental angiogenesis. One possible experiment is to examine developmental angiogenesis in early postnatal retinas by FTO staining. This would inform if they are preferentially expressed in tip cells or stalk cells or in

other cell types. And then by a single dose of injection of AAV-Fto vs. AAV-blank from P1, the authors could determine how Fto overexpression affects developmental retinal angiogenesis.

Thank you for your comment. The present study focused on role of FTO in pathogenic angiogenesis instead of physiological angiogenesis. We agree with the reviewer that “*The zebrafish experiment brings up more questions than the answers the authors hoped for.*” Therefore, as suggested by the reviewer, we have removed the zebrafish experiments on physiological angiogenesis, and decided not to discuss FTO’s role in developmental retinal angiogenesis. However, we believe the reviewer has raised an important concern. We will keep working on FTO’s role in developmental retinal angiogenesis in our further works.

To further tell whether FTO is preferentially expressed in tip cells or stalk cells or in other cell types, we analyzed FTO expression in different types of vascular endothelial cells using single cell RNA-Seq data obtained from P6 mice retina (GEO dataset: GSE169039). Seurat UMAP dimensionality reduction and clustering algorithms identified 9 populations of vascular cells (**Figure X2A**). As indicated by the single cell RNA-Seq data, FTO was mostly enriched in tip cells (**Figure X2B**).

Figure X2

Figure X2B. FTO expression in different types of vascular cell. (A) Seurat UMAP dimensionality reduction and clustering algorithms identified 9 populations of vascular cells (GEO accession number: GSE169039). **(B)**

Comparison of FTO expression in different types of vascular cells (GEO accession number: GSE169039).

We hope the reviewer will find it appropriate now.

Similar to what the authors have shown in vitro, the retinal flatmount samples could be utilized for assessing proliferation (EdU+ or Ki67+ cells), tip cell markers (Cxcr4, Apln, ESM1, Angpt2, Plaur), and mural cell markers (NG2, PDGFRb) to answer whether EC-pericyte crosstalk is affected by AAV-FTO.

Thanks for your comment. As suggested by the reviewer, we used *in vivo* EdU assay to analyze proliferation of vascular cells. Herein, we detected enhanced proliferation of vascular cells in neural retinas collected from the OIR mice receiving AAV-Fto injection compared to the OIR mice without injection or the OIR mice injected with AAV-blank (**Figure 3C**).

qPCR demonstrated increased expression of tip cell markers *Apln* and *Esm1* in neural retinas collected from the OIR mice injected with AAV-Fto compared to the OIR mice without injection or the OIR mice injected with AAV-blank (**Figures 3E-3F**).

We next used immunofluorescence staining of PDGFR β , a pericyte marker, to annotate FTO's function in regulating EC-pericyte crosstalk. Reduced pericyte coverage was detected in NVTs of OIR mice overexpressing Fto (**Figure 3G**), indicating FTO's role in interrupting EC-pericyte crosstalk. We have included the following statements in the **Results** section under the subtitle of "***FTO promotes endothelial tip cell formation to facilitate neovascularization in mice and zebrafish***":

- (1) "*In vivo* EdU assay also revealed enhanced proliferation of vascular cells in neural retinas collected from the OIR mice receiving AAV-Fto injection (**Figure 3C**)."
- (2) "Consistently, qPCR demonstrated increased expression of tip cell markers *Apln* and *Esm1* in neural retinas collected from the OIR mice injected with

AAV-Fto (**Figures 3E-3F**).”

(3) “We next used immunofluorescence staining of platelet derived growth factor receptor β (PDGFR β), a pericyte marker, to annotate FTO’s function in regulating EC-pericyte crosstalk. Reduced pericyte coverage was detected in NVTs of OIR mice overexpressing Fto (**Figure 3G**), indicating FTO’s role in interrupting EC-pericyte crosstalk.”

We hope the reviewer will find it appropriate now.

Related Fig S3B. Please comment on the lack of Flag band in AAV-Blank, which clearly contains Flag tag based on the design.

Thank you for your comment. As suggested by the reviewer, we have tried to show the Flag band using immunoblotting assay for three additional times with newly collected samples. However, the Flag band was still not detectable. A potential explanation was that the Flag polypeptide (2 kDa) is degraded during the ultrasonication of the retinal tissue. Alternatively, we used immunofluorescence staining to show the expression pattern of Flag in retina (**Supplementary Figure S4G**). We hope the reviewer will find it acceptable.

Related Fig S3C. Can the authors stain FTO instead of Flag?

Thanks for your comment. As indicated by the reviewer, we used immunofluorescence staining to show the FTO expression in the isolated mice retinal vasculatures. Consistent with the immunoblotting data, we detected increased Fto expression in the isolated retinal vasculatures of mice receiving intra-vitreous AAV-Fto injection compared to mice injected with AAV-blank (**Supplementary Figure S4E**). We hope the reviewer will find it acceptable.

Fig3 zebrafish experiment: The vascular development in zebrafish is stereotyped and has been well characterized, this is a good anchoring point to assess if and how FTO impacts developmental angiogenesis. For ocular vasculature in zebrafish, please refer to Kaufman R et al., 2016,

BMC Dev Biol. In this manuscript, the authors solely used relative density as a measure. However, from the images presented herein, the vessels throughout the body, not limited to the eye, seem to be affected in the Fto overexpression group. The trunk and head vessels, intersomitic vessels, all seem affected, and appear dilated. So is overexpression of Fto detrimental in physiological angiogenesis? The eye structure in the Fto mRNA + 3% glucose group seems to be severely affected.

I think this experiment brings up more questions than the answers the authors hoped for.

Thank you for your comment. As suggested by the reviewer, we have measured and compared the diameters of zebrafish retinal vessels. Fluorescence staining detected dilated retinal vessels in zebrafish injected with *fto* mRNA and maintained in 3% glucose water compared to the uninjected zebrafish in 3% glucose water and the control group (**Figure 3J**).

The vessels throughout the body were affected because we injected *fto* mRNA at a rather high concentration of 360 ng/μL, which caused systemic toxicity to zebrafish. Herein, zebrafish embryos injected with *fto* mRNA at the concentration of 120 ng/μL showed no remarkable systemic changes, and were collected and examined at 48 hours post fertilization (**Figures 3H-3J**). In the current **Figure 3I**, we have removed the 360 ng/μL data to avoid confusion. We have also excluded zebrafish with abnormal eye structures in the *fto* mRNA + 3% glucose group for calculation of retinal vascular intensity and diameters.

We agree with the reviewer that “*The zebrafish experiment brings up more questions than the answers the authors hoped for.*” The present study focused on role of FTO in pathogenic angiogenesis instead of physiological angiogenesis. Therefore, as suggested by the reviewer, we have removed the zebrafish experiments on physiological angiogenesis, and decided not to discuss FTO’s role in developmental retinal angiogenesis.

However, we believe the reviewer has raised an important concern. We will keep working on FTO's role in developmental retinal angiogenesis in our further works. We hope the reviewer will find it appropriate now.

Fig4. AAV experiment in STZ mice. Please provide a reason or supporting data for why two doses of intravitreal AAV-Fto are needed. As gene therapy, one treatment should provide long lasting expression. Repeated intravitreal injection in mice is doable but may increase the possibility of infection and subclinical microglia activation which may cloud conclusions.

It was unclear that the cotton-wool findings are associated with injection per se or treatments.

Thank you for your comment. Based on our preliminary experiment, repeated injection of AAV-Fto will maximize virus delivery and enhance Fto expression in mice retina. Consistently, repeated injections of AAV into the vitreous (once a month; Shan et al, *Circulation*, PMID: 28860123) or through the retro-orbital sinus (at P7 and P12; Chen et al, *The Journal of Clinical Investigations*, PMID: 33586674) have been previously reported. We hope the reviewer will find our explanation acceptable.

The cotton-wool findings are located in the posterior pole of the retina. Intra-vitreous injections were conducted through an incision into the sclera (1 mm posterior of the superior limbus) under microscopic to ensure that drugs were delivered into the vitreous chamber without injuring the retina. To avoid misunderstanding, we have detailed the procedure for intra-vitreous injections in the **Materials and Methods** section under the subtitle of "**Mouse breeding and manipulations**":

"For intra-vitreous injection, 1 μ L solution containing AAV-blank/AAV-Fto supernatant (10^{12} ~ 10^{13} genome copies/mL; AAV Serotype 2) was delivered into the vitreous chamber of mice through an incision into the sclera (1 mm posterior of the superior limbus) under microscopic control using a syringe with

a 33-gauge needle (Hamilton, Bonaduz, Switzerland).”

Collectively, we thus think the cotton-wool findings are more caused by FTO overexpression than injection per se or treatments. We hope the reviewer will find it reasonable.

Similar to the OIR experiment, a very important STZ alone (no AAV-blank or AAV-Fto) control is missing. Without that control, it is difficult to assess overexpression over procedural-related effects.

Thank you for your comment. As suggested by the reviewer, we have included an STZ alone control in all experiments in **Figures 4 and 6**. We hope you find it appropriate now.

What is the molecular weight for Texas Red Dextran used in this study?

The molecular weight for Texas Red Dextran used in this study is 70,000. We have included the following statement in the **Materials and Methods** section under the subtitle of “**Mouse breeding and manipulations**”:

“One mL Texas red dextran solution (70,000 MW; 2 mg/mL; Invitrogen) was injected into the left ventricle at an even speed for 1 minute.”

Judging vascular permeability on these low power images is very qualitative. A quantitative assay to determine permeability is needed. General steps include injecting the dye, perfusing to remove extra dye from circulation, then processing the retinas either to quantitatively measure the fluorescence, or image to locate areas where the dye has leaked out to stain underlying neural tissues.

Thank you for your comment. According to the reviewer’s suggestions, we have tried to conduct the quantitative assay to determine the permeability of retinal vessels. However, we failed the experiment due to technical difficulties and lack of samples. We have tried to use neural retinas collected from 10 mice per well, while the OD value was still undetectable. We have emailed

other groups that have successfully conducted the experiment for technical support. They told us that they get credible data by using 30 retinas per well. That will be 90 retinas per group (technical triplicate), and 360 retinas for all four groups. Due to the strict time limit of the journal, currently we do not have sufficient time and samples to complete this experiment. However, we believe the reviewer has raised an important concern. We will keep working on that in our further work. We hope the reviewer will understand our plight.

Alternatively, to better visualize the Evens blue leakage in neural retina, we have included the 3D reconstruction images in **Figure 4E**. We have additionally analyzed the extravascular leakage of Evens blue, including both overall intensity and fluorescence along with the retinal vasculature in **Figure 4E**.

Many previous studies judge vascular leakage using a single experiment (Hye-Yoon Jeon et al, *BMC Med*, PMID: 36782199; Yeon-Ju Lee et al, *Diabetes*, PMID: 27207524). In the present study, we used fluorescence fundus angiography, Texas red dextran and Evans blue assays to explore FTO's role on retinal vascular leakage. All three experiments consistently revealed that FTO overexpression aggravated diabetes-induced retinal vascular leakage (**Figures 4C-4E**). Again, thank you so much for your suggestion, which will indeed improve our work. We will keep working on that in our further work, and we hope you will understand our plight.

VE-Cadherin is a marker for adherens junction. At this resolution, it is impossible to tell if there is any break or de facto discontinuity. The reduction of VE-cadherin intensity could be due to injection procedures or sampling bias. The IB4 staining is of low quality. I suggest replacing IBA1 with PECAM1 for co-staining with VE-Cadherin and image at much higher resolution to look at the junctions. Ideally, one ought to do HRP perfusion and transmission electron microscopy to look at the integrity of junctions.

Furthermore, in the retinas and brains, tight junctions are more relevant for barrier function. The authors should carefully examine tight junction proteins such as claudin 5, ZO-1 and occludin, which are known to be affected by diabetes. Please refer to work from Antonetti lab.

Thanks for your comment. As suggested by the reviewer, we re-conducted the staining of VE-cadherin and IB4 and imaged at higher resolution. Consistent with our previous findings, the coverage of VE-cadherin to IB4 was found decreased in STZ mice receiving AAV-Fto injection compared to the other groups (**Figure 4F**).

Additionally, we conducted immunoblotting to detect the expression of ZO-1, which forms a confluent tight junction at the vascular endothelial cell membrane, in neural retinas collected from mice receiving distinct treatments. Our data revealed that expression of ZO-1 is reduced in neural retinas collected from STZ mice injected with AAV-Fto compared to the other groups (**Figure 4G**). We hope you will find it appropriate now.

Counting the number of pericytes based on NG2 staining is not recommended. Pericytes in the central nervous system cover extensive length on endothelial cells. The overall intensity appears to be similar, which would imply the pericyte coverage is largely fine. No singular pericyte marker exists. NG2 expression could be downregulated when pericyte numbers are normal. Therefore, it is important to include other pericyte markers such as PDGFRb to determine whether there is an effect on pericyte survival after Fto overexpression.

Thanks for your comment. As indicated by the reviewer, we reanalyzed the coverage of NG2 to IB4 positive areas. NG2 coverage was found down-regulated in retinal flat mounts of STZ mice injected with AAV-Fto compared to control, STZ mice without injection, and STZ mice injected with AAV-blank (**Figure 4H**). We further used PDGFR β and IB4 immunofluorescence staining to detect pericyte coverage of retinal vessels.

Consistent with the NG2 data, PDGFR β staining demonstrated reduced pericyte coverage of retinal vessels upon the combination of FTO overexpression and diabetes (**Figure 4I**). We have included the following statement in the **Results** section under the subtitle of “***FTO regulates EC-pericyte crosstalk and triggers diabetes-induced microvascular dysfunction in mice***”:

“We further used NG2/PDGFR β (pericyte markers) and IB4 immunofluorescence staining to detect pericyte coverage of retinal vessels. Consistently, both NG2 and PDGFR β staining demonstrated reduced pericyte coverage of retinal vessels upon the combination of FTO overexpression and diabetes (**Figures 4H-4I**).”

We hope you will find it appropriate now.

Trypsin digestion of retinas followed by PAS staining to visualize acellular capillaries could be further corroborated or replaced by immunostaining of Collagen IV, which also visualizes acellular capillaries and avoids the harsh treatment of trypsin prep.

We do appreciate the reviewer for this very helpful comment. As suggested by the reviewer, we have ordered two brands of antibodies against collagen IV (abcam256353 and Bio-Rad 134001) for immunofluorescence staining, and tried various dilutions. However, neither antibody worked in our system. Due to the strict time limit of the journal, currently we do not have sufficient time to try more antibodies. BTW, many previous studies visualize acellular capillaries using Trypsin digestion and PAS staining without immunofluorescence staining of collagen IV (Shan et al, *Circulation*, PMID: 28860123; Li et al, *eBioMedicine*, PMID: 35172268; Shi et al, *Acta Neuropathol Commun*, PMID: 33228786; Shanab et al, *Mol Ther Methods Clin Dev*, PMID: 26029724). Again, thank you so much for your suggestion, which will indeed improve our work. We will keep working on that in our further works. We hope the reviewer will understand our plight.

IBA1 indeed stains macrophages or microglia in the retina. Again, without STZ alone, it remains unclear whether the macrophages or microglia increase is caused by the injection procedures or actual treatments.

Thank you so much for your comment. As suggested by the reviewer, we have included the STZ alone group in **Figure 6**. According to our results, the increase of macrophages or microglia is caused by the actual treatments.

We agree with the reviewer that Iba-1 indeed stains macrophages or microglia in the retina. However, microglia are highly specialized tissue macrophages that serve as immune sentinels of the central nervous system (CNS) parenchyma, including the neural retina (Koren et al, *Immunity*, PMID: 30850344). Thus, Iba-1 has been widely used as a marker for microglia in the retina (Millsa et al, *Proc Natl Acad Sci U S A*, PMID: 34903661, Figure 1; Ma et al, *eLife*, PMID: 30666961, Figure 2) and in the CNS (Kang et al, *Nat Commun*, PMID: 36528681, Figure 7; Qureshi et al, *Cell Rep*, PMID: 35045281, Figure 1). We hope you will find it appropriate now.

Fig5E. Can the authors elaborate on how they achieve quantifications in Fig5E? The representative images seem to have sampling biases. IBA1 is more of a pan marker for microglia than an activated microglia marker.

Thanks for your comment. The second reviewer also raised a similar concern. Consistent to our quantifications in **Figures 6B** and **6D**, the activation status of microglia in **Figure 5F** was also determined based on morphological assessments. Microglia showing an amoeboid morphology was defined as activated microglia, which was consistent with previous studies (Figure 3A, Akhter et al, *Mol Immunol*, PMID: 33461764). We have updated the relevant statement in the **Results** section under the subtitle of “***FTO triggers vascular inflammation and regulates EC-microglia crosstalk in vitro***” as follows:

“Immunofluorescence staining of ionized calcium binding adapter molecule 1 (Iba-1), a calcium-binding protein that participates in membrane

ruffling and phagocytosis of activated microglia, revealed that percentage of activated microglia, represented by an amoeboid morphology, was increased upon co-cultivation with HUVECs overexpressing FTO (**Figure 5F**).”

We hope the reviewer will find it appropriate now.

How is VEGFA affected by Fto overexpression? As shown recently by Ambati lab (Wang et al., 2023, STTT), FTO induces VEGFA expression that attracts macrophages to the lesions. This very relevant reference is not cited in this manuscript. Nor VEGFA is carefully examined. If FTO overexpression induces VEGFA expression, that may explain the increased permeability, microglia changes, etc.

Thank you for your comment. As suggested by the reviewer, we have cited the reference by Wang et al in the current manuscript, and analyzed VEGF-A expression in HUVECs overexpressing FTO. ELISA identified up-regulated VEGF-A secretion in HUVECs overexpressing FTO (**Figure 5A**). We have updated the relevant statement in the **Results** section under the subtitle of “***FTO triggers vascular inflammation and regulates EC-microglia crosstalk in vitro***” as follows:

“VEGF-A is a proangiogenic and proinflammatory mediator in DR (28). Inhibition of FTO suppressed VEGF-A release in macrophages and retinal pigment epithelial (RPE) cells, which restrained angiogenesis and macrophage infiltration in choroidal neovascularization (29). Consistent with previous findings, enzyme-linked immunosorbent assay (ELISA) identified promoted VEGF-A secretion in HUVECs overexpressing FTO, implying the potential role of FTO in facilitating angiogenesis and inflammation (**Figure 5A**).”

We hope the reviewer will find it appropriate now.

Fig6. FTO expression and neuroinflammation

Again there lacks a very important control group which is STZ by itself.

Tmem119 is not nuclear staining. Please confirm the antibody. If no reliable antibody exists, an alternative approach would be to examine the mRNA expression level of Tmem119 and other activated microglia genes including Gpnmb, Lgals3, Trem2, Cst7, CD68, CD11c, etc by qPCR. Please provide a detailed description of how RGC count based on Tubb3/Tuj1 staining is achieved.

Thank you so much for your comment. As suggested by the reviewer, we have included the STZ alone group in all experiments in **Figure 6**. Due to the lack of proper antibody, we have replaced the immunofluorescence staining of Tmem119 with qPCR analyses of *Tmem119*, *Trem2*, *Lgals3*, and *Cd11c*. Consistent with the data in **Figures 6B-6H**, decreased mRNA expression of *Tmem119* (resting microglia) and increased mRNA levels of *Trem2*, *Lgals3* and *Cd11c* (activated microglia) were detected in neural retina of STZ mice intra-vitreally injected with AAV-Fto (**Figure 6I**), indicating that endothelial FTO overexpression induces microglia activation.

Additionally, to better describe the counting of RGC, we have included the following statement in the legend to **Figure 6K**:

“Tubb3⁺ cells, recognized by the round RGC bodies, were counted in three random areas within the radius of 0.5 to 1.5 mm from the optic disc, and averaged to estimate the RGC per mm² in four retinal flat mounts per group.”

We hope the reviewer will find it appropriate now.

Please comment on the discrepancies between Fig7C and Fig2F.

Thanks for your comment. As suggested by the reviewer, we have synchronized the cells and re-conducted the cell cycle analyses. The cell cycle results in the current **Figures 2F, 2G** and **7C** were consistent with each other. We hope the reviewer will find it appropriate now.

In Fig8H and 8J, the presented blots showed lower Gapdh and increased CDK2. This would skew the normalization. As Gapdh is often affected by

cell metabolism, can the authors use a different housekeeping protein for normalization?

Thanks for your comment. As indicated by the reviewer, we used β -actin instead of GAPDH for normalization. Consistently, CDK2 was found up-regulated in HUVECs exposed to high glucose (**Figure 8H**) and in neural retinas of STZ mice (**Figure 8J**). We hope the reviewer will find it appropriate now.

In Fig1E, the RNA-seq also finds an increase in CDK1 in L-FTO groups. Aleem et al., (2005) Nat Cell Biol originally showed that CDK1 compensates for CDK2 loss to regulate cell cycle entry. Have the authors examined CDK1 at protein level in their systems?

Thanks for your comment. As indicated by the reviewer, we tested the protein levels of CDK1 in HUVECs transduced with L-FTO or transfected with FTO-siRNA. As revealed by immunoblotting, CDK1 expression was up-regulated in HUVECs overexpressing FTO (**Figure X3A**), which was in consistent with the CDK2 expression (**Figure 8O**). CDK1 expression was down-regulated in HUVECs with FTO knocked down (**Figure X3B**), which was also similar to the CDK2 expression (**Figure 8Q**). Thus, CDK1 did not compensate for CDK2 loss to regulate cell cycle entry in our system, since it was also down-regulated upon knocking down of FTO in HUVECs. We hope the reviewer will find it acceptable.

Figure X3

Figure X3. CDK1 protein expression in HUVECs transduced with L-FTO (**A**) or transfected with FTO-siRNA (**B**) as revealed by immunoblotting.

Fig9. Lactate regulates FTO in vitro. This set of experiments is solely based on the HUVEC cells. Is it also translatable to in vivo? Have the authors examined the STZ retinas for lactate level? In the retina, Müller glia produces large amounts of lactate which is then metabolized by photoreceptors. It may be relevant to determine whether in STZ retinas there is an increase of lactate.

Thanks for your comment. As suggested by the reviewer, we examined the STZ retinas for lactate level. Our data identified increased lactate concentration in STZ retinas compared to the control group. We have included the relevant data in **Figure 9B**. We hope the reviewer will find it appropriate now.

Fig9A. What is the actual concentration of lactate?

Thanks for your comment. As suggested by the reviewer, we have provided the actual concentration of lactate in both **Figures 9A and 9B**. We hope you will find it appropriate now.

C646 is used at 100uM. This concentration is incredibly high for a small molecule. Have the authors varied concentrations? Or does that pose cellular toxicity and affect cell survival?

Thanks for your comment. We have varied the concentration for C646 in HUVECs and used the CCK-8 assay to detect its cellular toxicity. C646 at the concentration of 100 μ M was selected, which sufficiently suppressed *FTO* expression in HUVECs (**Supplementary Figure S5A**) without affecting cell viability (**Supplementary Figures S5B-S5D**). We hope you will find it appropriate now.

Fig11. Therapeutic possibility of FB23-2.

Instead of PBS, a more proper control would be a nanoparticle with M-vesicle, Dil and RGD peptide.

Thank you for your comment. As suggested by the reviewer, we have included the unloaded-NP group (nanoparticles without FB23-2 but with all the rest components, including M-vesicle, PLGA, Dil, RGD, PEG and DSPE) in **Figure 11**. According to our data, no difference in the NVT area, avascular area, and number of tip cells was detected between the PBS and the unloaded-NP injected OIR mice retina (**Figures 11K-11L**). We hope the reviewer will find it appropriate now.

Have the authors characterized the PK profile of NP-FB23-2? Is there a basis for daily injection?

Thank you for your comment. Herein, to provide a basis for frequency of administration, we measured the encapsulated FB23-2 release of NP-FB23-2 instead of its PK profile. We uncovered that ~ 50% of encapsulated FB23-2 is released from NP-FB23-2 after being incubated in PBS (pH 7.4) at 37 °C for 24 hours, while only ~6% of FB23-2 is released within the next 24 hours, providing a basis for daily injection (**Figure 11E**). We hope the reviewer find it acceptable.

What is the molecular weight of NP-FB23-2? Can it cross the blood-retinal barrier or the blood-brain barrier?

Thanks for your comment. We have detected enriched fluorescent signal of NP-FB23-2s and unloaded-NPs in retinal vessels of OIR mice (**Figure 11I**). No fluorescent signal was detected in OIR mice intravenously injected with PBS (**Figure 11I**). Our data demonstrated the *in vivo* retinal neovasculature-targeting capacity of NP-FB23-2. The size and internal molecule ingredients of each nanoparticle were not uniform, thus it's difficult to calculate the exact molecular weight of our nanoparticle. However, in the present study, we used a 10 kDa molecule weight cut off dialysis bag and filter tube to restrict the upper limit molecule weight of the nanoparticles. In studies by Li et al (Adv Sci, PMID: 34436822), Thamphiwatana et al (*Proc Natl Acad*

Sci U S A, PMID: 29073076) and Xia et al (*Acta Pharm Sin B*, PMID: 35646523), they also synthesized similar nanoparticles without measuring their molecular weights.

We think whether NP-FB23-2 can cross the blood-retinal barrier or the blood-brain barrier depends more on its diameter. Dynamic light scattering (DLS) measurements demonstrated that NP-FB23-2s show narrow size distributions with the average diameter of 201.71 ± 22.34 nm (**Figure 11C**). In the study by Singh et al, nanoparticles with the diameters from ~270 to 420 nm crossed the blood-retinal barrier in laser induced CNV mice retina (Figure 2; *Gene Ther*, PMID: 19194480).

Additionally, the surface Zeta potential of NP-FB23-2s was -26.31 ± 3.09 mV (**Figure 11D**). The negative surface charge will make the nanoparticles to penetrate through the retina more easily (Lyu et al, *Biomaterials*, PMID: 33529961; Xia et al, *Acta Pharm Sin B*, PMID: 35646523). We hope the reviewer find it acceptable.

Shown in Fig11I, OIR+NP-FB23-2 moderately but significantly reduced the NVT area. The error bar is outstanding. Please provide individual values to show data distribution. Did it also rescue the permeability, pericyte dropout and neuroinflammation?

Thanks for your comment. As suggested by the reviewer, we have shown all individual values in all figures with bar graphs. Our data have demonstrated the therapeutic efficacy of NP-FB23-2 on retinal neovascularization in OIR mice. In the present study, we selectively focused on the roles of NP-FB23-2 on vascular endothelial cells, especially its suppressive effects on neovascularization. We herein didn't test its role in inhibiting permeability, pericyte dropout and neuroinflammation in STZ mice since in this study they are more likely to be secondary effects through cross talks between endothelial cells overexpressing FTO and other retinal cells. Another reason is the space and time limitations. STZ mice show consistent, obvious and severe

retinal phenotypes since the age of 6 months. Thus, the studies will definitely take more than 3 months, which is in conflict with the journal's request to submit a revision within 3 months. However, we believe the reviewer has raised an important concern. We will keep working on that in our further work. We hope the reviewer find it acceptable.

Other minor comments:

Line 477. TEM is transmission electron microscopy, not transition.

Thank you for your comment. We have corrected the relevant statement.

Line 693. Continuous i.p. Injection? Do the authors mean daily?

Yes. Sorry for the confusion caused. We have modified the statement to:

“STZ mice were generated by daily intra-peritoneal injection of STZ (50 mg/kg; Sigma-Aldrich) for 5 days.”

Referee #2 (Comments on Novelty/Model System for Author):

This is an excellent paper. However, FTO therapy to modify mRNA methylation may have a broad impact on mRNA metabolism and protein translation. Therefore, safety is a potential concern and needs to be thoroughly investigated.

Referee #2 (Remarks for Author):

This study by Chen et al. sheds new insight into FTO's role in DR via demethylation of m6A that can affect ECs, EC-pericytes, and EC-microglia interactions. Also, a selective inhibitor of FTO, FB23-2 has been studied as a potential therapeutic agent against DR. Overall, the manuscript has a good flow with ample mechanistic data, and each conclusion was drawn after extensive validations to provide novel insights into the molecular mechanism of FTO involvement in PDR.

Here are some comments:

1. The authors showed that high glycemc conditions lead to reduced m6A levels and increased FTO levels. And overexpression of FTO leads to reduced m6A levels. For this experiment, HUVECs were grown under high glucose (25 mM) in vitro. A previous study has shown that high glucose can inhibit HUVEC growth (PMID: 8932990). So, it would be helpful to compare the HUVEC viability in NG vs HG conditions and normalize the m6A methylation and other enzyme expressions accordingly.

Thank you for your comment. We have also noticed that many studies identified inhibited vascular endothelial cell growth upon high glucose treatment. However, promoted proliferation, migration and tube formation of vascular endothelial cells have also been reported (Zhu et al, *EBioMedicine*, PMID: 31636010). Herein, as indicated by the reviewer, we used the CCK-8 assay to detect the cell viability of HUVECs upon high glucose treatment. Our

data suggested that HUVEC growth is not inhibited in high glucose condition (**Supplementary Figure S1**). We hope the reviewer will find it acceptable.

2. The relative mRNA expression of METTL3 is increased in HG compared to NG but lowered in STZ compared to ctrl. But the METTL3 protein levels are unchanged in all conditions. It would be helpful to explain this observation as well.

Thank you for your comment. We have repeated the qPCR assay and we found unchanged *METTL3* mRNA levels in all conditions, which is consistent with the immunoblotting results. We have updated the relevant figures (**Figures 1E-1F**), and we are sorry for the confusion caused.

3. It showed that FTO promotes endothelial cell cycle progression using independent techniques, including RNA-seq. It is unclear from the main text or from the methods section which RNA inputs (FTO overexpressed vs FTO KD or FTO overexpressed vs. ctrl) were used for RNA-seq analysis.

Thank you for your comment. As indicated by the reviewer, we have detailed the specific RNA inputs in the **Results** section under the subtitle of “***FTO promotes endothelial cell cycle progression and tip cell formation to facilitate angiogenesis in vitro***” as follows:

“A total of 1770 differentially expressed genes [Log_2 fold change (FC) >0.5 or <-0.5 ; $p<0.05$], consisting of 771 up-regulated and 999 down-regulated genes, were identified in HUVECs transduced with L-FTO compared to cells transduced with L-EV (**Figure 2A**).”

We hope you will find it appropriate now.

4. Tip cell markers are upregulated in L-FTO-treated HUVECs. In addition, it would be helpful to see tip cell markers' expression in FTO KD.

Thank you for your comment. As suggested by the reviewer, we have

tested tip cell markers' expression in cells transfected with FTO-siRNA. Decreased expression of tip cell markers, including *CXCR4*, *FSCN1*, *APLN*, *ESM1* and *PLAUR*, was detected by qPCR assay in HUVECs with FTO knocked down (**Figure 2M**). We hope you will find it appropriate now.

5. In the OIR experiment, AAV-FTO was injected at P12 and retinas were examined at P17. Five days might be too soon to assess the AAV-FTO expression in vivo. AAV in vivo takes at least 1-2 weeks for transgene to be expressed and reach to peak expression in ~4 weeks. Also in Fig S3, the timeline of examining AAV-Fto expression in mouse retinal vascular EC was not mentioned. Were neonatal or adult mice used in Fig. S3?

Thank you for your comment. As suggested by the reviewer, we have included the timeline of examining AAV-Fto expression in both infant and adult mice in **Supplementary Figures S4A** and **S4C**. We agree with the reviewer that "AAV in vivo takes at least 1-2 weeks for transgene to be expressed and reach to peak expression in ~4 weeks". However, in the OIR experiment, we are not able to wait until 4 weeks to observe the effects. Therefore, as to the infant mice, intra-vitreous injection was conducted at P12 and neural retinas were collected at P17, which was consistent with a previously reported protocol by Wu et al (Diabetic Retinopathy; Methods in Molecular Biology, vol 2678; PMID: 37326717). Immunoblotting assay revealed elevated Fto protein level in neural retina of infant mice receiving intra-vitreous AAV-Fto injection compared to mice injected with AAV-blank (**Supplementary Figure S4B**). We hope you will find it acceptable.

6. Line 208 "Collectively, our data suggested that FTO simultaneously suppressed healthy vascular network formation into the ischemic retina.". It is unclear how this conclusion was drawn.

Thank you for your comment and we are sorry for the confusion caused. We have removed the corresponding statement.

7. Authors used extensive validation to show that FTO regulates EC-pericyte, vascular inflammation, EC-microglia crosstalk, and neurodegeneration in vitro and in vivo. FFA, Texas red dextran and Evans blue assay showed increased leakage in STZ + AAV-FTO. Moreover, VE-cadherin and NG2 staining show increased discontinuity and reduced pericyte coverage of STZ + AAV-FTO retina, respectively. An additional comparison of ctrl vs STZ w/o AAV-blank or AAV-FTO for retinal leakage and pericyte coverage may help rule out any AAV-related effects.

Thank you for your comment. The first reviewer also raised a similar concern. As suggested by both reviewers, we have included STZ along groups in all *in vivo* experiments in **Figures 4 and 6**. We hope you will find it appropriate now.

8. Line 288 - "revealed that percentage of activated microglia, represented by reduced number of interactions of microglia". It is unclear how an increase in activated microglia correlates to reduced interactions.

Thanks for your comment. The first reviewer also raised a similar concern. Consistent to our quantifications in **Figures 6B and 6D**, the activation status of microglia in **Figure 5F** was also determined based on morphological assessments. Microglia showing an amoeboid morphology, typified by reduced interactions, was defined as activated microglia, which was consistent with previous studies (Figure 3A, Akhter et al, *Mol Immunol*, PMID: 33461764).

We have also updated the relevant statement in the **Results** section under the subtitle of "***FTO triggers vascular inflammation and regulates EC-microglia crosstalk in vitro***" as follows:

"Immunofluorescence staining of ionized calcium binding adapter molecule 1 (Iba-1), a calcium-binding protein that participates in membrane ruffling and phagocytosis of activated microglia, revealed that percentage of

activated microglia, represented by an amoeboid morphology, was increased upon co-cultivation with HUVECs overexpressing FTO (**Figure 5F**).”

We hope the reviewer will find it appropriate now.

9. The incidence of 586 m6A-hyper peaks upon FTO overexpression should be explained. It seems overall m6A is increased globally (465 hypo vs 586 hyper m6A) in MeRIP-Seq, but the dot blot assay of global m6A abundance shows a decrease upon FTO overexpression. This contradiction should be discussed.

We think the reviewer has raised a very important comment. To address the reviewer’s concern, we have discussed this issue with the technical support from Illumina. The overall m⁶A level does not depend solely on the number of hypo/hyper-methylated m⁶A peaks identified by MeRIP-Seq, but also on the overall change in fold enrichment of all m⁶A peaks. The number of identified hypo/hyper-methylated m⁶A peaks relies on the values of Log₂ FC and *p* we set. Similar to our findings, Wang et al identified more hypo-methylated (730) than hyper-methylated (846) m⁶A peaks in endothelial cells with METTL3 knocked down (*J Biomed Sci*, PMID: 32384926, Figure 4A). We hope the reviewer will find our explanation acceptable.

10. Other identified proteins including FDFT1 also involved in the cell cycle. So, it would be helpful to provide more insights into the rationale for selecting CDK2 for further studies.

Thank you for your comment. The *FDFT1* gene encoded protein is more involved in the cholesterol biosynthesis. As to cell cycle, *FDFT1* has been reported to regulate cell cycle of cancer cells but not vascular cells. Additionally, the pathological involvement of all the other 8 genes, including *FDFT1*, *LCLAT1*, *BIN1*, *RHOT2*, *HMGCS1*, *MSMO1*, *NSDHL*, and *PRRC2B*, in diabetic retinopathy or retinal neovascularization has never been revealed.

The *CDK2* gene encodes the cyclin dependent kinase 2 that participates

in cell cycle regulation, and its pathogenic role in diabetic retinopathy has been documented (Jiang et al, *J Clin Invest*, PMID: 32343678). We thus selected CDK2 for further studies. We have included the following statement in the **Results** section under the subtitle of “*FTO regulates CDK2 mRNA stability with YTHDF2 as the reader in an m⁶A-dependent manner*”:

“A total of 9 genes containing m⁶A-hypo peaks with altered protein expression were sorted out (**Figure 8E**). Among all, the CDK2 gene, encoding a serine/threonine protein kinase that participates in cell cycle regulation, was found involved in cell proliferation and DR (23, 41). However, the pathological involvement of all the other 8 genes in DR or retinal neovascularization has never been revealed. We thus select CDK2 as a potential downstream target of FTO in HUVECs for further investigations.”

However, we agree with the reviewer that other potential downstream targets and pathways of FTO in ECs may also exist. We therefore have included the following statement in the **Discussion** section:

“However, other potential downstream targets and pathways of FTO in ECs may also exist, which needs to be explored systematically.”

We hope the reviewer will find it appropriate now.

11. In Figure 8L, mutant FTO can be used as a control in RIP-qPCR assay.

Thanks for your comment. As suggested by the reviewer, we have conducted the RIP-qPCR assay in cells transfected with L-FTO^{MU}. According to our results, the binding was partly abandoned upon overexpression of the FTO mutant protein (**Figure 8L**). We hope the reviewer will find it appropriate now.

12. Line 408 "Actinomycin D was used to inhibit CDK2 transcription in HUVECs". It is unclear how Actinomycin D specifically inhibits CDK2 transcription.

Thank you for your comment. Actinomycin D non-selectively binds to DNA

and extensively inhibits the process of RNA transcription, including CDK2 transcription. As suggested by the reviewer and to avoid misunderstanding, we have changed the relevant statement as follows:

“Actinomycin D was used to inhibit the process of RNA transcription in HUVECs.”

13. In a previous study (PMID: 28400392), ECs isolated from fibrovascular membranes of PDR patients and healthy retinas were profiled by RNA-seq. Not sure whether FTO was upregulated in this study. The RNA-seq data available in NCBI should be mined and discussed.

Thank you for your comment. As suggested by the reviewer, we have analyzed the FTO expression using the RNA-Seq data. Transcriptomic analysis detected no difference in average FTO expression in CD31⁺ cells isolated from PDR patients-derived FVMs compared to retinas of non-diabetic post-mortem patients (**Figure X4**).

Figure X4. FTO expression in CD31⁺ cells isolated from retinas of non-diabetic post-mortem patients and PDR patients-derived FVMs.

One of our major concerns about this paper is that they recognize CD31⁺ cells as vascular endothelial cells in FVMs. CD31 is a surface membrane protein of both endothelial and immune cells and plays important roles in the interaction between the vascular and immune systems (Kim et al, *Circ Res*, PMID: 20634489; Zhang et al, *J Cereb Blood Flow Metab*, PMID: 37051650). Our group has previously identified microglia as the major cell

population in FVMs (Hu et al, *Diabetes*, PMID: 35061025). Thus, we think the CD31⁺ cells isolated from FVMs by Lam et al include not only endothelial cells but also large amounts of immune cells. Additionally, the sample size of this paper is also limited (N=4 for the control group, N=9 for the FVM group).

To better reveal the role of FTO in diabetic retinopathy, we analyzed and compared FTO's expression in different types of retinal cells using single cell RNA-Seq data obtained from STZ rats retina (GEO datasets: GSE209872). Single cell RNA-Seq data revealed that FTO was ubiquitously expressed in all types of retinal cells (**Figure X1A**). Further assessments revealed that differential expression of FTO between STZ and control rats is solely detected in retinal vascular endothelial cells (**Figure X1B**).

Figure X1

Figure X1. (A) Feature plot showing FTO expression in different types of retinal cells (GEO accession number: GSE209872). **(B)** Violinplots comparing FTO expression in different types of retinal cells between the STZ and control mice (GEO accession number: GSE209872).

We hope the reviewer will find our explanation acceptable.

Minor points

14. Minor details: In Fig S1B, No GFP signal. Only DAPI. Should GFP be changed to DAPI?

Thank you for your comment. We are sorry for the confusion caused. We have changed the label from “GFP” to “DAPI”.

15. In Fig. S3 legend title, "mice retinal vascular" should be changed to "mouse".

Thanks. We have changed the legend title as suggested by the reviewer.

16. "Proliferative membranes obtained from DR patients". If patients had PDR, "PDR patients" should be used for clarity.

Thank you for your comment. All FVMs were collected from patients with PDR. As indicated by the reviewer, we have updated relevant statements throughout the text.

17. FTO should be defined.

Thanks for your suggestion. The abbreviation FTO has been defined in the second paragraph of the **Introduction** section as follows:

"The fat mass and obesity-associated (FTO) protein, which mediates oxidative demethylation of different RNA species, acts as a regulator of fat mass, adipogenesis and energy homeostasis."

18. In Fig 6A, the reason for measuring *Il1b* and *Ccl2* expression levels is not clearly stated.

Thank you for your comment. We measured *Il1b* and *Ccl2* expression since they are pro-inflammatory genes. We have updated the relevant statement as follows:

"As revealed by qPCR assay, Fto overexpression aggravated retinal inflammation by elevating expression of pro-inflammatory genes *Il1b* (Rangaraju et al, *Mol Neurodegener*, PMID: 29784049) and *Ccl2* (Scholz et al, *J Neuroinflammation*, PMID: 26576678) (**Figure 6A**)."

We hope the reviewer will find it appropriate now.

Dear Dr. Liu,

Congratulations on a great revision! Overall, the referees have been positive. However there remain some minor concerns that we ask you to (non-experimentally) address in a revised version.

When you submit your revised version, please also take care of the following editorial items and add this also to your point-by-point response:

1. Please reduce the number of keywords to five.
2. Please add a Data Availability section as outlined in our online author guide.
3. Please add Jiansu "333" Advanced Talent-training Project to EJP online.
4. Please remove the author contribution section from the main manuscript.
5. Please review our new policy on conflict of interests on the author guide website and update the title of this section to: Disclosure and competing interests statement. Also please move this section to after the Acknowledgements.
6. Please add a section "The Paper Explained". See the author guide for more information and our website for examples.
7. For references, these should be listed alphabetically and 10 authors provided, followed by et al for papers with more than 10 authors.
8. For source data, please upload the completed checklist. Please provide source data for Figure 6E, L-P. Source data for Appendix figures should be zipped together in 1 file.
9. We include a synopsis of the paper (see website for examples). Please provide me with a general summary statement and 3-5 bullet points that capture the key findings of the paper.
10. We also need a summary figure for the synopsis. The size should be 550 pixels wide by 200-400 pixels high.
11. The figure legends do not comply with the journal guide. Please see our author guide and website for examples. A data information section is required in the legend of all figures.
12. Please provide N for figure 2a and 8a in the figure legends.
13. Please define the scale bar for figure 1j.

Thank you for the opportunity to consider your work for publication. I look forward to your revision.

Warm wishes,
Kelly

Kelly M Anderson, PhD
Scientific Editor
EMBO Molecular Medicine

*** Instructions to submit your revised manuscript ***

To submit your manuscript, please follow this link:

<https://embomolmed.msubmit.net/cgi-bin/main.plex>

- 1) a .docx formatted version of the manuscript text (including Figure legends and tables)
- 2) Separate figure files*
- 3) supplemental information as Expanded View and/or Appendix. Please carefully check the authors guidelines for formatting Expanded view and Appendix figures and tables at <https://www.embopress.org/page/journal/17574684/authorguide#expandedview>
- 4) a letter INCLUDING the reviewer's reports and your detailed responses to their comments (as Word file).
- 5) The paper explained: EMBO Molecular Medicine articles are accompanied by a summary of the articles to emphasize the major findings in the paper and their medical implications for the non-specialist reader. Please provide a draft summary of your article highlighting
 - the medical issue you are addressing,
 - the results obtained and
 - their clinical impact.This may be edited to ensure that readers understand the significance and context of the research. Please refer to any of our published articles for an example.
- 6) For more information: There is space at the end of each article to list relevant web links for further consultation by our readers. Could you identify some relevant ones and provide such information as well? Some examples are patient associations, relevant databases, OMIM/proteins/genes links, author's websites, etc...
- 7) Author contributions: the contribution of every author must be detailed in a separate section.
- 8) EMBO Molecular Medicine now requires a complete author checklist (<https://www.embopress.org/page/journal/17574684/authorguide>) to be submitted with all revised manuscripts. Please use the checklist as guideline for the sort of information we need WITHIN the manuscript. The checklist should only be filled with page numbers where the information can be found. This is particularly important for animal reporting, antibody dilutions (missing) and exact values and n that should be indicated instead of a range.
- 9) Every published paper now includes a 'Synopsis' to further enhance discoverability. Synopses are displayed on the journal webpage and are freely accessible to all readers. They include a short stand first (maximum of 300 characters, including space) as well as 2-5 one sentence bullet points that summarise the paper. Please write the bullet points to summarise the key NEW findings. They should be designed to be complementary to the abstract - i.e. not repeat the same text. We encourage inclusion of key acronyms and quantitative information (maximum of 30 words / bullet point). Please use the passive voice. Please attach these in a separate file or send them by email, we will incorporate them accordingly.
- You are also welcome to suggest a striking image or visual abstract to illustrate your article. If you do please provide a jpeg file 550 px-wide x 300-800px high.
- 10) A Conflict of Interest statement should be provided in the main text
- 11) Please note that we now mandate that all corresponding authors list an ORCID digital identifier. This takes <90 seconds to complete. We encourage all authors to supply an ORCID identifier, which will be linked to their name for unambiguous name identification.

Currently, our records indicate that the ORCID for your account is 0000-0003-1605-1964.

Link Not Available

Photos 400-800 DPI

*Additional important information regarding figures and illustrations can be found at <https://bit.ly/EMBOPressFigurePreparationGuideline>. See also figure legend preparation guidelines: <https://www.embopress.org/page/journal/17574684/authorguide#figureformat>

***** Reviewer's comments *****

Referee #1 (Comments on Novelty/Model System for Author):

This is an excellent paper that sheds new insights to FTO in vascular biology and in retinal vascular diseases such as diabetic retinopathy.

Referee #1 (Remarks for Author):

All my comments are addressed and satisfied.

Minor comments:

1. line 217: "... or the OIR mice injected with AAV-Fto" should be "AAV-blank".
2. For in vivo EdU experiment, I am curious if the authors have paid attention to neovascular tufts (NVTs). Presumably NVTs should have more EdU+ cells.

Referee #2 (Comments on Novelty/Model System for Author):

The technical quality is high. Finding is novel. The therapeutic impact against FTO may be limited due to its broad impact on mRNA metabolism. This is defined by the nature of the protein. Nonetheless, it's a valuable paper to advance scientific knowledge. The revisions have addressed all of my concerns.

Referee #2 (Remarks for Author):

The topic is excellent to scientists in the field. However, nonspecialists may still have difficulties to understand this complicated paper. I am satisfied with the revised manuscript.

*Editor****1. Please reduce the number of keywords to five.***

Thanks. We have reduced the number of keywords to 5.

2. Please add a Data Availability section as outlined in our online author guide.

Thanks. We have included the Data availability section in the current draft as requested.

3. Please add Jiansu "333" Advanced Talent-training Project to EJP online.

Thanks. We have included the Jiansu "333" Advanced Talent-training Project to EJP online. However, this project does not have a Grant Reference Number.

4. Please remove the author contribution section from the main manuscript.

Thanks. There is no author contribution section in the current draft.

5. Please review our new policy on conflict of interests on the author guide website and update the title of this section to: Disclosure and competing interests statement. Also please move this section to after the Acknowledgements.

Thanks. We have updated this section as requested by the editor.

6. Please add a section "The Paper Explained". See the author guide for more information and our website for examples.

Thanks. We have included the "The Paper Explained" section in the current draft.

7. For references, these should be listed alphabetically and 10 authors provided, followed by et al for papers with more than 10 authors.

Thanks. We have updated the format for references in the current draft.

8. For source data, please upload the completed checklist. Please provide source data for Figure 6E, L-P. Source data for Appendix figures should be zipped together in 1 file.

Thanks for your reminder. We have embedded all source data for Appendix figures in 1 file. As to Figure 6E, this is a diagram without source data. There is no Figures 6L-6P in our manuscript. BTW, we are not able to download the checklist for source data of our current manuscript. We would be grateful if the editor could send us the updated form.

Figure 6

9. We include a synopsis of the paper (see website for examples). Please provide me with a general summary statement and 3-5 bullet points that capture the key findings of the paper.

Thanks. We have included the synopsis of the paper in a separate manuscript file and uploaded the file.

10. We also need a summary figure for the synopsis. The size should be 550 pixels wide by 200-400 pixels high.

Thanks. We have uploaded the synopsis image as requested.

11. The figure legends do not comply with the journal guide. Please see our author guide and website for examples. A data information section is required in the legend of all figures.

Thanks. We have updated the Figure legend section as requested.

12. Please provide N for figure 2a and 8a in the figure legends.

Thanks. We have provided N for Figures 2A and 8A in the figure legends as requested.

13. Please define the scale bar for figure 1j.

Thanks. We have defined the scale bar for Figure 1J in the figure legend as requested.

Referee #1 (Comments on Novelty/Model System for Author):

This is an excellent paper that sheds new insights to FTO in vascular biology and in retinal vascular diseases such as diabetic retinopathy.

Referee #1 (Remarks for Author):

All my comments are addressed and satisfied.

Minor comments:

1. line 217: "... or the OIR mice injected with AAV-Fto" should be "AAV-blank".

Thanks. We have changed the word "AAV-Fto" with "AAV-blank".

2. For in vivo EdU experiment, I am curious if the authors have paid attention to neovascular tufts (NVTs). Presumably NVTs should have more EdU+ cells.

Thanks for your comment. We agree with the reviewer that NVTs will have more EdU⁺ cells. However, it is difficult to calculate the exact number of EdU⁺ cells in NVTs due to its disordered vascular structures. Moreover, we detected that such enrichment was not limited to the NVTs but also in other parts of the retina.

Referee #2 (Comments on Novelty/Model System for Author):

The technical quality is high. Finding is novel. The therapeutic impact against FTO may be limited due to its broad impact on mRNA metabolism. This is defined by the nature of the protein. Nonetheless, it's a valuable paper to advance scientific knowledge. The revisions have addressed all of my concerns.

Referee #2 (Remarks for Author):

The topic is excellent to scientists in the field. However, nonspecialists may still have difficulties to understand this complicated paper. I am satisfied with the revised manuscript.

2nd Jan 2024

Dear Dr. Liu,

Congratulations on an excellent manuscript, I am pleased to inform you that your manuscript has been accepted for publication in the EMBO Molecular Medicine. Thank you for your comprehensive response to referee concerns and for providing detailed source data. It has been a pleasure to work with you to get this to the acceptance stage.

I will begin the final checks on your manuscript before submitting to the publisher next week. Once at the publisher, it will take about 3 weeks for your manuscript to be published online. As a reminder, the entire review process, including referee concerns and your point-by-point response, will be available to readers.

I will be in touch throughout the final editorial process until publication. In the meantime, I hope you find time to celebrate!

Warm wishes,
Kelly

Kelly M Anderson, PhD
Scientific Editor
EMBO Molecular Medicine
